# Design of precision therapeutics for a CKD risk allele by targeting Shroom3-Rock interaction

Anand Reghuvaran [1,9], Ashwani Kumar [1,9], Qisheng Lin[2], Nallakandi Rajeevan [3], Khadija Banu[1], Zeguo Sun[4], Hongmei Shi[1], Gabriel Barsotti[1], EM Tanvir[1], John Pell[1], Sudhir Perincheri[5], Chengguo Wei [4], Bhavya Bharathan[1], Marina Planoutene[4], Anne Eichmann[6], Valeria Mas[7], Weijia Zhang [4], Lloyd G. Cantley [1], Leyuan Xu [1], Bhaskar Das[8,10], John Cijiang He [4,10] & Madhav C. Menon [1,10] ✉

Enhancer variants in Shroom3 associate with renal fibrosis (TIF), but with reduced albuminuria. Detailed mechanisms for these pleiotropic effects are unclear. Here, we focus on identifying the specific profibrotic Shroom3 motif and separating this from its anti-proteinuric function. Given the role for Rho-kinases (Rock) in TIF, and the interaction of Rock with Shroom3 ASD2-domain, we hypothesized that Shroom3-mediated Rock-activation is crucial for profibrotic function. To test this, we develop transgenic tools that overexpress wild-type- (WT-Sh3) or ASD2-domain deletion- Shroom3 (ASD2Δ-Sh3). During TIF, Shroom3 and Rock co-expression occur in injured tubular cells and fibroblasts. In tubular- & fibroblast- lines, ASD2Δ-Sh3 overexpression reduce Rock activation, and pro-fibrotic/pro-inflammatory transcripts downstream of TGFβ1/Wnt/Ctnnb1-signaling vs WT-Sh3. In vivo, inducible global-, or tubular-specific-, but not fibroblast-specific-, ASD2Δ-Sh3 overexpression mitigate TIF, vs WT-Sh3 overexpression. Importantly, ASD2Δ-Sh3 mice do not develop albuminuria, while overexpression of a distinct Fyn-binding deficient mutant Shroom3 (FBDM-Sh3) induces albuminuria. We then develop small molecule inhibitors of Shroom3-Rock interaction (P2Is) and confirm Rock inhibition with these agents in WT-Sh3 cell lines. Our lead P2I from these studies, BT1137, mitigates Rock-activation, profibrotic signaling and TIF in WT-Sh3 mice. Hence, we delineate the profibrotic Shroom3 motif and develop therapeutics for kidney disease from Shroom3 excess.

In native and allograft kidneys, chronic kidney disease (CKD) is characterized by progressive renal interstitial fibrosis and tubular atrophy (TIF) reflecting accruing damage. In humans, while genome-wide association studies (GWAS) have identified candidate CKD susceptibility loci[1,2], the detailed mechanistic basis of single-nucleotide polymorphisms (SNPs)-variant associations with renal

histology/function in CKD remains limited[3]. This has hindered the development of therapeutics based on GWAS. Specifically, targeting fibrosis mechanisms in the native or allograft CKD context has not successfully translated to the bedside in clinical trials, despite promising preclinical data[4]. As a result, a significant percentage of patients with susceptibility loci for CKD or allograft injury will

eventually progress to end-stage renal disease (ESRD) requiring dialysis or transplantation[5,6].

Within the Shroom3 gene locus, multiple intronic SNPs, including rs17319721, have been associated with CKD in GWAS[1,2,7]. We and others identified that the A-allele at rs17319721 has TCF7L2-dependent enhancer function on *Shroom3* in renal epithelial cells[8,9]. In renal allograft biopsies, enhanced *SHROOM3* expression with the donor A-allele was associated with increased TIF[8] and reduced glomerular filtration rate (eGFR) by 1-year post transplant[10]. We showed that A-allele cells increased *SHROOM3* expression in a Tcf7l2-dependent manner after TGFβ1 treatment, and more SHROOM3 promoted profibrotic marker production, creating a cross-talk between TGFβ1 and Wnt/Ctnnb1 pathways. In vivo, inducible global or renal-tubular specific *Shroom3* knockdown in mice significantly inhibited TIF[8]. These data suggested a pathogenic role for SHROOM3 in TIF, and potential for testing the role of SHROOM3-antagonism as a therapeutic strategy for TIF in CKD in A-allele kidneys. However, subsequent data from our/other groups showed a protective role for Shroom3 in glomerular development[11–13], and indicated the development of proteinuria when Shroom3 was knocked down in adult podocytes[14,15]. Together, these data suggest an adverse association for rs17319721 & elevated *Shroom3* with renal TIF in CKD, but also caution that globally antagonizing Shroom3 would incur the risk of proteinuria. These dichotomous responses to Shroom3 loss led us to carefully analyze the protein structure of Shroom3 and investigate specific domains that could independently regulate cytoskeletal organization or profibrotic signaling.

Shroom3 belongs to the SHROOM family of proteins[16] containing a PDZ domain and two Apx/Shrm-specific domains called ASD1 and ASD2[17]. The ASD1 domain directly binds F-Actin[11], while the ASD2 domain binds and activates Rho-associated coiled-coil kinases (ROCKs)[18]. We also reported a previously unidentified -PxxP- domain located between PDZ and ASD1-domains, which was critical for Fyn-binding and activation in podocytes, and that adult mice with Shroom3 knockdown phenocopied Fyn-KO mice, with inhibited Nphs1 phosphorylation and Actin cytoskeletal organization demonstrating that this motif was crucial in adult podocyte homeostasis[14]. Subsequently, we reported that Shroom3 knockdown in glomerular podocytes induced a minimal change-like injury (MCD), due to Fyn inactivation with collateral activation of AMPK[15], and inhibition of AMPK in this context induced podocyte loss and focal sclerosis. Notably, impaired Fyn activation in podocytes has also been specifically associated with human MCD[19]. These suggested that the Fyn-binding motif of Shroom3 mediated its antiproteinuric effects in adult animals, distinct from the developmental role in glomerulogenesis reported before[11,12].

In this work, we performed a series of experiments that focused on identifying and specifically targeting the profibrotic motif of Shroom3. We first developed deletional variants of each of the consensus domains of Shroom3 and screened them in vitro in multiple cell lines (based on single cell transcriptome data), to identify a key role for the Rock-binding, ASD2-domain in profibrotic signaling, facilitating Tgfβ1 and Wnt/Ctnnb1 pathways. These data were consistent with prior antifibrotic roles for the Rock-inhibitor, Hydroxy-Fasudil[20–22]. To then study the in vivo role played by Shroom3 motifs in TIF, we developed inducible, cell-type-specific overexpression mice to induce either intact Shroom3 excess (analogous to the risk-allele carriage with high expressor kidneys) or overexpression of a variant of Shroom3 without the ASD2-motif. We compared these in multiple TIF models and demonstrated that the ASD2-motif is the profibrotic motif during TIF. Based on in vivo findings with genetic deletion studies of the ASD2-domain, we designed, synthesized and screened small molecules that bind to Rho-kinases and prevent interaction with SHROOM3 (P2Is), and demonstrated efficacy of a lead P2I against TIF in vivo. Hence, we unravel underlying pathophysiologic mechanisms associating Shroom3 SNPs with CKD and allograft injury and show proof-of-principle for designing Shroom3-targeted precision therapeutics.

## Results

### ASD2Δ-Sh3-overexpression reduces ROCK activation and signaling

We previously reported that tubular-specific Shroom3 knockdown alone could mitigate TIF in a UUO model[8]. Recent functional genomics studies evaluating CKD-associated SNPs using single-cell ATAC-seq also concluded that single-cell, cis-eQTL effects of Shroom3 intronic variants are identifiable in proximal tubular cells (PTs) in human CKD[23]. These pointed to injured PTs (iPTs) as key players for TIF from excess Shroom3. We also considered that fibroblasts, which are key cells for matrix production during TIF, express Shroom3 and show high Rock1, Rock2 expression in published data from murine TIF models[24–27]. Podocytes show high expression of Shroom3 at rest and injury, but are not expected to play key roles in these TIF models after PT cell injury[24–26]. Hence, we focused on tubular and fibroblast cell lines, which were potential sites of Shroom3 interaction with Rock1 and/or Rock2 during TIF for subsequent mechanistic experiments.

To screen for the profibrotic domain in Shroom3[28], we first generated deletional mutants of Shroom3, each without one of the three known consensus domains (PDZ, ASD1, or ASD2) [1a]. The ASD2-domain is known to bind and activate ROCK1 and ROCK2[29]. We confirmed that ASD2-domain deletion in SHROOM3 (ASD2Δ-Sh3) abolished ROCK1- [Fig. 1b] and ROCK2 binding [Fig. S1a] in HEK-293T tubular cell lines vs other deletional mutants. We then screened each domain-deletion variant against WT-Sh3 (whole Shroom3) using TGFβ1/Smad reporter HEK-293T blue™ cells. In this assay, which detects secreted alkaline phosphatase in response to Smad3/4 signaling, ASD2Δ-Sh3 inhibited TGFβ1 signaling reporter at baseline and in the presence of TGFβ1 [S1b]. In HEK-293T cells, we co-transfected SMAD3/4-reporter luciferase constructs with either WT- or ASD2Δ-Sh3 plasmids and analyzed TGFβ1-signaling with/without TGFβ1. Again, the ASD2Δ-Sh3 construct significantly reduced TGFβ1-signaling vs WT-Sh3 [Fig. S1c]. We also evaluated other tubular lines- (mouse Inner medullary collecting duct [mIMCD3]) and fibroblasts (NIH/3T3) stably overexpressing either WT-Sh3- or ASD2Δ-Sh3 and studied Rock-activation and pro-fibrotic signaling. ASD2Δ-Sh3 overexpression reduced phospho-Mypt (pMypt) & total Rock1 as compared to WT-Sh3 and/or control vector [1c-d], altered the actin cytoskeleton [S1d-e] and reduced cell migration of IMCD cells [1e-f] vs WT-Sh3. In the presence of Rock-inhibitor HF (10 μM), the difference in migration between ASD2Δ-Sh3 cells vs. WT-Sh3 significantly narrowed [S1f] (although it was not abolished) [Figs. 1f vs S1g]. Transcripts downstream of Wnt/Ctnnb1- signaling, Tgfβ1-related profibrotic markers and pro-inflammatory chemokines were all significantly reduced in IMCD cells overexpressing ASD2Δ-Sh3, while WT-Sh3 overexpression resulted in increases of these transcripts vs. control-vector cells [1 g]. pMypt levels and gene-expression changes demonstrated that ASD2Δ-Sh3-cells slightly inhibited Rock-activation even when compared to the Control vector, suggesting a dominant negative effect of the ASD2Δ-Sh3 variant on endogenous Shroom3. WT-Sh3- induced pMypt excess and transcript changes were reduced by HF, suggesting these effects were mediated by Rock-activation [Fig. S1h, i]. Next, in 3T3 fibroblasts overexpressing ASD2Δ-Sh3 or WT-Sh3, ASD2Δ-Sh3 also inhibited pMypt [Fig. S1j], profibrotic marker production (albeit to a lesser extent) [1 h], and proliferation [1i] vs WT-Sh3-overexpression. Hence, we observed a key role of the ASD2 domain of Shroom3 on Rock binding/activation, impacting the cytoskeleton, cell migration, and profibrotic signaling in tubular-/fibroblast cells.

### Generation of inducible, global, ASD2Δ-Sh3 overexpression mice

To evaluate the role of the ASD2-domain of Shroom3 in vivo, we generated Doxycycline(DOX)-inducible (reverse tetracycline transactivator [rtTA] system), transgenic mice with Flag-tagged, ASD2Δ-Sh3, that could be mated with global (CAGS-rtTA- CMV early enhancer/

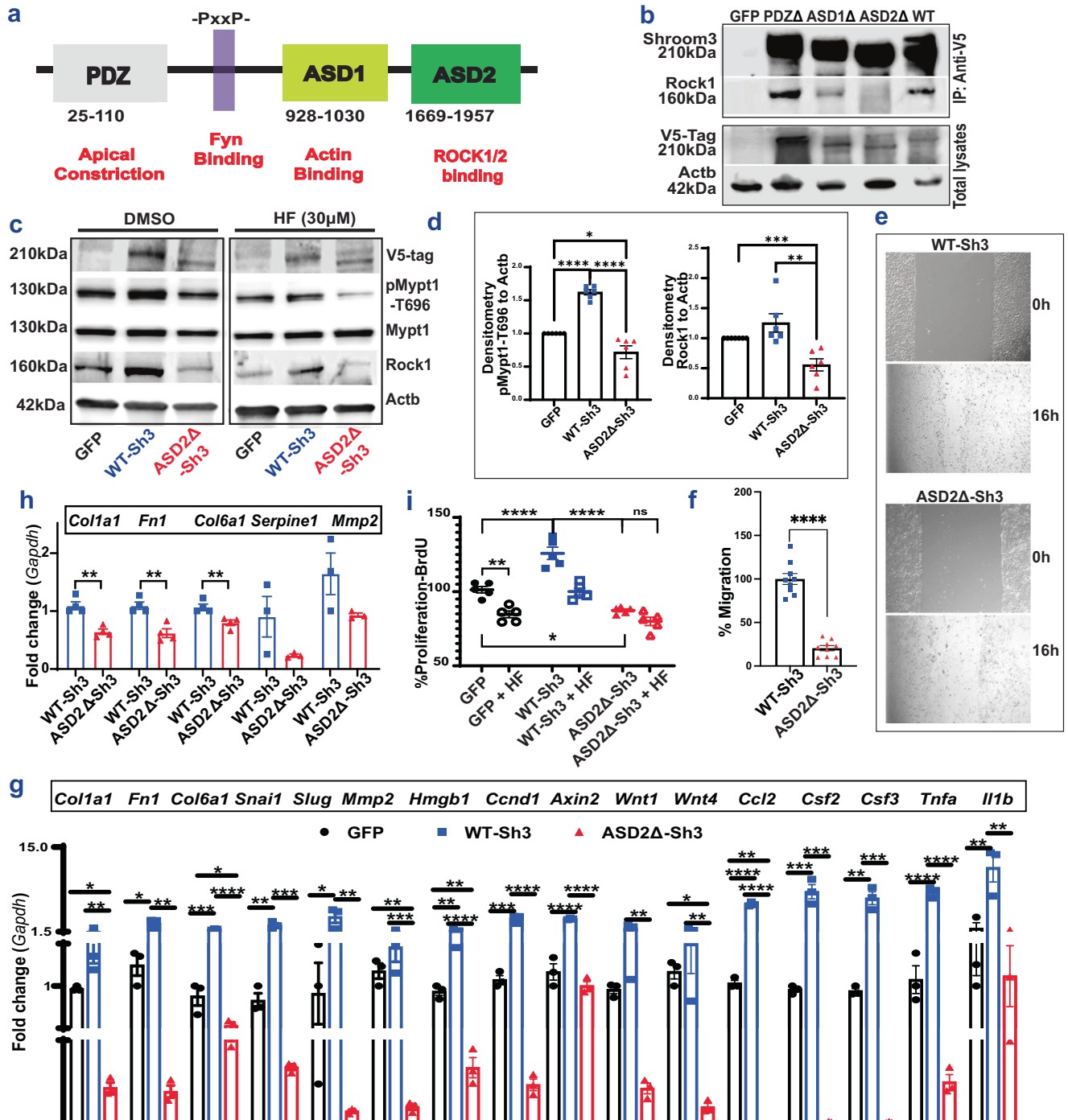

**Fig. 1 | ASD2Δ-Sh3-overexpression reduces ROCK activation in tubular cells and fibroblasts. a** The conserved domains of Shroom3 are shown (PDZ, ASD1, ASD2 domains and the Fyn-binding site). **b** Representative immunoblots of total cell lysates of HEK-293T overexpressing PDZΔ-Sh3, ASD1Δ-Sh3 or ASD2Δ-Sh3, which were immunoprecipitated with anti-V5 and probed for SHROOM3/ ROCK1. Total lysates were probed with anti-V5-tag or ACTB (one of 2 replicates). **c** Representative immunoblots from Dmso/HF-treated IMCD cells overexpressing GFP, WT-Sh3, and ASD2Δ-Sh3 probed for V5(Shroom3)/pMypt1/Mypt1/Rock1/Actb, and (**d**) relative density of pMypt1:Actb and Rock1 (*n* = 6 biological replicates; Unpaired T tests; ASD2Δ-Sh3 vs. GFP (*p* < 0.001), ASD2Δ-Sh3 vs. WT-Sh3 (*p* = 0.003)). **e** Representative phase contrast micrographs (4x) of scratches at 0 h and 16 h of IMCD cells over-expressing WT-Sh3/ ASD2Δ-Sh3 with (**f**) the corresponding bar graph quantifying the migrated area per field of ASD2Δ-Sh3 IMCD at 16 h (normalized to WT-Sh3 fields) obtained from 3 fields/well, & 3 wells/biological replicate

(Mann-Whitney *p* < 0.0001). **g** Bar graph representing the relative mRNA expressions of pro-fibrotic/Wnt signaling transcripts, inflammatory/injury- and tubular homeostasis- genes in WT-Sh3- vs ASD2Δ-Sh3 overexpressing IMCD cells (*n* = 3 biological replicates; One-way ANOVA Tukey's post test *p*-values are in the source data file). **h** Bar graph representing the relative mRNA expression of markers of profibrotic genes in 3T3 cells (Unpaired *T*-test *p*-values for *Col1a1*(*p* = 0.0017), *Fn1*(*p* = 0.0031), *Col6a1* (*p* = 0.009) (*n* = 4 biological replicates), and for *Serpine1*(*p* = 0.13), *Mmp2* (*p* = 0.12), respectively (*n* = 3 biological replicates). **i** Dot-plot shows BrdU incorporation per cell (as % of GFP-control) in 3T3 fibroblasts over-expressing WT-Sh3- vs ASD2Δ-Sh3/GFP-control with HF (hollow symbols) and without HF (solid symbols) (*n* = 5 biological replicates; One-way ANOVA with post-hoc Tukey's test *p*-values shown in source data file). [Line and whiskers indicate mean ± SEM; Two-tailed *p*-values were used and denoted as *\**p* < 0.05, **\**p* < 0.01, ***\**p* < 0.001, ****\**p* < 0.0001; HF = Hydroxy Fasudil].

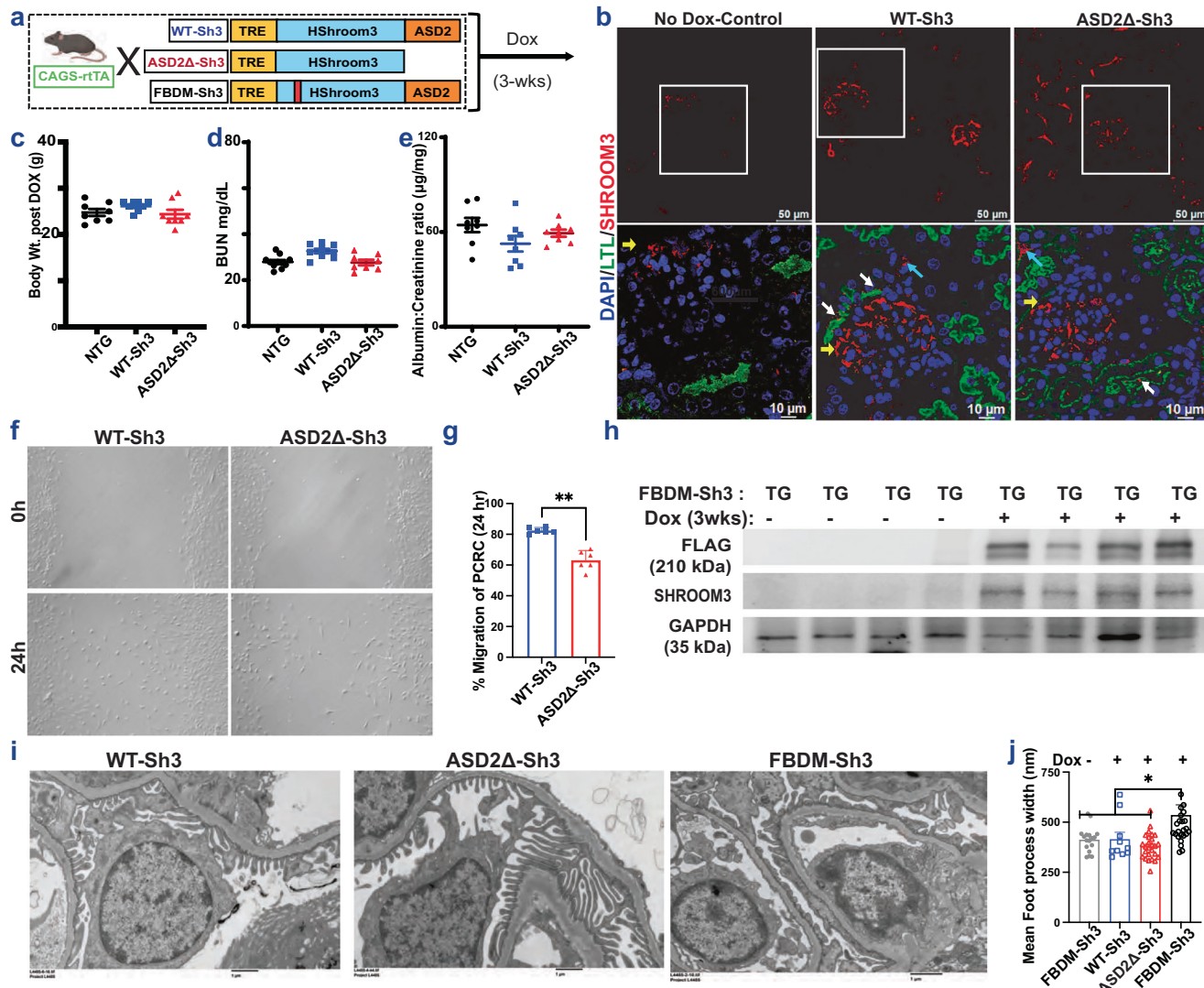

**Fig. 2 | Generation and phenotyping of global, ASD2Δ-Sh3 overexpression mice. a** Schema of generation of the Tetracycline-inducible global overexpression mice for WT-Sh3, ASD2Δ-Sh3 and FBDM-Sh3 (Created in BioRender. Caplan, M. (2026) https://BioRender.com/nxzeqir) **b** Validation of the overexpression by immunostaining for SHROOM3 (Red) in upper panel; The insets (lower panel) show co-staining with LTL-Green as white arrow(tubular), yellow(podocyte), and blue arrow (interstitial). Dot-plots of baseline measurements in Non-transgenic (NTG)/WT-Sh3/ASD2Δ-Sh3 mice of (**c**) body weights (grams; $n = 8$ animals/line) (**d**) blood urea nitrogen (BUN mg/dL; $n \geq 8$ animals /line) and (**e**) Albumin to creatinine ratio post Doxycycline induced overexpression ($n = 8$ animals/line; one-way ANOVA with Tukey post-test $p$-values for NTG vs WT-Sh3 ($p = 0.122$), NTG vs ASD2Δ-Sh3 ($p = 0.644$), WT-Sh3 vs ASD2Δ-Sh3 ($p = 0.491$)). In (**c–e**), blue symbols = WT-Sh3, red symbols = ASD2Δ-Sh3, black symbols = NTG (non-transgenic) mice, respectively. **f** Phase contrast micrographs of scratches at 0 h and 24 h on primary cortical renal cells (PCRCs) isolated from WT-Sh3/ ASD2Δ-Sh3 transgenic (TG) mice and (**g**) the corresponding bar graph

quantifies the average percent migrated area from 3 fields/well, including 2 wells/ mouse with 3 mice/group (Mann-Whitney $p = 0.002$). **h** Representative immu- noblots of whole kidney lysates from FBDM-Sh3 mice (Dox-fed and Normal chow- fed; $n = 4$ each) probed for Shroom3 and FLAG before or after DOX induction. **i** Representative electron micrographs (4800X) of glomeruli of WT-Sh3/ ASD2Δ- Sh3 vs FBDM-Sh3-TG mice to assess foot processes width (FPW). **j** In the bar graph, each dot represents the mean FPW measured per image per group from representative animals [($n = 1$, 1, 2 and 2 in FBDM-Sh3 (No Dox), WT-Sh3, ASD2Δ- Sh3 and FBDM-Sh3, respectively)]. FBDM-Sh3 ( + DOX) was compared with every other group by the Mann-Whitney Test, and exact $p$-values are shown in the source data file. Only free glomerular loops from ≥5 glomeruli in each animal were included for analyses. An average of $356.83 \pm 111$ foot processes per animal was counted [Line and whiskers indicate mean ± SEM; Two-tailed $p$-values denoted as *$p < 0.05$, **$p < 0.01$. DOX = Doxycycline, LTL = Lotus tetragonolobus lectin, FBDM = Fyn binding domain mutant].

chicken β-actin) or tissue specific-rtTA, and compared these to intact Shroom3-overexpressing mice (WT-Sh3 mice) [shown in Fig. 2a]. Both WT-Sh3 and ASD2Δ-Sh3 constructs were downstream of tetracycline response elements (TRE).

We induced overexpression of Shroom3 variants in these lines by DOX feeding [See "methods"]. Among multiple transgenic lines initially generated, a founder line each for ASD2Δ-Sh3 and WT-Sh3 was selec- ted for inducible- and optimal- (2–3-fold) overexpression of Flag- tagged Shroom3 variants in kidney cortex by 3 weeks DOX [Fig. S2a, b].

By immunofluorescence (IF) staining [Fig. 2b] in CAGS-rtTA mice, overexpression in glomerular, tubular and interstitial kidney cells was observed for each of these lines vs non-DOX fed littermates, with apical localization of Shroom3 in tubular cells (near Lotus tetragonolobus lectin (LTL) positivity), as described before[30]. By qPCR for exon- specific primers for the human ASD2-domain, only WT-Sh3 mice showed overexpression of this motif as expected [Fig. S2c]. These mice were born at normal rates, and inducible overexpression of ASD2Δ-Sh3 or WT-Sh3 by itself was not associated with weight loss up to 4-wks

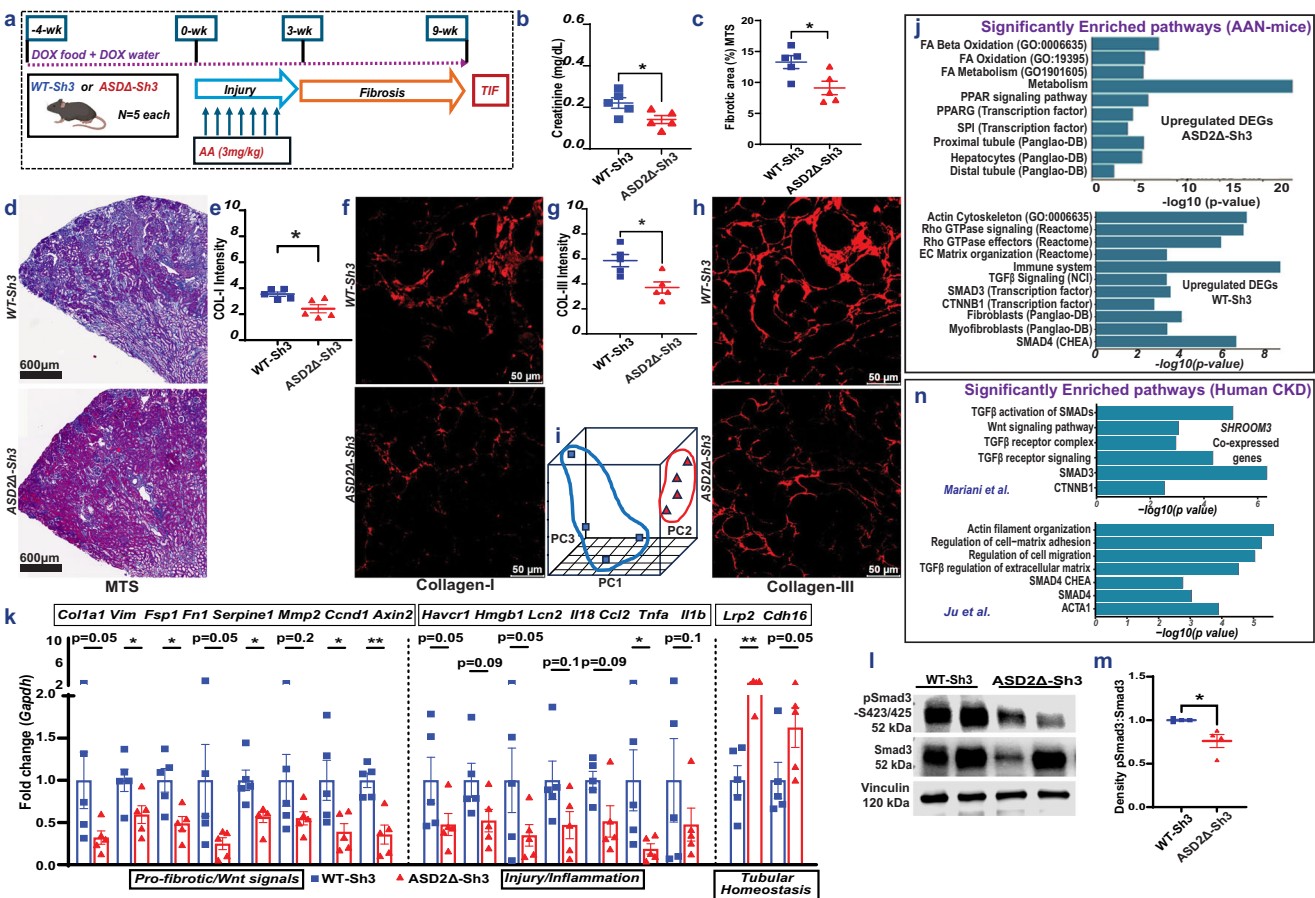

**Fig. 3 | Global ASD2Δ-Sh3 mice showing reduced TIF vs WT-Sh3 overexpression in AAN model. a** Schema of AAN in CAGS-rtTA/WT-Sh3 or ASD2Δ-Sh3 (*n* = 5 mice each; Created in BioRender. Caplan, M. (2026) https://BioRender.com/nxzeqir). Blue, red symbols depict WT-Sh3, ASD2Δ-Sh3 mice, respectively. **b** Dot-plot of serum creatinine levels in AAN mice at 9-weeks (*n* = 5 each; Unpaired *T*-test *p* = 0.039) (**c**) Dot-plot quantifications of blue-stained areas (10 fields/animal; *n* = 5 animals/group; Unpaired *T*-test *p* = 0.024) and (**d**) corresponding representative slide-wide images (4X) showing Masson's Trichrome staining (MTS) in AAN in CAGS-rtTA/WT-Sh3 vs ASD2Δ-Sh3. **e–h** Dot-plots showing (**e**, **g**) quantification of intensity of fluorescence and (**f**, **h**) corresponding representative immuno-fluorescence (IF) images (at 40X) for Collagen-I, and Collagen-III of AAN kidneys in CAGS-rtTA/WT-Sh3 vs ASD2Δ-Sh3, respectively (10 hpf/animal; *n* = 5 animals/group; Mann-Whitney *p*-values for Collagen-I (*p* = 0.032) and Collagen-III (*p* = 0.032)). **i** Bulk transcriptome evaluation of injured kidneys by RNAseq was performed at 9-wks CAGS-rtTA-WT-Sh3 or -ASD2Δ-Sh3 mice (*n* = 4 each). The read count data were analyzed by DESeq2 using a negative binomial generalized linear model. Principal component analysis plot shows clustering between the two

conditions. **j** Bar graphs showing significantly selected enriched pathways (EnrichR) with upregulated DEGs in ASD2Δ-Sh3 (upper) or WT-Sh3 (lower panel) (adjusted *p*-values from Fisher's exact test; *n* = 4 each). Fold changes/ *p*-values for upper and lower panels of DEGs are in Supplementary dataset S1a and S1b, respectively. **k** Bar graph representing the relative mRNA expressions of pro-fibrotic transcripts, injury/inflammation- and tubular homeostasis genes from AAN-kidneys of WT-Sh3 and ASD2Δ-Sh3 (*n* = 5 each; Unpaired *T*-tests or Mann-Whitney tests (if non-normal distribution) were used for comparison and exact p-values are in the source data file). **l** Representative immunoblots of p-Smad3/Smad3/Vinculin from kidney tissue lysates of WT-Sh3 vs. ASD2Δ-Sh3 mice (*n* = 2 each), and (**m**) corresponding dot-plots for quantification (Mann-Whitney *p* = 0.029; *n* = 4 mice/group) (**n**) Bar graphs show analogous pathways obtained from genes co-expressed with SHROOM3 in 2 human CKD datasets (adjusted *p*-values from Fisher's exact test; See references 32 & 34 for study details) [Line and whiskers indicate mean ± SEM; Two-tailed *p*-values denoted as *\*p* < 0.05, \*\**p* < 0.01; DEGs = Differentially expressed genes; hpf = high power field; AAN = Aristolochic Acid Nephropathy].

DOX [2c]. Both transgenic lines had similar blood urea nitrogen (BUN) and Creatinine levels [2d, S2d], and neither line showed increased albuminuria vs non-transgenic littermates [Fig. 2e]. We isolated and cultured primary renal cells (PCRCs) of ASD2Δ-Sh3 and WT-Sh3 after DOX-induction[31], and observed significantly reduced migration of ASD2Δ-Sh3 PCRCs vs WT-Sh3 PCRCs [2f–g], as seen in vitro with stably overexpressing IMCD-lines.

In parallel, we developed transgenic Fyn-binding domain mutant-Shroom3 overexpressing mice (FBDM-Sh3; with -AxxA- in place of -PxxP- at positions 442-445 between the PDZ- and ASD1-domains [see Figs. 1b[14], 2a], which were also mated with CAGS-rtTA animals. We confirmed overexpression of flag-tagged Shroom3 mutant in kidney cortex upon DOX [Fig. 2h], with overexpression in glomerular cells on IF [Fig. S2e]. While DOX feeding of adult CAGS-rtTA/ASD2Δ-Sh3 or CAGS-rtTA/WT-Sh3 mice did not induce albuminuria [Fig. 2e] or podocyte

foot processes effacement (FPE), inducible overexpression of FBDM-Sh3 did induce albuminuria [Fig. S2f] & FPE [2i–j]. Together, these data showed a role for the Shroom3 Fyn-binding domain in mediating the anti-proteinuric function of Shroom3 in podocytes and explain why global overexpression of WT- or ASD2Δ-Sh3 (both of which have the intact FBD) does not induce proteinuria or podocyte injury.

## Global ASD2Δ-Sh3 overexpressing mice show reduced TIF vs WT-Sh3 in AAN

To study the role of the ASD2-domain of Shroom3 in TIF development, we induced Aristolochic acid nephropathy model (AAN) in ASD2Δ-Sh3 and WT-Sh3. AAN was induced after transgene induction using a multi-dose regimen [see "methods"; Fig. 3a]. Mice were evaluated at 9-weeks, allowing for post-injury remodeling and TIF. At 9-weeks, CAGS-rtTA/ASD2Δ-Sh3 showed reduced azotemia vs CAGS-rtTA/WT-Sh3 mice

[Figs. 3b, S3a]. Both groups showed increased albuminuria after AAN induction vs baseline, but without significant differences between groups [Fig. S3b]. CAGS-rtTA/WT-Sh3 mice showed significantly increased TIF by Masson-trichrome stain [Fig. 3c, d], increased Collagen-I [3e–f], Collagen-III [3g–h], and Fibronectin [S3c–d] staining by IF, vs CAGS-rtTA/ASD2Δ-Sh3. In both mouse lines, kidney expression of Shroom3 was confirmed by qPCR (including ASD2-domain specific SHROOM3 and mouse Shroom3 primers) [Fig. S3e] and by IF [S3f]. Interestingly, WT-Sh3 mice showed reduced LTL-positive tubules with (a) more pronounced Shroom3 staining in tubular cell bases (instead of LTL-adjacent pattern seen in uninjured animals), and (b) greater interstitial Shroom3 expression, while ASD2Δ-Sh3 showed better preserved LTL-positivity and apical Shroom3 staining in tubules, consistent with reduced tubular injury in ASD2Δ-Sh3 at 9-weeks [S3f]. Consistent with in vitro data, lysates from injured and uninjured kidneys in ASD2Δ-Sh3 mice showed reduced pMypt and Rock1 vs WT-Sh3 lysates [Figure S3g].

To evaluate differentially expressed genes (DEG) between the kidneys of CAGS-rtTA/ASD2Δ-Sh3 & -WT-Sh3 during TIF, RNAseq was performed on total RNA from CAGS-rtTA/ASD2Δ-Sh3 & -WT-Sh3 kidneys at 9 weeks [see "methods"]. A principal component analyses plot showed clustering of ASD2Δ-Sh3 & -WT-Sh3 transcriptomes, demonstrating transcriptome-wide differences in TIF kidneys between these genotypes [Fig. 3i]. Differential gene expression was analyzed after aligning reads with both mouse (mm39) as well as aligning human orthologs of transcripts with the human (GRCh38) transcriptome to facilitate downstream enrichment analyses [see "methods"]. Significant DEGs identified by DESeq from both alignments are tabulated in Supplementary dataset S1a, b, respectively. In GRCh38 alignment, we identified 534 DEGs with a p-value threshold $p < 0.01$ [Fig. S3h, Supplementary dataset S1a, b]. Significantly upregulated DEGs in WT-Sh3 transcriptome (downregulated in ASD2Δ-Sh3) demonstrated enrichment of signals related to small GTPase/Rho-kinase activation, along with enhanced TGFβ1/Wnt-βCatenin signaling, extracellular matrix (ECM) production and fibrosis [3j]. Consistent signals of Rho-kinase activation and fibrogenesis were also identified when the top 500 upregulated DEGS in WT-Sh3 mice by fold change were evaluated [$p < 0.05$; Fig. S3k]. Conversely, upregulated DEGs in ASD2Δ-Sh3 demonstrated signals of preserved proximal tubular cell homeostasis and metabolism, suggesting reduced injury [Figs. 3j, S3h]. We also validated key mRNA signals by qPCR. Profibrotic gene transcripts related to Tgfβ1 or Wnt/Ctnnb1 signaling were significantly elevated in WT-Sh3 kidneys, while markers of tubular homeostasis were reduced [3k], a finding orthogonally confirmed by IF for Cdh16 (Figure S3i, j). Proinflammatory transcripts (chemokines, immune cell-related) also tended to be increased in WT-Sh3 vs ASD2Δ-Sh3 kidneys, similar to data in WT-Sh3-IMCD cell lines both by RNAseq and qPCR [Figs. S3h, 3k]. Phosphorylated Smad3 was significantly increased in lysates from WT-Sh3 kidneys vs ASD2Δ-Sh3 by immunoblotting [Fig. 3l, m].

To compare these gene expression changes in WT-Sh3 overexpression with human CKD, we performed co-expression analyses to identify genes that significantly correlated with Shroom3 from tubulo-interstitial transcriptomes of three human CKD cohorts within Nephroseq (R ≥ 0.4; 429 ± 83 genes)[32–34]. Enrichment analyses of co-expressed genes revealed significant enrichment of small-GTPase Rho-/TGFβ1-/Wnt-βCatenin- signaling consistent with data from WT-Sh3 overexpressing mice with TIF in our AAN model [Figs. 3n, S3l vs 3j, S3k] and showed clear translational relevance of our murine model.

### Global ASD2Δ-Sh3 overexpressing mice show reduced TIF vs WT-Sh3 in UUO
We evaluated a second model of TIF in CAGS-rtTA/ WT-Sh3 and -ASD2Δ-Sh3 mice – UUO as shown in Fig. 4a. UUO was induced after transgene induction with DOX as described before (see "methods")[35],

and obstructed kidneys were evaluated for TIF at 7 days. qPCR confirmed increases in human ASD2-domain specific primers in WT-Sh3 UUO kidneys [Fig. 4Sa], while mouse Shroom3-specific primers remained similar between UUO kidneys of both transgenic animals [Fig. S4b]. In the UUO model again, WT-Sh3 mice showed increased TIF by Trichrome staining [Fig. 4b-c], Picrosirius red staining for Collagens [4d–e] and Fibronectin by IF [4f–g, S4c; see "methods"] vs ASD2Δ-Sh3 mice. Immunoblotting of cortical lysates confirmed increased phosphorylation of Smad3 in WT-Sh3 animals [Fig. S4d, e]. Analogously, profibrotic, EMT-related, Wnt-signals and pro-inflammatory signaling gene transcripts were significantly increased in WT-Sh3 UUO kidneys vs ASD2Δ-Sh3 UUO kidneys [4 h]. These data demonstrated mitigated TIF in a second model by global over-expression of ASD2Δ-Sh3 compared to conditions of WT-Sh3 excess.

### Tubular-specific ASD2Δ-Sh3 mice show reduced TIF vs WT-Sh3
Based on the expression of Shroom3-Rock in tubular cells [Fig. 1a], and reduced TIF in post-tubular injury models, we crossed our Shroom3 transgenic mice with Pax8-rtTA (Paired box 8- rtTA) lines for tubular-specific over-expression. The mRNA expression analysis of the kidney cortex of these Pax8-rtTA-Shroom3 mice confirmed overexpression of ASD2-domain specific sequences in WT-Sh3 mice vs other lines [Fig. S5a] after DOX-induction. Dox-feeding did not induce increased creatinine/BUN or albuminuria in Pax8-rtTA/WT-Sh3 mice vs -ASD2Δ-Sh3 or non-transgenic (NTG) mice [Fig. S5b–d].

We then induced AAN after 4 weeks of transgene induction with Dox and compared Pax8-rtTA/WT-Sh3 mice vs ASD2Δ-Sh3 as well as Dox-fed non-transgenic littermates [Fig. 5a]. Immunostaining of AAN kidneys (9-wk) from Pax8-rtTA/WT-Sh3 mice & -ASD2Δ-Sh3 animals confirmed increased Shroom3 staining in tubular cells [Fig. S5e]. WT-Sh3 mice had greater azotemia [Figs. 5b, S5f] than ASD2Δ-Sh3 or NTG animals. Similar to CAGS-rtTA animals, Pax8-rtTA/WT-Sh3 with greater injury showed more cytoplasmic Shroom3 staining, while Pax8-rtTA/ASD2Δ-Sh3 tubules with reduced injury showed maintained apical localization of Shroom3 (in LTL-positive areas) [S5e], and preserved Cdh16 staining [Fig. 5c–f]. Pax8-rtTA/WT-Sh3 mice showed increased Collagen-I staining by IF [Fig. 5d–g; see "methods"], and more TIF by Trichrome staining [Fig. 5e–h] vs ASD2Δ-Sh3 mice or NTG. By qPCR, profibrotic gene transcripts related to Tgfβ1/Wnt/Ctnnb1 signaling and injury/inflammation were consistently elevated in Pax8-rtTA/WT-Sh3 kidneys, while markers of tubular homeostasis were higher in Pax8-rtTA/ASD2Δ-Sh3 kidneys [5i]. Pax8-rtTA/WT-Sh3 mice with greater overall injury & TIF also tended to have greater albuminuria [Fig. S5g].

In a second TIF model of UUO, we compared Pax8-rtTA/WT-Sh3 mice vs -ASD2Δ-Sh3 mice [Fig. 5j]. Here again, Pax8-rtTA/ ASD2Δ-Sh3 kidneys showed significantly reduced Collagen-I and Fibronectin by IF [Fig. 5k–n], and reduced TIF by trichrome staining [Figs. 5o, S5h]. Profibrotic and pro-inflammatory gene signals were significantly reduced in Pax8-rtTA/ ASD2Δ-Sh3 kidneys, while tubular homeostasis markers were better preserved [5p]. Hence, deletion of the ASD2-domain during Shroom3 excess in injured tubular cells provided protection from increased TIF observed in WT-Sh3 mice using either model of fibrosis.

### Fibroblast-specific ASD2Δ-Sh3 mice show similar TIF to WT-Sh3
To study the role of excess Shroom3 ASD2 domain during TIF development in fibroblasts, Shroom3 transgenic mice were crossed with Pdgfrb-rtTA (platelet-derived growth factor receptor beta-rtTA) lines to generate fibroblast-specific overexpression mice [Fig. 6a]. Pdgfrβ(+) cells isolated (by magnet-assisted cell sorting (see "methods")) were tested for enrichment by qPCR for Pdgfrb [Fig. S6a]. The mRNA expression of the ASD2-domain sequence was higher in the Pdgfrβ(+) cells obtained from DOX-induced, Pdgfrb-rtTA/WT-Sh3 kidneys vs -ASD2Δ-Sh3 [Fig. S6b]. Mice from both lines, after inducing transgene expression, were subjected to right UUO to study TIF [Fig. 6a].

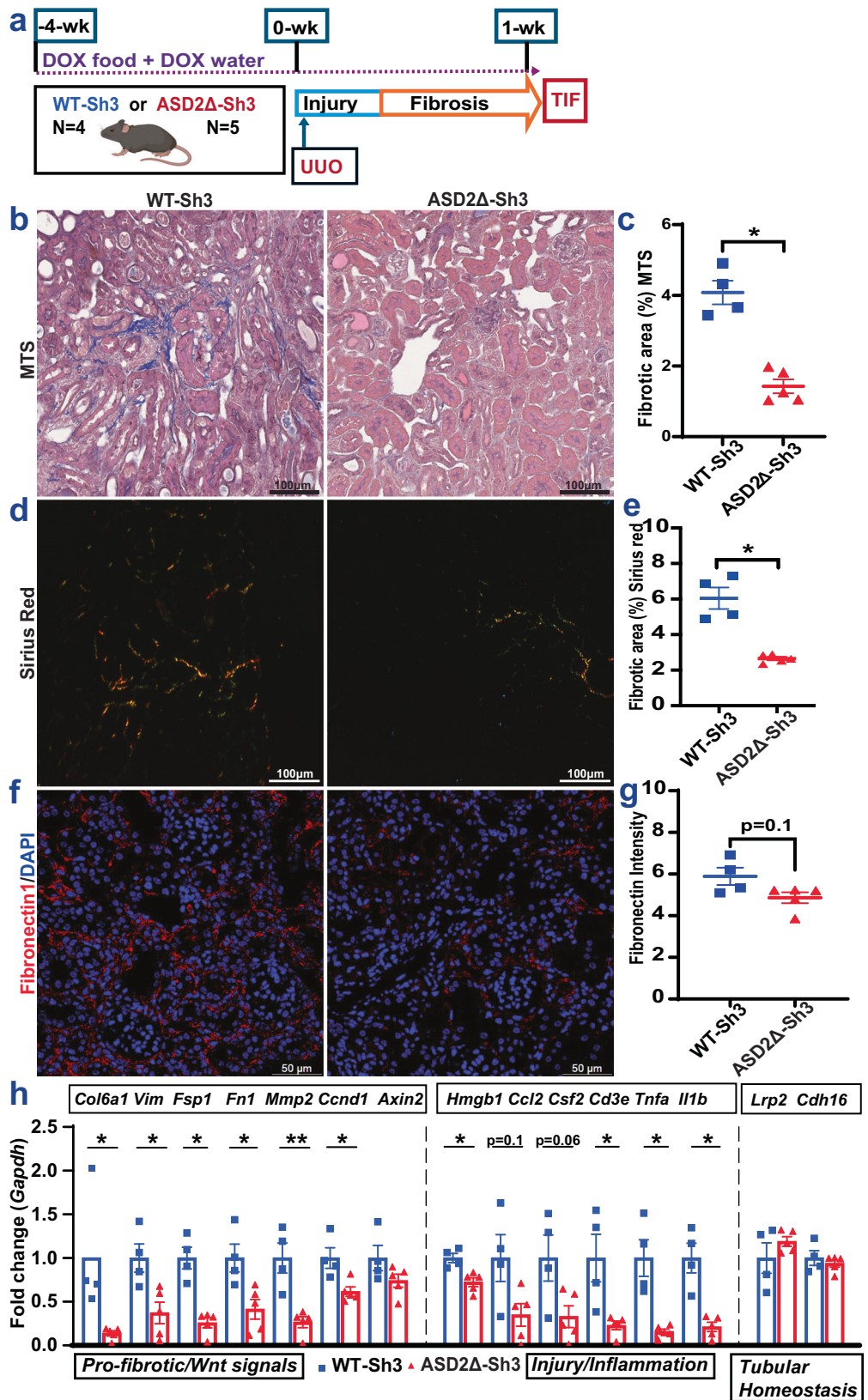

The obstructed kidneys showed no significant differences between the two lines in TIF as demonstrated by trichrome staining [Fig. 6b, c], Collagen-I [6d–e] and Collagen-III [6f–g] immunostaining. The gene transcripts of profibrotic and pro-inflammatory markers, and Wnt/Ctnnb1 signaling were also similar between WT-Sh3 vs ASD2Δ-Sh3

[Fig. 6h]. Hence, these data suggested that enhanced Shroom3-Rock interaction within injured tubular cells in global- or tubular-specific-WT-Sh3 mice is the main source of increased TIF in conditions of Shroom3 excess, while the contribution from fibroblasts alone is insufficient to significantly impact TIF after UUO.

**Fig. 4 | Global ASD2Δ-Sh3 showing reduced TIF vs WT-Sh3 overexpression in UUO model. a** TIF was induced by UUO model in CAGS-rtTA/WT-Sh3 (*n* = 4) or -ASD2Δ-Sh3 (*n* = 5) mice (Created in BioRender. Caplan, M. (2026) https://BioRender.com/nxzeqir) Blue symbols depict WT-Sh3 and red depict ASD2Δ-Sh3 mice. **b** Representative images as well as (**c**) corresponding dot-plot quantifying area of blue stain on Masson's Trichrome staining (MTS; obtained at 20X magnification) in UUO kidneys from WT-Sh3 vs ASD2Δ-Sh3 are shown (> 10 fields/animal; *n* = 4 vs. 5 in WT-Sh3 vs ASD2Δ-Sh3 groups, respectively; Mann-Whitney *p* = 0.016) (**d**) Representative images and (**e**) dot-plots quantifying picrosirius red staining (by plane polarized light at 20X) are shown to evaluate collagen deposition in UUO kidneys (8–10 fields/animal; *n* = 4 vs. 5 WT-Sh3 vs ASD2Δ-Sh3, respectively; Mann- Whitney *p* = 0.016) (**f**) Representative fluoromicrographs (40X), as well as (**g**) dot-plots are shown quantifying intensity of Fibronectin by immunofluorescence (8 hpf/animal; *n* = 4 vs. 5 WT-Sh3 vs ASD2Δ-Sh3, respectively; Mann-Whitney *p* = 0.1). **h** Bar graph representing the relative mRNA expressions of pro-fibrotic transcripts, injury/inflammation- and tubular homeostasis genes in the UUO kidneys between WT-Sh3 and ASD2Δ-Sh3 mice (Unpaired *T*-tests or Mann-Whitney tests (if non-normal distribution) were used for comparison and exact p-values are given in the source data file; *n* = 4 vs. 5 in WT-Sh3 vs ASD2Δ-Sh3, respectively). [Line and whiskers indicate mean ± SEM; Two-tailed *p*-values denoted as *\*p* < 0.05, *\*\*p* < 0.01. hpf = high power field; UUO = Unilateral ureteral obstruction].

## Synthesis and screening of Shroom3-Rock interaction inhibitors

Based on these experiments showing that inhibiting Shroom3-Rock interaction in conditions of global- or tubular- Shroom3 excess could ameliorate TIF, we designed and synthesized Shroom3-Rock interaction inhibitors (P2Is) [named D1–D7; see "methods" & supplementary materials[36]]. We developed these initial seven P2Is using computational docking with existing protein data bank (PDB) ID: *4L2W* of ROCK1, selecting the lowest energy poses for the most favorable interactions between the ligands and the protein. Figure S7a shows the structure of our hit compound BT-584 from initial screening (D4, among D1-D7). As an example, the docking of our lead compound BT-1137 (D4G) and its interaction sites with the Shroom binding domain (SBD) of ROCK1 are depicted in Fig. 7a, b. [D4G = potassium trifluoro (4-(2-thioxo-1,2,3,4,5,6-hexahydrobenzo [h] quinazolin-4-yl) phenyl) borate].

D1–D7 were initially screened using Rock activity assay (see "methods"), and pMypt1 levels in HEK-293T cells overexpressing WT-Sh3 and compared with Rock-inhibitor Fasudil (HF as a positive control for ROCK inhibition). ROCK activity assay on Wt-Sh3-overexpressing HEK-293T cells with the compounds D1–D7 (50 μM) showed that compounds D3−5 & D7 had more potent ROCK inhibition [Fig. 7c]. D4 (BT584; mol wt = 337.35) showed potent inhibition of Rock activity (65% inhibition vs DMSO, & similar to HF (30 μM), respectively). In IMCD cells, the pMypt1 levels were dose dependently reduced by D4 (at 50-& 100 μM), compared to D5−D7 [Fig. S7b].

Using structure-activity relationship studies (SAR) [shown in Fig. S7c], we next developed 7 derivative compounds of our hit compound D4 (called D4A-D4G) with the goal of improving potency. These compounds were tested for efficacy- and toxicity at 50 μM in both WT-Sh3-IMCD and WT-Sh3-3T3 cells and compared with HF (30 μM). WT-Sh3-IMCD cells treated with D4D-D4G inhibited pMypt1 levels, similar to HF-treated, WT-Sh3-IMCD cells, while Rock1 levels were significantly reduced by D4G alone [Fig. 7d–f]. Only D4F and D4G showed significant inhibition of profibrotic marker and cytokine transcripts (similar to studies with ASD2Δ-Sh3) in IMCD cells [Fig. S7d]. As Rock inhibition has been associated with suppression of cell cycle[37], cytokinesis[38,39] and cell proliferation[40] we checked the effect of P2Is on WT-Sh3-3T3 proliferation. All compounds significantly inhibited 3T3 proliferation (vs DMSO), although D4G tended to have a greater effect [Fig. 7g]. In toxicity studies, D4D showed the greatest toxicity in IMCD cells and in fibroblasts (Cell-glo titer ATP assay) [Figs. 7h, S7e, respectively]. D4E, F & G were similar to HF in ATP assays. These studies at a 50 μM concentration of D4A-G are summarized in the heatmap in Fig. 7i.

Based on these, we carried forward D4E, F & G for both efficacy and toxicity testing at progressively lower concentrations (1–10 μM). Here, D4G showed more potent reduction in pMypt1 [Fig. S7f] and reduced markers of Wnt-signaling by qPCR in WT-Sh3-IMCDs [Fig. 7j]. Similarly, in WT-Sh3-3T3, D4G showed consistently reduced pMypt1 levels [Fig. S7g], cell proliferation and profibrotic marker production [S7h, i, respectively]. Toxicity studies showed that D4E, -F, &-G had significantly lower toxicity at 10 μM compared to HF-10 μM in WT-Sh3 IMCD [Fig. 7k] & 3T3 (7 l). Hence, from these studies, which are summarized in Fig. 7l, D4G emerged as our lead compound.

We confirmed the ability of D4G to inhibit Shroom3-Rock interaction. Here, lysates from HEK-293T cells overexpressing WT-Sh3 treated with vehicle- or D4G (10 μM), were immunoprecipitated with anti-V5 antibody. These were compared with ASD2Δ-Sh3 overexpressing HEK-293T lysates, similarly immunoprecipitated with anti-V5. As shown in Fig. 7m, the interaction of SHROOM3 with ROCK1 was diminished with D4G-treatment vs DMSO-treatment in WT-Sh3 overexpressing HEK-293T and absent with ASD2Δ-Sh3. Notably, the interaction of SHROOM3 with ACTIN−an ASD1-domain mediated effect, was not affected by D4G or ASD2Δ-Sh3 [Fig. 7m]. To evaluate any paralog-specific effects of D4G, we knocked down either Rock1 or Rock2 in WT-Sh3 HEK-293 cells. After confirming ROCK1 and ROCK2 knockdown [Fig. 7n], we performed the ROCK activity assay using D4G. In these experiments, D4G inhibited ROCK1 and ROCK2 nearly identically [7o], suggesting the lack of any preferential inhibition of either paralog by our lead P2I. Finally, using a kinase hot spot assay done in the *absence* of SHROOM3, to compare D4G with HF, we evaluated in a targeted manner (a) Rock1, Rock2 inhibition, (b) inhibition of off-target kinases, i.e., serine-threonine kinases previously reported inhibited with HF[41,42], (c) Fyn (src-family tyrosine kinases related to podocyte function[14,43]. HF inhibited Rock1, Rock2 (as expected), but showed significant off-target kinase inhibition (except Fyn), while D4G did not inhibit any kinase in the absence of Shroom3 [Fig. S7j], suggesting specificity for inhibition of P2I.

## D4G (P2I) ameliorated TIF progression in Tubular-Specific WT-Sh3 mice

Based on in vitro *Rock activity assay*, using D4G on WT-Sh3 IMCD cells demonstrating EC50 (half maximal efficacy concentration) of 57 μM [Fig S8a], and IC50 (half maximal inhibitory concentration) of 613.2 μM [Fig. S8b], we next performed pilot UUO experiments to test the in vivo anti-TIF effect of D4G using an empiric initial D4G dose of 50 mg/kg (*n* = 6 each), which corresponded to 2.5 times EC50, but was well below IC50. Dox-induced tubular-specific overexpression, Pax8-rtTA/WT-Sh3 mice were given D4G, 50 mg/kg intra-peritoneally (IP), or Vehicle after the induction of UUO, and TIF was studied [Fig. 8a]. No weight loss or humane concern was seen with D4G treatment [Fig. S8c]. No increase in albuminuria was seen with D4G vs vehicle [Fig. S8d]. UUO kidney lysates from mice given D4G showed reduced pMypt1, pSmad3, Rock2 [Fig. 8b–e], and Rock1 [Fig. S8e, f] in kidney lysates, with significantly reduced fibrosis shown by trichrome staining [Fig. 8f, g], and IF for Collagen-I, Collagen-III and Fibronectin deposition [Fig. 8h–m] vs vehicle-treated UUO kidneys. Similar to UUO kidneys of ASD2Δ-Sh3 mice, D4G-treatment reduced profibrotic Tgfb1/Wnt/Ctnnb1 signals, and proinflammatory transcripts, and relatively preserved markers of tubular homeostasis [Fig. 8n]. Analogously, Cdh16 protein levels were also greater with D4G by IF [Fig. S8g, h]. Non-obstructed left kidneys from D4G-mice did not demonstrate increases in mRNA expression of injury markers, vs vehicle [Fig. S8i], nor did they exhibit overt injury in PAS-stained sections [Fig. S8j]. In summary, based on our sequential screening, we successfully showed anti-fibrotic efficacy of Shroom3-Rock P2Is in vivo for TIF in mice with WT-Sh3 overexpression.

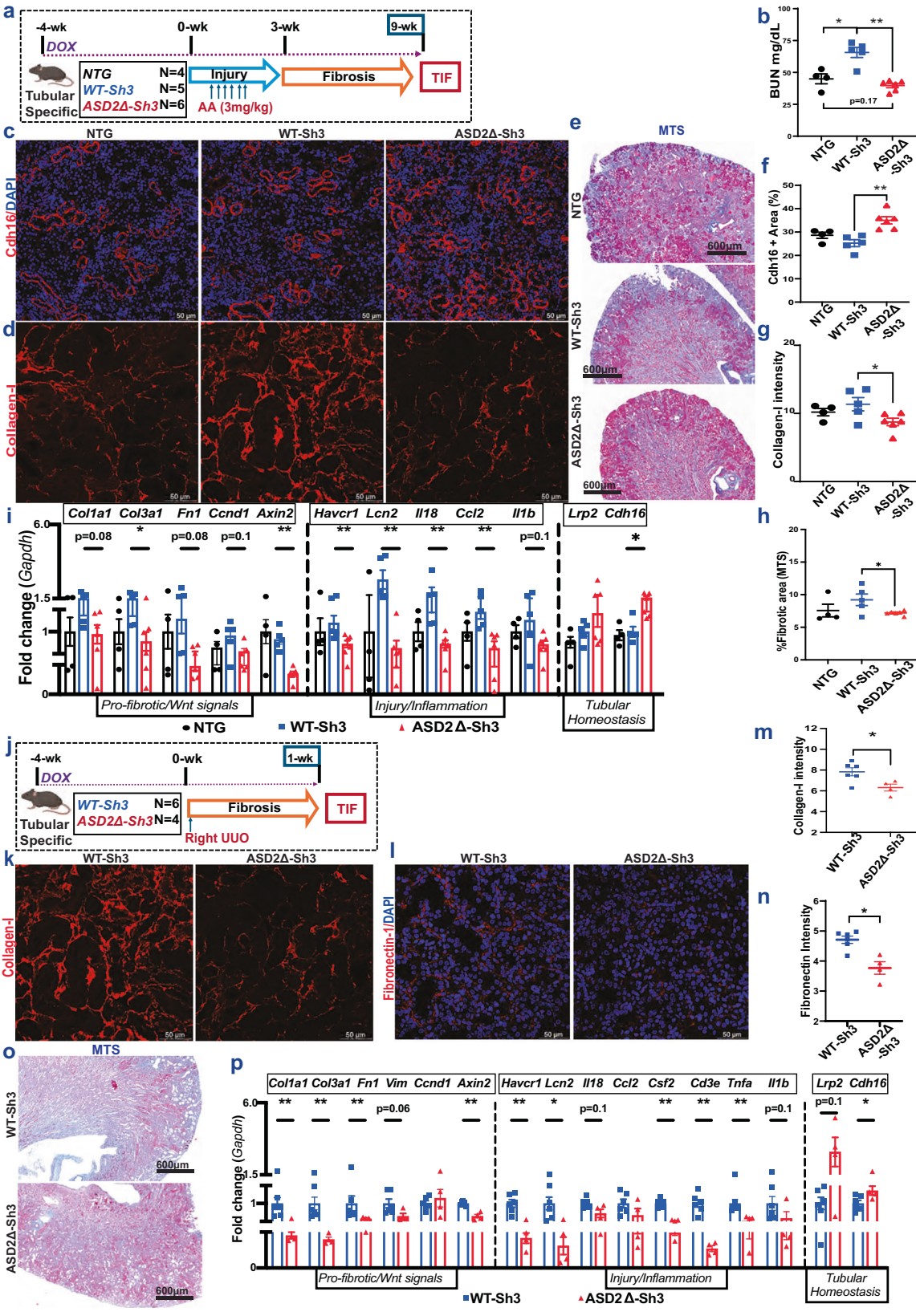

Taken together, these data demonstrate that in a milieu of excess SHROOM3 in injured tubular cells, ASD2-domain-Rock interaction plays a key role in the development of TIF and progressive CKD by promoting proinflammatory and profibrotic signaling. We show translational relevance of our findings to human CKD, and in pilot studies show efficacy of a P2I (D4G) to target this protein-protein interaction [Fig. 9].

## Discussion

While multiple loci related to CKD have been identified by GWAS, few loci/genes from unbiased analyses have made significant progress towards precision medicine approaches. For instance, intronic variants in the Shroom3 gene have been repeatedly identified as associated with CKD[23,44,45]. Using mechanistic studies, we previously showed that

**Fig. 5 | Tubular-specific ASD2Δ-Sh3 mice showing reduced TIF vs WT-Sh3.**
**a** Schema of AAN injury/TIF in Pax8-rtTA/WT-Sh3 (*n* = 5) or -ASD2Δ-Sh3- (*n* = 6) or NTG- (*n* = 4) mice. Blue, red, and black symbols represent WT-Sh3, ASD2Δ-Sh3, or NTG, respectively. **b** Dot-plot showing BUN in NTG, WT-Sh3, and ASD2Δ-Sh3 mice in AAN mice at 9 weeks (Mann-Whitney test *p*-values for each two-group comparison are in the source data file) (**c–h**) Representative fluoromicrographs of (**c**) Cdh16 IF (20X), (**d**) Collagen-I IF (40X) and (**e**) slide-wide photomicrographs (4X) of Masson's Trichrome staining (MTS) to evaluate TIF. Corresponding dot-plots quantify (**f**) the area of Cdh16, and in (**g**) the mean fluorescence intensities (12 hpf /animal) of Collagen-I, while in (**h**) the blue-stained area (10 hpf/animal) in Trichrome-stained sections. Statistical analyses in (**f–h**) compare WT-Sh3 (*n* = 5) vs ASD2Δ-Sh3 (*n* = 6) by Unpaired *T*-tests, and exact *p*-values are in the source data file. **i** Bar graph representing the relative mRNA expressions of pro-fibrotic/Wnt signaling-, inflammatory/injury- and tubular homeostasis- genes from kidneys tubular-specific WT-Sh3, ASD2Δ-Sh3, and NTG mice in AAN (Unpaired *T*-tests or Mann-Whitney tests (if non-normal distribution) used for each two-group comparison with exact p-values in source data file; *n* = 4 vs 5 vs 6 mice in NTG, WT-Sh3 & ASD2Δ-Sh3, respectively). **j** TIF was induced by UUO model in Pax8-rtTA/WT-Sh3- (*n* = 6) or -ASD2Δ-Sh3- (*n* = 4) mice. **k–n** Representative fluoromicrographs (40X) of (**k**) Collagen-I IF (40X), (**l**) Fibronectin IF (40X) are shown. Corresponding dot-plots quantify mean fluorescence intensities (8–10 hpf/animal) of (**m**) Collagen-I (Mann-Whitney *p* = 0.038), and (**n**) Fibronectin (Mann-Whitney *p* = 0.019) staining WT-Sh3 vs ASD2Δ-Sh3 (*n* = 6 vs. 4 mice). **o** Representative slide-wide photomicrographs of Masson's Trichrome staining (MTS) to evaluate fibrosis in UUO kidneys of WT-Sh3/ASD2Δ-Sh3 mice. **p** Bar graph representing the relative mRNA expressions of profibrotic/Wnt signaling-, and inflammatory/injury- and tubular homeostasis- genes from UUO kidneys of tubular-specific WT-Sh3/ASD2Δ-Sh3 mice (Unpaired *T*-tests or Mann-Whitney tests (if non-normal distribution) were used for comparison with exact *p*-values in source data file; *n* = 6 vs. 4 WT-Sh3 vs ASD2Δ-Sh3, respectively). [Line and whiskers indicate mean ± SEM; Two-tailed *p*-values denoted as *$p < 0.05$, **$p < 0.01$; NTG = non-transgenic; IF = Immunofluorescence]. 5a, 5j created in BioRender. Caplan, M. (2026) https://BioRender.com/nxzeqir.

the rs1721931 Shroom3 variant, when present in donor kidneys during transplantation, was associated with both increased intrarenal expression of SHROOM3 and increased TIF (CADI-score) by 12-months[8]. Independent work has also supported a regulatory role for intronic Shroom3 variants (rs17219731, rs142647267, rs4859682) in renal epithelial cells[9,46], and more specifically in proximal tubular cells[23]. We also reported that shRNA-mediated Shroom3 knockdown in vivo mitigated TIF in a UUO model and identified crosstalk between TGFβ1- and Wnt/Ctnnb1 pathways mediated by Shroom3[8]. These data supported the concept that targeting Shroom3 in patients, or transplanted donor-kidneys with Shroom3 SNPs, could be an exploitable therapeutic strategy. However, multiple reports have since shown that Shroom3 has potentially beneficial roles in podocytes both during development[12,47], and in adult animals[14], and non-selectively antagonizing Shroom3 protein would incur the unacceptable cost of proteinuria. Our group evaluated the role of Shroom3 in podocyte function in adult animals and reported a previously unidentified Fyn-binding motif in Shroom3. We showed that adult mice with podocyte-specific Shroom3 knockdown phenocopied Fyn-KO mice, demonstrating that this domain played a key role in actin cytoskeletal organization and adult podocyte homeostasis[14].

In the current series of experiments, we tested the hypothesis that a distinct domain, i.e., the ASD2-(Rock-binding) domain in Shroom3, plays the key role in mediating the association of high-expressor Shroom3 SNPs with TIF in CKD. Here, we first screened deletional variants without Shroom3 consensus domains in vitro using TGF-β reporter assays and identified a reduction of TGFβ1/Smad signaling in the ASD2Δ-Sh3 variant. Based on published reports including single cell data suggesting that the roles of Shroom3 and either Rock paralogue during TIF could be localized to injured tubular cells & fibroblasts[8,24–27], we tested ASD2Δ-Sh3- vs WT-Sh3- overexpression in tubular cells as well as fibroblast lines and observed reduced Rock activation, proinflammatory/profibrotic signaling in ASD2Δ-Sh3-fibroblasts (albeit with lower effect sizes in fibroblasts). ROCK1 levels were also increased by WT-Sh3, and reduced by ASD2Δ-Sh3, as the binding of the ASD2-domain of Shroom3 with Rock may improve its subcellular localization[48,49]. Rock1 and Rock2 are regulated by distinct sets of transcription factors during cell proliferation[50,51], but during TIF states may be regulated by TGFB1-signaling[52,53], which also upregulates Shroom3[8,54,55]. In vivo using transgenic mice in two models of TIF, we identify that increased TIF & CKD during global Shroom3 excess are mitigated when the ASD2Δ-Sh3 variant, i.e., with the inability to bind and activate Rock1 or-2, is overexpressed in place of WT-Sh3. Our results are consistent with the prior reported benefits of ROCK inhibitors in experimental TIF[56]. Transcriptomic data confirmed the activation of Rock-, TGFβ1- & Wnt/Ctnnb1- mediated profibrotic signaling in the presence of CKD with excess WT-Sh (vs ASD2Δ-Sh3 variant), and comparison to human CKD transcriptome using co-expression

analyses showed relevance of these findings to human TIF kidneys in CKD. Using tubular-specific overexpression (Pax8-rtTA) mice, we isolated the injured tubular cells as the source where excess Shroom3-Rock interaction during CKD promoted injury, and where antagonizing Shroom3-Rock interaction in high expression states could reduce injury and TIF. Pax8 promoter is pan-tubular, albeit not PT-specific; however, to our knowledge, no PT-specific rtTA is currently available. Critically, while ASD2Δ-Sh3 and WT-Sh3 variants did not induce proteinuria or podocyte injury, FBDM-Sh3 overexpression in mice induced podocyte injury, showing distinct motif-specific effects from SHROOM3 occurring in a cell-type-specific manner. Indeed, in the setting of diabetic kidney injury, global Rock1 deficiency or podocyte-specific Rock2 deficiency led to less albuminuria and a protective phenotype[57,58]. Furthermore, via a distinct 14-3-3 binding motif[59] Shroom3 could regulate Synaptopodin levels in podocytes[14] and facilitate RhoA signaling[60], providing potential explanations for the minimal impact of ASD2-motif antagonism on podocyte phenotypes.

Next, using the crystal structure of ROCK1 (PDB ID: *4L2W*) to model the SBD, and applying rational and fragment-based based drug discovery approaches, we developed seven compounds that could bind to Rock1 and inhibit its interaction with SHROOM3 (target-to-hit). We focused on ROCK1 during design since the crystal structure of ROCK2 (PDB ID: 7JOV) is less well known. However, ROCK1 and ROCK2 have similar SBDs, and our data showed the efficacy of D4G on ROCK2 levels in WT-Sh3 in vitro and in vivo. We screened these initial compounds for efficacy and toxicity in an in vitro setting of WT-Sh3 overexpression, where BT-584 (D4) emerged as our hit compound. D4 was modified using SAR studies, and D4A-G were screened for efficacy (at progressively lower concentrations) and toxicity. BT1137 (D4G) arose as our lead compound, showing the best efficacy and low toxicity. We also observed better specificity for P2I inhibition, unlike HF, which showed significant off-target inhibition against several kinases[41]. Finally, we tested D4G in a UUO model with resultant reduced TIF, consistent with in vitro studies. Similar to the modified transcriptional levels observed in the ASD2Δ-Sh3 UUO kidneys, the P2I treatment also lowered the transcripts of Ccnd1, which is downstream of Wnt-mediated profibrotic signaling[61]. This indicates concordant signals aligning the mechanism of the antifibrotic effect of P2Is, with the transgenic model of deficient Shroom3-Rock interaction.

Our work has important implications for CKD. First, our experimental work ascribes a detailed mechanism to the frequently identified association of highly prevalent intronic cis-eQTL SNPs in Shroom3 (allele frequencies - 0.29–0.4) with CKD in the general population. Next, this mechanism likely has application to both native and allograft CKD occurring in donors with risk alleles[8,62]. Critically, the protein-protein interaction-based, pro-injury signaling mechanism that we describe allowed for the development of P2Is that inhibited Shroom3-Rock interaction and mitigated TIF[36]. It is also important to discuss the

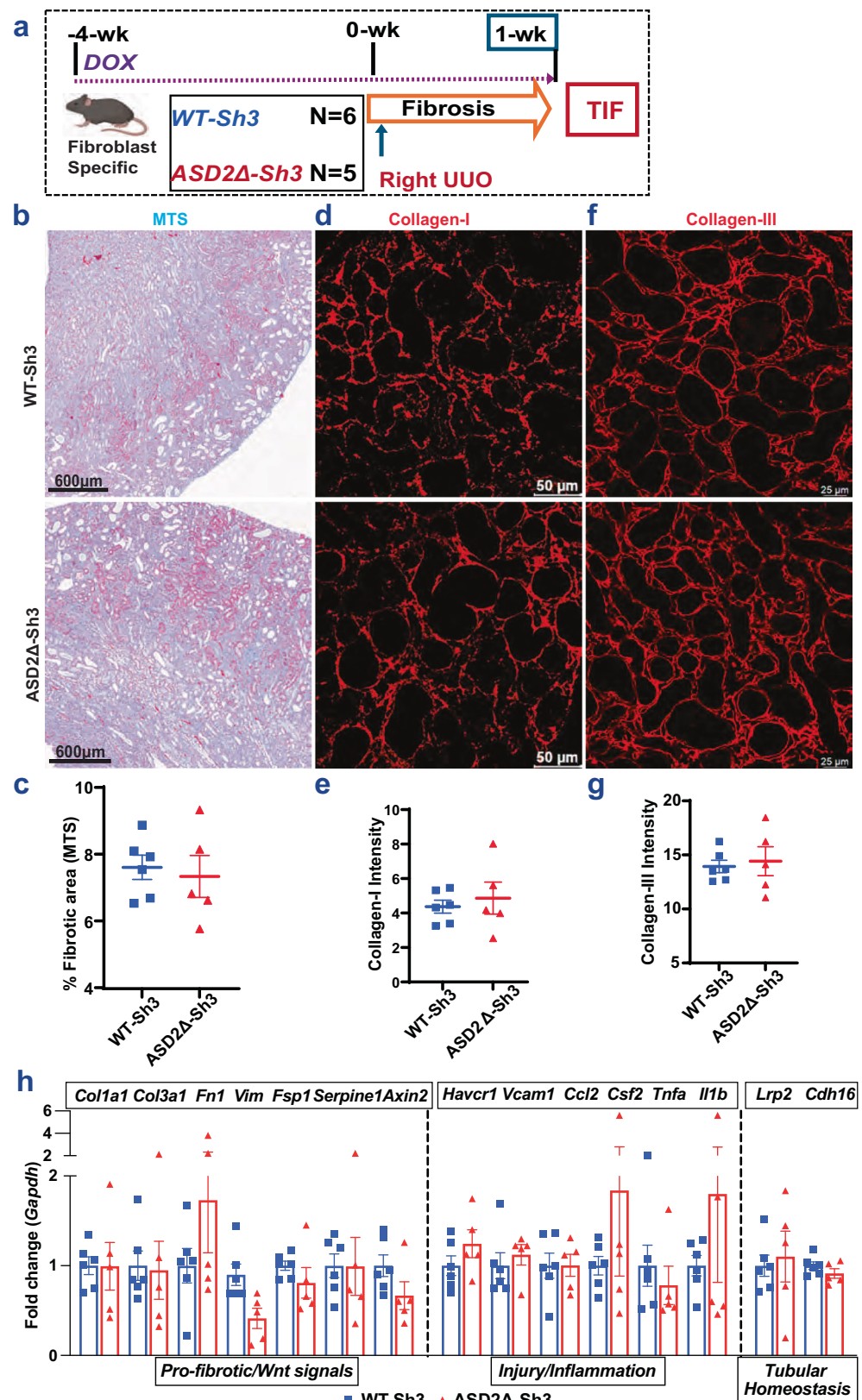

implications of our work for acute kidney injury (AKI). We note that in contra-distinction to multiple reported adverse associations of Shroom3-SNPs with CKD, a potential protective role for a linked intronic Shroom3-SNP in AKI in humans was reported recently[63], and increased AKI also occurred in heterozygous global Shroom3-knockout mice[30]. Facilitation of Rho-kinases by SHROOM3 could aid

in spindle formation and cytokinesis during cell division, and inhibiting activation of ROCKs via ASD2 could reduce PT proliferation after injury[64,65], consistent with our observations in vitro in IMCD lines. Since cell proliferation is required for tubular repair[66], delineating the role of increased Shroom3-Rock interaction during AKI, recovery from AKI vs TIF/CKD is needed to avoid any potential harm during therapeutic use

**Fig. 6 | Fibroblast-specific ASD2Δ-Sh3 mice showing similar TIF to WT-Sh3 overexpression in UUO model. a** Schema of TIF induction by UUO in Pdgfrb-rtTA/WT-Sh3 (*n* = 6) or ASD2Δ-Sh3 (*n* = 5) mice, respectively (Created in BioRender. Caplan, M. (2026) https://BioRender.com/nxzeqir). In all graphs, blue symbols depict WT-Sh3 and red symbols depict ASD2Δ-Sh3 mice. **b** Representative slide wide images (4X) as well as corresponding (**c**) dot-plots quantifying area of blue stain on Masson's Trichrome staining (MTS at 20X magnification) in Pdgfrb-rtTA/WT-Sh3 or -ASD2Δ-Sh3 are shown (≥ 10 fields/animal; *n* = 6 vs 5 mice). **d** Representative fluoromicrograph images (40X), as well as (**e**) dot-plots quantifying intensity of Collagen-I by immunofluorescence (10 hpfs/ animal) are shown

(WT-Sh3 vs. ASD2Δ-Sh3; *n* = 6 vs 5 mice) **f** Representative fluoromicrograph images (40X), as well as (**g**) dot-plots quantifying intensity of Collagen-III by immunofluorescence (10 hpf per animal) are shown (WT-Sh3 vs. ASD2Δ-Sh3; *n* = 6 vs 5 mice). **h** Bar graph representing the relative mRNA expressions of pro-fibrotic/Wnt signaling-, and inflammatory/injury- and tubular homeostasis- genes from UUO kidneys between Pdgfrb-rtTA/WT-Sh3 or ASD2Δ-Sh3 kidneys (*n* = 6 vs 5 mice). [Line and whiskers indicate mean ± SEM. Mann-Whitney tests were used for two-group comparisons and no significant differences were identified in any comparison (i.e. *p* < 0.05)].

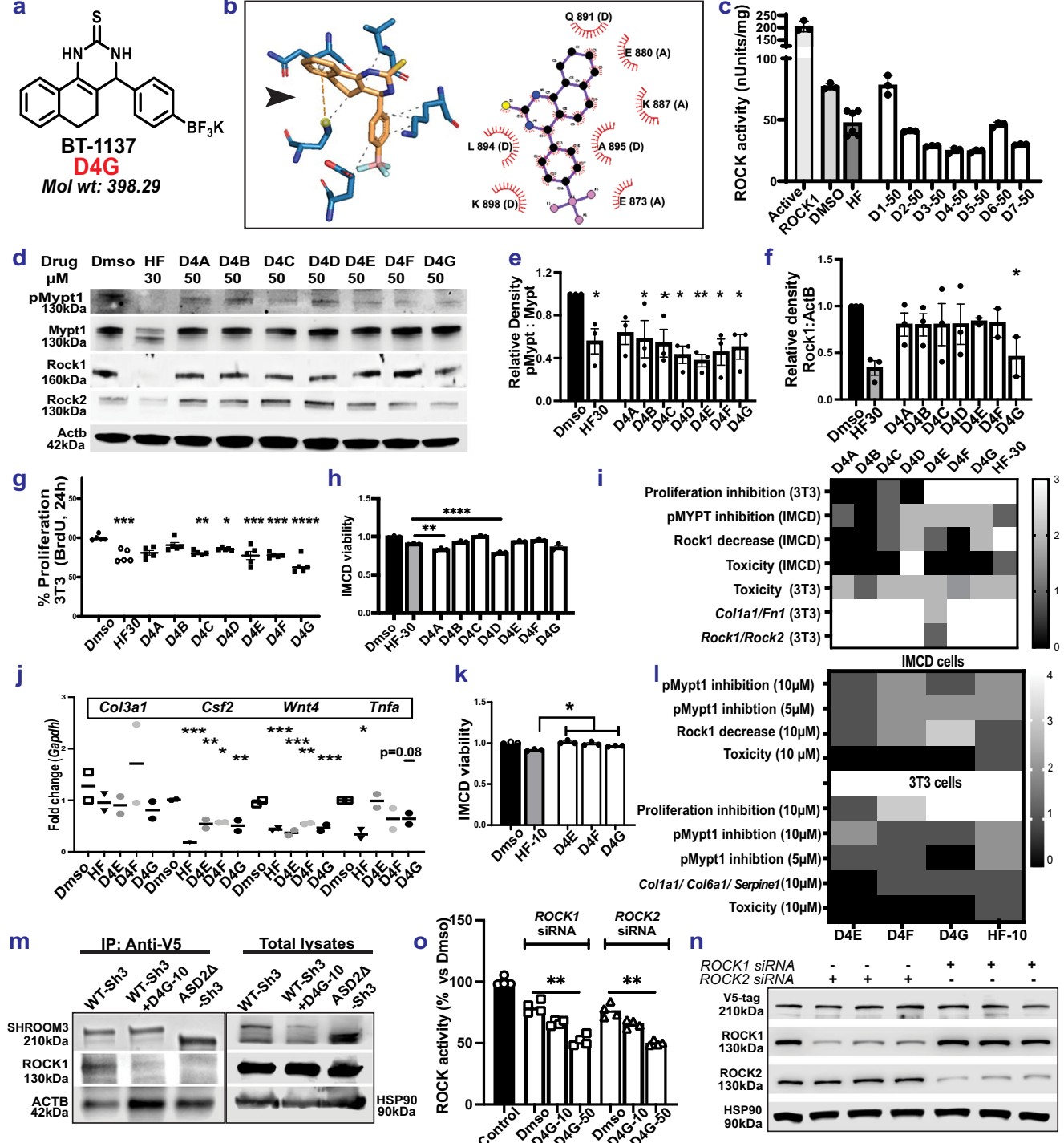

**Fig. 7 | Design and Synthesis of Shroom3-Rock interaction inhibitors to inhibit ROCK activity. a** Structure of compound BT1137 (D4G). **b** Structural representations of the docking of BT1137 with ROCK1 at the SHROOM-binding domain showing amino acid residues at interaction sites. **c** Bar graph representing ROCK inhibition by compounds D1-D7 at 50 μM in HEK293T cell lysates ($n = 3$ technical replicates (6 for HF) vs. DMSO-treated cells (12 h). **d** Representative immunoblots from WT-Sh3-IMCD lysates treated with D4A-D4G (50 μM) or HF (30 μM) probed for pMypt1/Mypt1/Rock1/Rock2/Actb. Corresponding bar graphs quantify (**e**) pMypt1/Mypt1 ($n = 3$ experiments) and (**f**) Rock1:Actb ($n \geq 2$ experiments). Unpaired $T$-test $p$-values comparing each group vs. DMSO are in the source data file. **g** Dot-plot of WT-Sh3-3T3 proliferation upon treatment with 50 μM D4A-D4G (normalized to DMSO-treatment; $n = 5$ experiments; One-way ANOVA with Tukey's post tests are in the source data file). **h** Bar graph depicting the toxicity assay on WT-Sh3-IMCD cells 24 h post treatments with 50 μM D4A-D4G ($n = 3$ experiments; One-way ANOVA with Dunnett's tests D4A vs HF ($p = 0.006$), D4D vs HF ($p < 0.0001$)). **i** Heatmap summarizing in vitro effects of D4A-D4G at 50 μM on WT-

Sh3-IMCD/3T3 cells. **j** Dot-plot depicting the relative mRNA expressions of pro-fibrotic/pro-inflammatory markers plot ($n = 2$ experiments; One-way ANOVA & Dunnett's tests vs. Dmso shown in source data file). **k** Bar graph shows cell viability proportions in WT-Sh3-IMCD cells treated with 10 μM D4E-D4G ($n = 3$ experiments; One-way ANOVA & Dunnett's tests ($p = 0.0003, 0.001, 0.02$, respectively for D4E, D4F, D4G $vs$ HF) (**l**) Heatmap summarizing the in vitro effects of D4E-D4G at 5–10 μM on WT-Sh3-IMCD/3T3 cells. **m** Representative immunoblots from lysates of WT-Sh3-HEK293T with/without D4G, or ASD2Δ-Sh3-HEK293T, and corresponding total cell lysates probed for SHROOM3, ROCK1 and ACTB or HSP90 ($n = 2$ sets). **n** Representative immunoblots from lysates of WT-Sh3-HEK293T probed for SHROOM3, ROCK1, ROCK2 and HSP90, +/- Rock1- or Rock2- silencing ($n = 3$ sets each). **o** Bar graphs represent ROCK-activity in Rock1 or Rock2 silenced WT-Sh3-HEK293T lysates (Friedman's ANOVA with Dunn's tests $p$-value 0.0094 (D4G–50 vs DMSO with either $Rock1$ or $Rock2$ silencing). [Line and whiskers indicate mean ± SEM; Two-tailed $p$-values denoted as *$p < 0.05$, **$p < 0.01$, ***$p < 0.001$, ****$p < 0.0001$].

of P2Is. We acknowledge that we did not conduct such time course experiments. However, we note that ASD2Δ-Sh3 was continuously expressed in our models throughout the AKI phase, and we still observed a benefit in TIF. Next, in spite of this potential adverse effect of inhibiting ROCKs during AKI, Rock-inhibitors have consistently shown benefit in models of AKI[64,67]. A possible explanation is the observation that inhibition of Shroom3-Rock interaction in tubular cells alone markedly inhibited *Ccl2*, a chemokine signal known to be facilitated by rho-kinase activation[31,68,69]. Ccl2 mediates immune cell recruitment via its receptor Ccr2 and is a potent facilitator of tubular-inflammatory cell crosstalk after injury[27]. Hence, in spite of potential deleterious effects on tubular cell proliferation during AKI, excess Shroom3-Rock interaction in iPTs overexpressing *SHROOM3* would enhance crosstalk between iPTs and interstitial-inflammatory and/or stromal -cells, culminating in increased "fibro-inflammation" and progressive CKD. Our findings also support single-cell functional genomics studies that identify iPTs as sites of CKD heritability via SNPs[23], and are consistent with studies identifying signaling from iPTs as initiators of CKD progression[70,71].

Our data has some limitations. First, while in vitro in 3T3 cells, we observed a benefit with P2Is; in vivo, fibroblast-specific ASD2Δ-Sh3 was not protective for TIF. This may be because levels of endogenous expression of Shroom3 may attenuate the dominant negative effect of ASD2Δ-Sh3 in cell types such as fibroblasts and podocytes with high constitutive Shroom3 levels. While Rock1- & 2 show near ~90% homology, these have non-redundant cell-specific and temporal expression in kidney cells during injury[72], whether inhibiting one or both kinases is essential for the anti-fibrotic effect still needs to be defined. From the multiple single-cell expression datasets[24–26], it is apparent to us that cell-type-specific expression patterns of each Rock paralog and Shroom3 are variable and sometimes mismatched between datasets. This variability may have resulted from technical variability between datasets (single cell vs single nuclear, etc). To minimize the impact of this in our work, we mined multiple datasets, which included multiple injury models and modalities of single-cell transcriptome assessment to provide a cell-type rationale for our mechanistic studies. Finally, although the beneficial role of inhibiting Shroom3-Rock interaction on TIF was not model-specific, and we demonstrated that D4G has in vivo efficacy as a P2I in a UUO model of TIF, evaluation of P2Is in other in vivo models, including pharmacokinetics/pharmacodynamic profiles, human organoid testing and longer-term toxicity studies, is essential. All these studies are critical for translational potential and the ultimate development of refined Shroom3-Rock P2Is to treat CKD related to Shroom3-excess & Shroom3 SNPs.

In summary, our data reports a critical role for Shroom3-Rock interaction in injured tubular cells as a mechanism underlying increased TIF in situations of Shroom3 excess. The absence of the

ASD2 domain reduces ROCK activation and subsequent pro-fibrotic-, pro-inflammatory signals, and ECM deposition to ameliorate TIF progression. Our work unravels a plausible and targetable pathway that is specifically applicable to patients with progressive kidney disease who harbor the intronic enhancer Shroom3 SNPs that associate with CKD in GWAS and could have greater applicability.

## Methods
The research complies with the ethical regulations for biomedical research enforced by Yale University. All animal protocols were approved by the Yale University Animal Care and Use Committee (IACUC protocol number- 20363).

### Cell culture
HEK293T (CRL-3216) cell line was obtained from Dr Cijiang John He (Icahn School of Medicine, Mount Sinai), mIMCD-3 (CRL-2123) cell line, and NIH/3T3 (CRL-1658) cell line were gifted by Dr Stefan Somlo (Yale) and Dr Lloyd Cantley (Yale), respectively. All cell lines were originally obtained from ATCC. The mIMCD-3 cells and the HEK293T/NIH/3T3 cells were grown in DMEM/F-12 and DMEM, respectively, supplemented with 10% FBS and 1% penicillin-streptomycin and maintained at 37 °C in a humidified atmosphere containing 5% CO2.

### Transient overexpression studies
Human SHROOM3 mutant constructs (GenScript) with deletions of PDZ, ASD1 or ASD2 domains, as well as WT, were transfected onto HEK-293T cells using PolyJet transfection (SignaGen Laboratories, Rockville, MD)[8]. All constructs used had a V5-tag, and the plasmids (5 μg) were mixed in culture media (500 μL) and added to a microfuge tube containing 5 μL transfection reagent mixed in culture media (500 μL). This mixture was vortexed and incubated for 10 min at room temperature. Then the mixture was added to culture dishes plated with the target cells and incubated for 48 h for the transfection to complete.

### Luciferase assay
HEK-293T cells were co-transfected with SBE4-Luc (0.5 μg) and pRL plasmids (0.2 μg) as 5 μg of either of the SHROOM3 mutant constructs, using the PolyJet Transfection Kit according to the manufacturer's instructions (SignaGen) as described above. The transfected cells were treated with 5 ng/mL TGFβ1 or vehicle for 15 min, and Luciferase activities were measured using the Dual-Luciferase Reporter Assay Kit (E1910; Promega).

### Generation of stably transfected cell lines
HEK-293T cells were used as producer cells for lentiviruses with 2nd generation lentiviral system, using one transfer plasmid, one packaging plasmid (psPAX2, Addgene: #12260) and one envelope plasmid (pVSVG, Addgene: #8454). The cells were transfected with

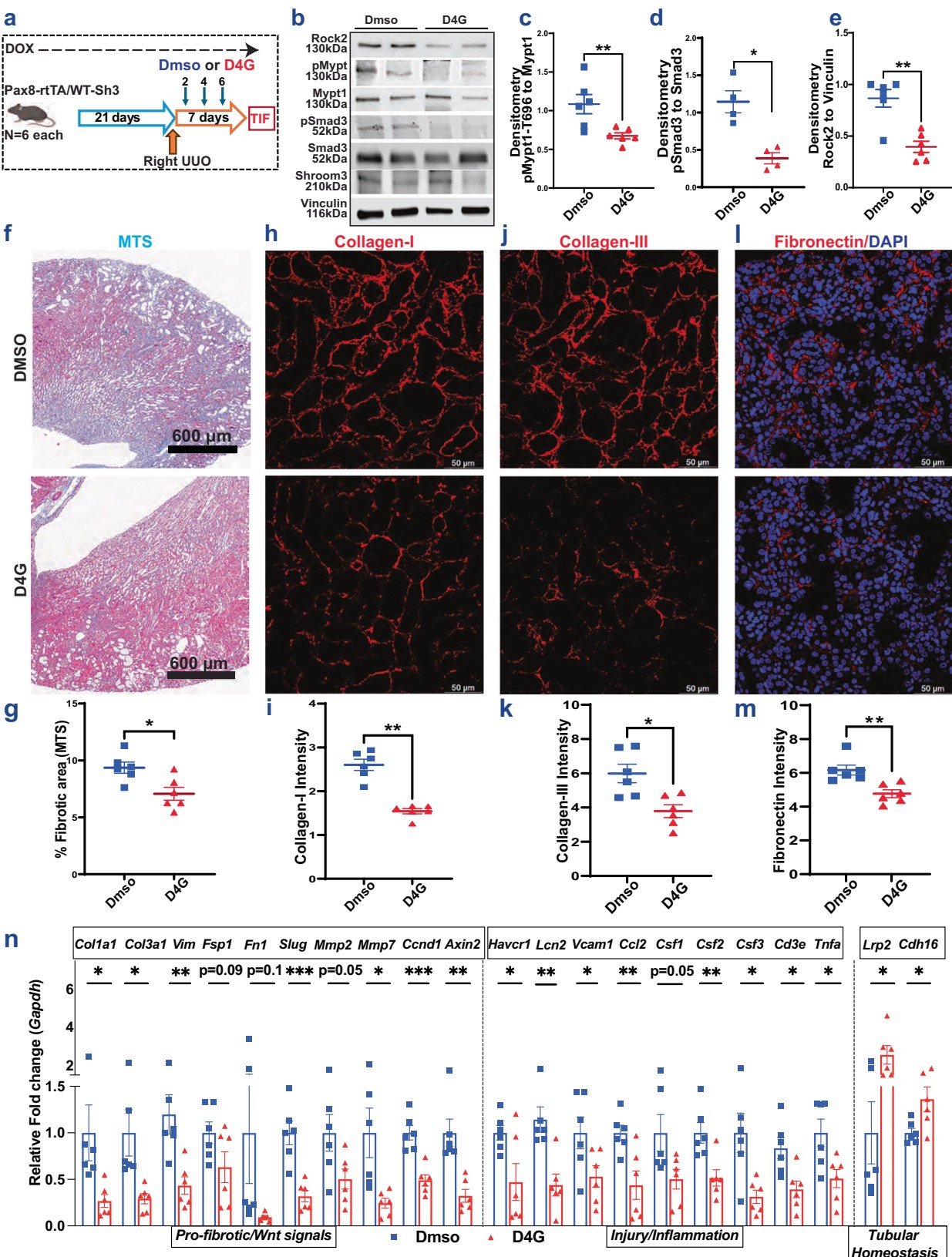

5 μg each of the GFP, WT- or ASD2Δ- Shroom3 lentiviral plasmid (pLenti CMV/TO Puro DEST; GenScript), packaging plasmid (pPACK), and envelope plasmid (pVSV) using PolyJet transfection reagent (#SL100688, SignaGen Labs) to generate mammalian VSV-pseudotyped lentiviral expression constructs[8]. The media containing lentiviral constructs were collected, filtered and concentrated

using Lenti-X Concentrator (#631231; Takara) following the manufacturer's protocol. These lentiviral supernatants were used to infect IMCD and 3T3 cells at a 1:50 ratio with culture media supplemented with 8 μg/mL polybrene and incubated in a $CO_2$ incubator for 48 h. The stably overexpressing cells were selected by puromycin (#ant-pr-1, InvivoGen) treatment (1 μg/mL) for 1–2 weeks before using

**Fig. 8 | D4G ameliorated TIF progression in Tubular-Specific WT-Sh3 overexpression mice. a** Schema of TIF induction by UUO in Pax8-rtTA/WT-Sh3 mice followed by treatment with drug D4G (*n* = 6) or vehicle (5% DMSO in Corn oil; *n* = 6; created in BioRender. Caplan, M. (2026) https://BioRender.com/nxzeqir). In all graphs, blue and red symbols represent DMSO and D4G, respectively. **b** Representative immunoblots from 2 mice in each group showing Rock2, pMypt1/Mypt1, pSmad3/Smad3 and Vinculin on UUO-kidney lysates from DMSO and D4G treated mice, and (**c–e**) dot-plots show corresponding densitometric quantification comparing DMSO and D4G treatments for (**c**) pMypt1:Mypt1 (*n* = 6 each; Mann-Whitney *p* = 0.009), (**d**) pSmad3:Smad3 (*n* = 4 each; Mann-Whitney *p* = 0.028), & (**e**) Rock2:Actb (*n* = 6 each; Mann-Whitney *p* = 0.009). **f** Representative slide wide images for MTS (4X) as well as (**g**) dot-plots quantifying area of blue stain on MTS in DMSO or D4G-treatment are shown (>10 fields/animal, *n* = 6 mice/group; Mann-Whitney *p* = 0.015). **h** Representative fluoromicrograph images (40X), as well as (**i**)

dot-plots quantifying intensity of Collagen-I by immunofluorescence (10 hpf per animal, *n* = 6 mice/group; Mann-Whitney *p* = 0.002) in Pax8-rtTA/WT-Sh3 treated with DMSO or D4G limbs are shown. **j** Representative fluoromicrograph images (40X), as well as (**k**) dot-plots quantifying intensity of Collagen-III by immunofluorescence from DMSO or D4G-treatment mice are shown (10 hpf per animal, *n* = 6 mice/group; Mann-Whitney *p* = 0.026). **l** Representative fluoromicrographs (40X), as well as (**m**) dot-plots quantifying intensity of Fibronectin by immunofluorescence from DMSO or D4G-treatment mice are shown (8–10 hpfs/animal, *n* = 6 mice/group; Mann-Whitney *p* = 0.002). **n** Bar graph representing the relative mRNA expressions of pro-fibrotic/Wnt signaling-, and inflammatory/injury- and tubular homeostasis- genes from UUO kidneys (Exact *p*-values are given in the Source data file). [hpf = high power field; MTS = Masson's Trichrome Stain; Line and whiskers indicate mean ± SEM; Two-tailed *p*-values denoted as \**p* < 0.05, \*\**p* < 0.01].

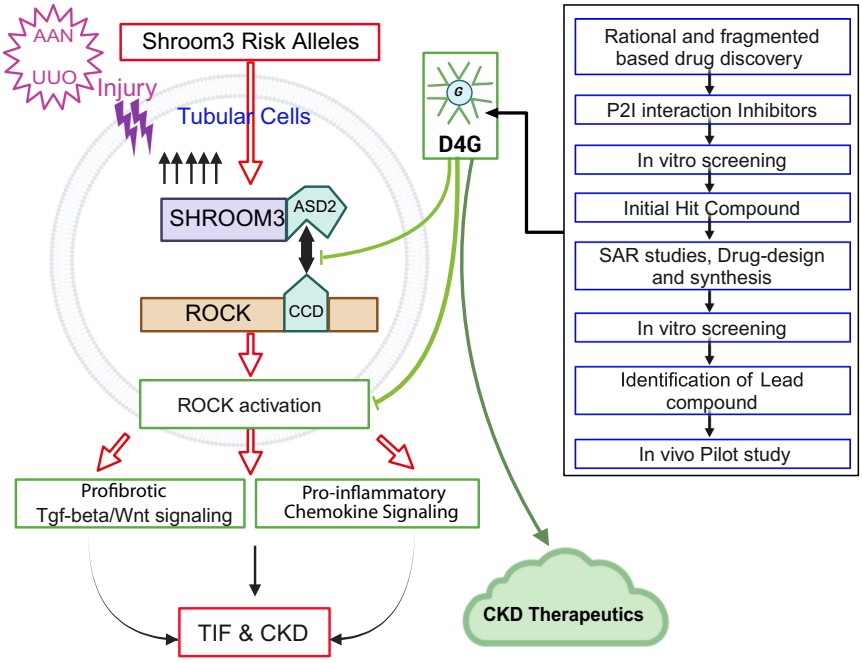

**Fig. 9 | Summary figure showing efficacy of a P2I (D4G) to target this protein-protein interaction.** WT-Sh3 overexpressing mice were generated to mimic Shroom3 excess in the kidney cells of humans with CKD-associated, enhancer, Shroom3 SNPs. During injury (AAN or UUO), excess SHROOM3 facilitates interaction with Rho-kinases in iPTs. In this milieu, augmented ROCK1, −2 activation within iPTs leads to increased TIF after injury via increased Wnt/Ctnnb1- and Tgfβ1- profibrotic signals as well as pro-inflammatory signals, promoting "fibro-inflammation", progressive TIF after injury and CKD. This likely explains the mechanism

underlying the risk of CKD with Shroom3 SNPs and illustrates the key cell type involved. In support of this, targeting Shroom3-Rock interaction (between the ASD2 domain of Shroom3 and the Coiled coil domain [CCD] of Rock, which houses the Shroom binding domain) by genetic or pharmacologic means (P2Is) during tubular Shroom3 excess rescues the increased TIF, reflecting the therapeutic potential of this strategy for TIF and CKD progression in at-risk patients. (iPTs -injured proximal tubular cells). Created in BioRender. Caplan, M. (2026) https://BioRender.com/6ykyx16.

---

them for experiments. The overexpression was confirmed by qPCR and western blot.

## Western blot

Mouse kidney tissues, cell lines and primary renal cells were lysed using RIPA buffer added with Halt Protease and Phosphatase Inhibitor (78440, Thermo Scientific) as mentioned elsewhere (24300173). Briefly, the tissue or cells were homogenized in the lysis buffer using mechanical digestion (Precellys homogenizer, Bertin technologies) and incubated on a rotator at 4 °C for 30 min. The lysate was centrifuged at 12000 x g at 4 °C for pelleting down the undigested cell debris. The protein concentrations of the supernatant lysates were determined by BCA assay (23225, Pierce-Thermofisher). Loading buffer was then added, and the mixtures were denatured at 95 °C. The proteins were separated on 4–15% or 10% gels (Bio-Rad) by SDS-PAGE and transferred to an Immobilon-P membrane (Millipore), followed by blocking with 5% BSA or 5% skim milk. The primary antibodies used

were: anti-MYPT (D6C1) (1:2000; Rabbit mAb #8574, Cell Signaling), anti-phosphoMYPT-T696 (polyclonal) (1:1000; #5163, Cell Signaling), anti-ROCK1 (C8F7)(1:2000; 4035S, Cell Signaling), anti-ROCK2 (D1B1) (1:2000; #9029, Cell Signaling), anti-HSP90 (C45G5) (1:2000; #4877, Cell Signaling), anti-SHROOM3 (polyclonal) (1:500; #SAB3500818, MilliporeSigma), anti-SHROOM3 (polyclonal) (1:500; #LS-C679459, LSBio), anti-V5(1:1000; #V8012, MilliporeSigma), anti-FLAG (1:500; #A8592, MilliporeSigma), anti-Gapdh (6C5)(1:2000; #AM4300, Invitrogen), anti-β-Actin (AC-15) (1:2000; #A5441, MilliporeSigma), anti-phospho-SMAD3- Ser423/425 (C25A9) (1:1000; #9520, Cell Signaling), anti-SMAD3 (C67H9) (1:1000; #9523, Cell Signaling), anti-Vinculin (7F9) (1:5000; #sc-73614, SCBT). The secondary antibodies used for detection included horseradish peroxidase (HRP)-conjugated anti-rabbit (#AP307P, MilliporeSigma) and anti-mouse (#AP308P, MilliporeSigma) polyclonal antibodies, with a dilution ranging from 1:8000 to 1:12000. Licor Odyssey Fc imager (LicorBio, Lincoln, NE) with Image Studio Lite Ver5.2 was used for western blot image acquisition.

Densitometry was performed on images of Western blots using ImageJ software.

## Immunoprecipitation

PCDEST SHROOM3 (PDZΔ, ASD1Δ, ASD2Δ and WT) and GFP (control vector) overexpressing HEK-293T cells were generated in 15 cm culture dishes. Protein lysates post 36-h of transfection were immunoprecipitated with Anti-V5-tag mAb-Magnetic Beads (MBL #M167-11) and run on PAGE gels as described elsewhere[73], and immunoblotted for anti-Shroom3 (polyclonal)(1:500; #SAB3500818, MilliporeSigma), anti-Rock1 and anti-Rock2 (1:2000; #4035 (C8F7); 1:2000; #9029 (D1B1) Cell Signaling). For testing the lead compound D4G for Shroom3-Rock interaction inhibition, similarly, HEK-293T cells were transfected with WT-Sh3- or ASD2Δ-Sh3 constructs. Either DMSO or D4G at 50 mM was added after 12 h to plates with WT-Sh3-IMCD. Protein lysates were collected at 36 h post-transfection, run on PAGE gels, and immunoblotted as above.

## HEK-293T SMAD reporter (blue) cell studies

PCDEST SHROOM3 (PDZΔ, ASD1Δ, ASD2Δ and WT) and GFP (control vector) overexpressing Smad reporter HEK-Blue™ cells (InvivoGen) were generated in 6-well plates by transient transfection (*n* = 4 replicates each). After 24 h, cells were serum starved overnight, and either DMSO or TGFβ1 (5 ng/mL). Supernatant was removed and assayed for secreted alkaline phosphatase activity (SEAP) using Quantiblue solution (per manufacturer). Cells were lysed, and lysates immunoblotted for assaying level of transfection.

## Immunocytochemistry

WT-Sh3 and ASD2Δ-Sh3 overexpressing IMCD cells were plated in 12-cm wells on cover slips, serum-deprived for 8 h and treated with DMSO or Fasudil (#S1573, Selleckchem) for 24 h. The treated cells were fixed and permeabilized (36% HCHO, 0.1% Triton X-100 in PBS). The cells were blocked with 3% BSA/10% normal goat serum, followed by incubation with primary antibody (1:100) against anti-ROCK1 (polyclonal) (PA5-22262; Thermo Scientific). F-Actin staining for observing cytoskeletal changes in SHROOM3-transfected cells was done using 100 nM Acti-stain 488 phalloidin (#PHDG1, Cytoskeleton Inc.) along with anti-rabbit Alexafluor−568 secondary antibody (1:300; #A-11036; Invitrogen) in 1% BSA-PBS. The fluoromicrographs were analyzed using ImageJ v2.1.0 (NIH, MD),

## Cell proliferation assays

Cell lines overexpressing the SHROOM3 variants were plated on to 96 well plates and BrdU incorporation was performed according to the manufacturer's protocol (#QIA58, MilliporeSigma). For drug screening experiments, different concentrations of drugs were added to the cells and incubated for 24 h. The BrdU reagent was added to each well 12 h before the completion of incubation. The colorimetric measurements corresponding to the BrdU incorporation were done at 450 nm using BioTek Synergy Multimode Reader (Agilent Technologies Inc., Santa Clara, CA).

## Cell viability/toxicity assays

Cell lines overexpressing the intact SHROOM3 were plated at a density of $5 \times 10^5$ cells/ml in 96-well white opaque plates (Corning). After 24 h of treatments with Fasudil or drugs, 100 µL of CellTiter-Glo 2.0 (Promega) reagent was added to each well, mixed for 2 min on an orbital shaker and further incubated for 10 min. The luminescence measurements were done using a BioTek Synergy Multimode Reader.

## Primary cultured renal cells

The primary renal cells were isolated from mouse kidneys and cultured for 5 days[27]. Briefly, kidneys were harvested, minced, and digested in Liberase enzyme (0.5 mg/mL) with DNase I (100 mg/mL) (#5401151001,

#10104159001, Roche Diagnostics) and 0.5 mM $MgCl_2$ for 30 min. The resulting cell suspension was filtered, washed, and incubated with RBC lysis buffer (#00-4333-57, Invitrogen) for 3 min, followed by washing with cold HBSS. The resultant cell pellet was resuspended in DMEM with 10% FBS and 1% penicillin-streptomycin and seeded into culture dishes. After 5 days of expansion and media changes, primary renal cell cultures (P0 PCRCs), which are 75% proximal tubules, were harvested and cryopreserved.

## Scratch wound assay

PCRCs or IMCD cells expressing Shroom3 constructs were seeded in a 12-well culture plate at a density of approximately 70–80% confluence and incubated overnight at 37 °C in a 5% CO₂ atmosphere. A sterile 10 µl pipette tip was used to create perpendicular-line scratches across the cell layer, ensuring uniform width and depth. The plates were gently washed with sterile PBS to remove non-adherent cells. Fresh culture medium without FBS was added to each well, with or without experimental treatments. Images of the scratch area were captured at time zero (T0) using a microscope. Plates were then incubated as before. At 16 h (IMCD) and 24 h (PCRC), images were taken of the scratch area to observe cell migration. The scratch areas were analyzed using ImageJ software to determine the percentage of wound closure over time.

## Murine inducible global WT-Shroom3, ASD2Δ-Shroom3 and FBDM-Shroom3 overexpression models

Plasmid constructs of WT, ASD2Δ and FBDM Shroom3 (under TRE) tagged with FLAG were co-transfected with a plasmid construct with rtTA under a CAGS promoter (MGI:6451722) to generate inducible expression models. The resultant 3 fused plasmids were purified and used for transfecting mouse embryonic stem cells, followed by the generation of founder mice lines by the mouse core facility. For each construct, 5–7 founder lines were generated harboring random insertions in the genome in the FVB/NJ (Strain:#:001800) background. The mice from each founder line were tested for optimal (2–3 fold) and inducible overexpression in kidney- and tail- tissue post DOX induction. Founder mice of either gender (6–8 weeks old) were provided DOX chow (#TD.08541; Inotiv [Envigo]) and DOX (#D3447; Millipore Sigma) water (200 mg/L) for 4 weeks and tested for mRNA by RT-qPCR and protein overexpression by Western blot. One founder line with the optimal overexpression for each of the Shroom3 mutants was used for in vivo experiments.

*Tubular specific WT-Shroom3 and ASD2Δ-Shroom3 overexpression models:* The WT-Sh3 and ASD2Δ-Sh3 mice (CAGS-rtTA null) were each crossed to mice expressing Pax8-rtTA (Strain #007176; C57BL/6) for tubular-specific overexpression under DOX induction[8].

*Fibroblast specific WT-Shroom3 and ASD2Δ-Shroom3 overexpression models:* The WT-Sh3 and ASD2Δ-Sh3 mice (CAGS-rtTA null) were each crossed to mice expressing Pdgfrb-rtTA (Strain #:028570; C57BL/6) for fibroblast-specific overexpression under DOX induction[74].

All mice were maintained in individually ventilated cages (IVC) on a 12-h light and 12-h dark cycle at a temperature between 68–79 °F and humidity between 30–70% with free access to standard chow until used for experiments.

## Injury models

All injury models were initiated in 10–12 week-old mice post-induction of overexpression by DOX. In the Aristolochic acid nephropathy (AAN) model, all mice received intraperitoneal injections (3 mg/kg) of aristolochic acid (AAI, #A9451; MilliporeSigma) every third day for 3 weeks for a total of 8 injections and were euthanized 6 weeks after the last injection[75,76]. In the Unilateral ureteral obstruction (UUO) model[77], mice were anesthetized with ketamine/xylazine and maintained on a 37 °C heat pad during surgery. Both genders were used for UUO

experiments, but only male mice were used for the AAN model owing to the resistance in female mice to AAN. The right kidney was exposed via a flank incision, the ureter was ligated, and the mice were euthanized after 1 week. The kidney tissues were collected for histology, immunostaining for fibrosis markers, RNA isolation for RT-qPCR, and protein extraction for Westerns. Mouse cartoon images in Figs. 3–6, 8 depicting the schema of injury models were created in BioRender. Caplan, M. (2026) https://BioRender.com/nxzeqir.

## Reverse transcription qPCR

Intron-spanning primer sets were designed for all assayed genes using Primer-BLAST (NCBI), and PCR amplicons were confirmed by both melting curve analysis and agarose gel electrophoresis and obtained from Integrated DNA Technologies (Coralville, IA). *Gene* expression was assayed by qPCR using Applied Biosystems 7500 Real-Time PCR System (ThermoScientific, Waltham, MA). Amplification curves were analyzed using the automated StepOne software v2.3 via the ΔΔCt method. *Gapdh* was used as an endogenous control. See Supplementary dataset S1c for primer sequences.

## Bulk RNA-seq analyses

DNase treated RNA extracted from AAN-kidneys of WT-Sh3 and ASD2Δ-Sh3 ($n = 4$) were depleted of rRNA and underwent quality control (QC) analysis at the Yale Center for Genome Analysis (YCGA) followed by library prep and Poly(A) RNA sequencing (NovaSeq, Illumina). The raw Illumina RNAseq paired-end data were quality controlled and adapter-trimmed using FastQC version 0.12.1[78,79] and BBDuk version 39.11[80,81]. We mapped the RNAseq data to the reference genome using STAR version 2.7.11a[82]. For read alignment, we used both the mouse reference genome (GRCm39) and the human reference (GRCh38). By mapping to human reference, we obtained the gene expression profiles in human orthologs. We computed the read counts in gene sets from mapped data using the Subread R package, FeatureCounts (version 2.0.0)[83]. To calculate differential gene expressions, we used DESeq2 version 1.44.0[84]. The principal component analysis (PCA) was computed using the DESeq2 plotPCA function.

## Immunohistochemistry/immunofluorescence

For immunostaining, 5 µm sections of formalin-fixed paraffin-embedded kidney tissues (processed at Yale Pathology Tissue Services (YPTS) facility) were deparaffinized and subjected to antigen retrieval (Retrievagen (pH 6), BD Biosciences), to unmask the antigens, followed by incubation with overnight primary antibodies: anti-SHROOM3 (polyclonal) (1:500; #LS-C679459, LSBio), anti-Collagen-I (polyclonal) (1:200; #1310-01) and anti-Collagen-III polyclonal)(1:200; 1330-01) from Southern Biotech; anti-FLAG (D6W5B)(1:100; #14793, Cell Signaling), anti-Fibronectin (polyclonal)(1:100; #SAB5700724, MilliporeSigma); rabbit polyclonal antibody against mouse Megalin (1:1000; anti-MC220 (PMID: 15180987)) and mouse monoclonal antibody against KSP-Cadherin (clone:4F6/F6) (1:1000) kindly provided by Dr Robert Brent Thomson at the Yale Nephrology. Anti-rabbit Alexafluor-568 secondary antibody (polyclonal)(1:300; #A-11036; Invitrogen) and Alexafluor-594 rabbit anti-Goat (polyclonal)(1:300; #A-11080; Invitrogen) secondary antibodies in 0.1% BSA (in PBS) were used. FITC labeled Lotus Tetragonolobus Lectin (1:100; #L32480, Invitrogen) was used to stain the brush border of proximal tubules for co-localization with Shroom3. DAPI (1 µg/mL; #D9542; MilliporeSigma) was used to stain nuclei. Images were captured using STELLARIS Confocal Microscope (Leica Microsystems, IL) using Leica LASX v1.44. Fluorescence intensities or percent positive area were measured from 8–10 high-power fields (hpf) per slide using ImageJ v2.1.0 (NIH, MD).

## Fibrosis score determination

For visualizing the collagen fibers, 5 µm sections of formalin-fixed kidney tissues were deparaffinized and stained with Masson's Trichrome stain

(MTS) and/or picrosirius red stain (both stained at YPTS facility). The slides were scanned using a digital slide scanner (Motic Digital Pathology) and the images captured using Aperio ImageScope v12.4.6.5003 (Leica Biosystems). The blue area (MTS) or the plane polarized light Sirius red-stained area was assessed from 10 hpf using ImageJ v2.1.0.

## Electron microscopy

Glutaraldehyde-fixed samples were processed, embedded in epoxy resin and underwent ultramicrotomy[85]. The electron micrographs were captured at the Electron Microscopy & Cryo Electron Microscopy facility in the Center for Cellular and Molecular Imaging (CCMI), Yale. The average foot process width was determined as described elsewhere[86]. In brief, for all animals, three to five toluidine blue semithin sections were reviewed, and those containing more than five glomeruli were selected for ultrathin sectioning on grids. These grids were stained with uranyl acetate and lead citrate using the standard protocol employed by the central CCMI facility at Yale. Images of free glomerular capillary loops were captured at 1200× to 4800× (direct magnification), and micrographs from each group were analyzed to determine the mean foot process width (FPW) using ImageJ v2.1.0.

## Development of P2I inhibitors

The crystal structure of SHROOM3-ROCK interaction is not well known, while the interaction sequences in the SBD – i.e., ROCK1 (AA 836- 913), ROCK2 (AA 856–936) with SHROOM3 ASD2 domain (AA 1563- 1986), are known[36]. The interaction is conserved cross-species[41,42]. The crystal structure of the SBD of human Rock1 (PDB ID: *4L2W*) was first prepared, and possible binding cavities were identified for ligand interaction. Using a rational and fragment-based drug discovery (FBDD) approach, taking Pdb ID: 4L2W (Shroom-Binding domain of human Rock1) into consideration, we synthesized initial compounds and performed screening to identify the hit compound. We were informed by a prior Shroom3-Rock2 interaction inhibitor that had been screened in neuronal cells in vitro[36]. Virtual screening was carried out to dock each of the P2Is, selecting the lowest energy poses for the most favorable interactions between the ligands and the protein. Using this, we developed 7 P2Is for initial screening (D1–D7). Our analysis revealed that primary interactions between P2Is and the protein were hydrophobic and π–π interactions [Fig. 7b]. D1-to-D7 were screened using ROCK assay (below) and pMypt levels, to identify our hit compound BT584 (D4). D4 was active in the 50 micromolar range. SAR studies were then performed for D4 (as shown in Fig. S7c) to develop and synthesize seven more compounds, D4A-D4G. D4A-D4G were screened in two cell types of WT-Sh3 overexpression–IMCD (tubular cells) and 3T3 (fibroblasts) for toxicity and efficacy studies (shown in Figs. 7, S7). Each lot of drugs synthesized for in vitro and in vivo studies was confirmed using their nuclear magnetic resonance spectra. Using our lead compound BT1137 (D4G) as an example, a sample fluoro-NMR spectra from BT1137 (confirming 3 Hydrogen, 13 Carbon, and 3 Fluorine moieties), and representative drug synthesis steps are included in Supplementary dataset S2.

## Rock, Rock2 knockdown studies

Small interfering RNAs (siRNA) (from Horizon Discovery Bioscience) were used to silence either Rock1(#L-003536-00-0005) or Rock2 (#L-004610-00-0005) in WT-SH3-HEK293T cells, and the effect of the drug D4G on residual Rock activity was assayed. Transfections of 20 pmol of the siRNAs were done using Lipofectamine RNAiMax (Invitrogen, #13778150) following the manufacturer's instructions and the knockdown efficiency was assessed by Western blot. Drug treatments were done 60 h post-transfection, and cells were harvested at 72 h for assays. Results were compared with cells transfected with ON-TARGETplus Non-targeting Pool (Horizon Discovery, # D-001810-10-05).

## Rock activity assay

For the primary screening of the first set of drugs, HEK-293T cells were treated with HF and the compounds for 12 h and lysed with the cell lysis buffer available with the ROCK activity assay kit (#CY-1160, MBL Life Science). For screening the derivatives of the drug D4, IMCD cells overexpressing WT-Sh3 were treated as above and lysed after 12 h treatments. For evaluating the efficacy of the lead compound D4G on Rock1/2 separately, Rock1-knockdown and Rock2-knockdown WT-Sh3-HEK293T cells were treated with D4G during the last 12 h of transfection and were harvested for cell lysate preparation. The assay was performed with the cell lysates (prepared using 2x Cell Lysis Buffer, #RABLYSIS1; Millipore Sigma) as per kit instructions.

## In vitro drug screening

IMCD cells were treated with D1–D7 at 50 μM and 100 μM for 6 h and assayed for pMypt levels for initial screening. After developing the derivatives of D4, WT-Sh3 overexpressing IMCD and 3T3 cells were treated with 50 μM of D4A-D4G for protein analysis- IMCD (6 h), qPCR analysis(12 h) and toxicity and proliferation assays (24 h). On identifying the potential drug candidates, D4E-D4G were tested at 1–10 μM concentrations for protein analysis, qPCR analysis and toxicity/proliferation assays as described above.

## In vivo drug testing (D4G)

Adult Pax8-rtTA/WT-Sh3 mice were DOX fed to induce tubular-specific transgene induction. After 4 weeks, the right UUO was performed (as above). Starting at day 1 after UUO on day-0, either D4G (50 mg/kg/dose) or DMSO (5%)(#D2653; Millipore Sigma) suspended in corn oil (#C8267; Millipore Sigma) was injected intraperitoneally every 48 h till sacrifice on day-7. Serial weights were monitored. Urine and blood were collected at sacrifice, and kidney tissue was harvested, processed and stored for RNA, protein, and histology evaluation.

## Statistical analysis

Statistical analyses were performed using GraphPad Prism 10 (Dotmatics, CA). Data are presented as the mean ± SEM. Two-tailed Unpaired t-tests or Mann–Whitney tests (where data were non-normally distributed) were performed for univariate comparisons between two groups. The EnrichR analysis for the DEGs used Fisher's exact test, while the DESeq2 analysis of the DEGs involved a negative binomial generalized linear model. One-way ANOVA was used while comparing more than 2 groups, with Tukey's/Dunnett's or Holm's-Sidak/Bonferroni-Holm's tests for subgroup comparisons. Statistical significance was considered with a two-tailed $p < 0.05$.

## Figures and Illustrations

The figures were made using Adobe Illustrator v28.0 (Adobe Inc., 2024) and the graphs were made using GraphPad Prism. The schematic illustrations used in the figures were scientific images/illustrations from BioRender (Toronto, ON). Mouse cartoon images in Figs. 2a, 3a, 4a, 5a, 5j, 6a, 8a were created in BioRender. Caplan, M. (2026) https://BioRender.com/nxzeqir and the summary figure (Fig. 9) was created in BioRender. Caplan, M. (2026) https://BioRender.com/6ykyx16.

## Reporting summary

Further information on research design is available in the Nature Portfolio Reporting Summary linked to this article.

## Data availability

The authors declare that all other relevant data supporting the findings of this study are available in this article and its Supplementary Information files. Source data supporting the figures of this manuscript are provided with the paper. The raw files of Bulk RNA sequencing data are available at the GenBank Bioproject under Accession PRJNA1221928.

Any data not available in the manuscript files listed above, or within the Genbank submission or on the github repository will be made available from the corresponding authors upon request. Source data are provided with this paper.

## Code availability

The analysis code used with RNAseq data has been deposited and is publicly available in GitHub at https://github.com/nrajeevan/Shroom3-Rock_interaction_and_profibrotic_function.git.

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

## Acknowledgments

MCM acknowledges funding from the NIH (grants R01DK122164, R01DK132274, and R21AI178705), and the Department of Defense, grants HT94252310454 and HT94252310441. MCM also acknowledges research support from the Blavatnik Fund at Yale (Accelerator award) and a pilot award from CTSA Grant UL1 TR001863. The authors acknowledge the support from Yale Center for Genomic Analyses (YCGA), Yale Pathology Tissue Services (YPTS) and Yale Animal Resources Center (YARC). The authors gratefully acknowledge Dr. Michael Caplan (Yale) for providing access to his BioRender account for figure preparation.

## Author contributions

A.R. and A.K. contributed to the experimental work, data analysis, data interpretation, drafting, and editing of this manuscript; Q.L. Experimental, data analysis work, and manuscript editing; N.R. supervised the RNA sequencing and was involved in data generation and manuscript drafting; Z.S. was involved in drafting the manuscript, RNA sequencing data interpretation and data collation from published scRNAseq datasets; K.B. contributed to experimental and data analysis; H.S. Mice surgeries and genotyping work; G.B. contributed to histological quantitation and manuscript editing; E.M.T. and J.P. were involved in data curation, analysis and editing the manuscript; S.P. interpreting pathological data from mice; C.W. contributed to data curation and interpretation and editing of the manuscript; B.B. contributed to data generation and curation, validation of mouse models; M.P. contributed to data generation, characterization of the mouse models; A.E. Data interpretation edited and reviewed this manuscript; V.M. data generation and interpretation. W.Z. RNA sequencing data interpretation. L.G.C. Data interpretation edited and reviewed this manuscript. L.X. DNA sequencing data generation and interpretation. B.D. Design and synthesis of Shroom3-Rock interaction inhibitors/drug derivatives and data interpretation. C.J.H. Conceptualized and designed this study, data interpretation, and supervised/edited this manuscript. M.C.M. Conceptualized and designed this study, data interpretation, drafted/edited and supervised this manuscript. All authors reviewed and approved the manuscript.

## Competing interests

L.G.C. declares roles as advisor to Pfizer for AKI and CKD and to Dropshot for AKI. The remaining authors declare no competing interests.

## Additional information

[1]Section of Nephrology, Department of Internal Medicine, Yale University School of Medicine, New Haven, CT, USA. [2]Department of Nephrology, Renji Hospital, School of Medicine, Shanghai Jiao Tong University, Shanghai, China. [3]Biomedical Informatics and Data Science, Yale University School of Medicine, New Haven, CT, USA. [4]Division of Nephrology, Department of Medicine, Icahn School of Medicine at Mount Sinai, New York, NY, USA. [5]Department of Pathology, Yale University School of Medicine, New Haven, CT, USA. [6]Department of Molecular and Cellular Physiology, Yale University School of Medicine, New Haven, CT, USA. [7]Surgical Sciences Division, Department of Surgery, School of Medicine, University of Maryland, Baltimore, MD, USA. [8]University at Buffalo, SUNY Buffalo, Buffalo, NY, USA. [9]These authors contributed equally: Anand Reghuvaran, Ashwani Kumar. [10]These authors jointly supervised this work: Bhaskar Das, John Cijiang He and Madhav C Menon. ✉e-mail: madhav.menon@yale.edu

