## [Transparent Peer Review file · Nature Communications]

Design of precision therapeutics for a CKD risk allele by targeting Shroom3-Rock interaction

Corresponding Author: Madhav Menon

Version 0:

Reviewer comments:

Reviewer #1

(Remarks to the Author)

Dr. Menon and team investigate the role of the Shroom3-ROCK interaction in fibrosis and explores the potential of Shroom3-ROCK interaction inhibitors (P2Is) as a therapeutic strategy for CKD. The work is comprehensive, employing in vitro, in vivo, and computational approaches. The use of different transgenic mouse models strengthens the study, and the development of D4G as a potential therapeutic is a notable advancement. However, several areas require clarification and improvement. Key mechanistic aspects remain unclear, particularly regarding ROCK1 vs. ROCK2 specificity, the dominant-negative effect of ASD2Δ-Sh3, and the role of fibroblast-derived Shroom3 in fibrosis. Additional protein-level validation would support RNA-seq findings if possible.

Below are my specific comments.

Figure 1C, D, and Figure S1H: The study primarily focuses on ROCK1, but it remains unclear whether ASD2Δ-Sh3 also affects ROCK2. Given that ROCK1 and ROCK2 have overlapping yet distinct functions, including Western blot data for ROCK2 in Figure 1E would clarify whether ASD2Δ-Sh3 selectively impacts ROCK1 or both isoforms. If ROCK2 was not assessed, a brief explanation of the rationale would be helpful.

Figure 1E: The pMYPT1 blot is normalized to β-Actin rather than total MYPT1, which may not accurately reflect phosphorylation changes. Normalization to total MYPT1 would provide a more precise assessment.

Figure S1H, S7B, and S7F (lower panel): pMYPT1 is shown without a corresponding total MYPT1 blot, which is provided in Figure S3E. Including a total MYPT1 blot here would improve data interpretation and completeness.

Figure 1F: The wound healing assay is an appropriate functional test, but the inclusion of ROCK inhibitors in WT-Sh3 and ASD2Δ-Sh3 cells would confirm that ASD2 deletion directly affects migration through ROCK inhibition.

Figure 2B: Prior scRNA-seq data suggest that ASD2Δ-Sh3 expression is minimal in podocytes. Could the authors clarify whether ASD2Δ-Sh3 primarily affects tubular cells and does not influence podocyte function? Explicitly confirming this would strengthen the interpretation of ASD2Δ-Sh3's tissue-specific effects.

Figure 2I-J: Electron microscopy results show that ASD2Δ-Sh3 does not cause podocyte foot process effacement, whereas FBDM-Sh3 induces podocyte structural abnormalities and albuminuria (Figure S2F). Given the essential role of ROCK in podocyte cytoskeletal regulation, it would be helpful if the authors could elaborate on why ASD2Δ-Sh3 does not disrupt podocyte structure.

Is this due to a cell-type-specific effect, or could differences in ROCK activation between podocytes and tubules explain this phenomenon? A brief discussion on this point would provide additional mechanistic insight.

Figures 3C-E and 4C-F: While the study demonstrates that ASD2Δ-Sh3 reduces fibrosis in multiple models, it remains unclear whether its protective effects are solely fibrosis-related or if they also extend to direct tubular protection. Since Figure S3A suggests potential tubular preservation in ASD2Δ-Sh3 mice, including tubular injury markers such as Kim-1, NGAL, and Lcn2 would help distinguish whether ASD2Δ-Sh3 protects tubules beyond its effects on fibrosis. Providing this information would strengthen the interpretation of ASD2Δ-Sh3's role in kidney injury.

Figures 3I-K and 4G: RNA-seq analysis identifies fibrosis-related gene changes, but key fibrosis markers (TGF-β, COL1A1, α-SMA) were not validated at the protein level. Western blot or IHC confirmation of these markers would strengthen the mechanistic conclusions.

Figure 5: ASD2Δ-Sh3 overexpression in Pax8-rTA mice reduces fibrosis, but its direct effects on tubular injury remain unclear. Similar to Figures 3 and 4, including tubular injury markers (Kim-1, NGAL, Lcn2) would clarify whether ASD2Δ-Sh3

protects tubules independently of its fibrosis-reducing effects.

In addition, analyzing the correlation between fibrosis severity and ACR would provide further insights into ASD2Δ-Sh3's functional impact on kidney health.

Figure 6B, D: Most of the figures in this manuscript, Collagen I was primarily used to assess fibrosis, while Figure 6 focuses on Collagen III staining. Could the authors share the rationale behind this choice? Was the collagen I also analyzed in this fibroblast-specific ASD2Δ-Sh3 overexpressing mice compared to WT mice ?

Figure 7D-F: D4G effectively inhibits ROCK activity, but its selectivity for ROCK1 vs. ROCK2 remains unclear. Conducting ROCK1 vs. ROCK2 knockdown experiments would clarify whether D4G preferentially inhibits one isoform.

Figure 8: D4G reduces fibrosis and ROCK activity in UUO-induced injury, but further validation is needed. Western blot analysis for fibrosis markers such as α-SMA, COL1A1, and TGF-β1 would provide further mechanistic confirmation of D4G's anti-fibrotic effects.

Additional Minor comments:

Figure S2A: The expression of the Shroom3 mutant in Figure S2A is not clearly distinguishable. It would be helpful if the authors could indicate the expression regions with arrows or other markers to improve clarity. This would make it easier for readers to interpret the data.

Figure Organization & Consistency : For better readability and consistency, it would be beneficial if all figures followed a uniform layout and orientation. Aligning the figure panels in a more structured and consistent manner would improve the overall presentation of the data.

Reviewer #2

(Remarks to the Author)

Reviewer #3

(Remarks to the Author)

This manuscript describes the identification of a Shroom3 and ROCK interaction site and its potential applications for developing drugs to treat CKD. The manuscript is well written and organized, and the MOA studies are clear and well demonstrated. The identification of hit/probe molecules for this PP interaction by AI drug discovery were also well described and the compound optimization (limited) is also rational. The identifications of both the PP interaction and the probe compounds are novel. Therefore, the manuscript could be accepted for publication in Nat Comm. From the view point of drug discovery and medicinal chemistry, for both the authors and the editor to consider, the following comments might be taken into consideration (optional):

1, The potassium trifluoroborate moiety (in compd 4G4 or BT1137) has never before be used in a drug-like compound, what is the rational the researchers advanced this compound into animal studies? any anticipated tox and side effects?

2, In animal studies, Cmpd 4G4 was dosed IP at 50 mg/kg. What is the rational for this dosage and the rational for IP dosing (not po dosing). Any in vitro and in vivo PK data obtained for compd 4G4? What is or expected to be the plasma exposure level of compd 4G4 at this IP dosage? Is this exposure level much higher than the cell-based EC50/IC50 (also need to take into calculation for the plasma-protein binding of compd 4G4 in mice plasma). If the 50 mg/kg dosage was chosen to present just because it showed good enough efficacy? If so, maybe the dose-response graph should be shown.

3, what is the formulation for the IP dosing? Does this formulation help the complete dissolution of 4G4 after IP dosing? A common problem is that, for some compounds which are not so soluble, IP dosing might over-estimate the efficacy due to the solubility issues. If 4G4 precipitated after IP dosing, 4G4 will be slowly released, and thus IP dosing will functionally enhance the psudo in vivo stability of 4G4 and hence increase the in vivo half-life.

4, I noticed that cell assays in HEK293 cells were used to screen compounds and to evaluate the cell-based efficacy of the compounds. I have no objections for this and it actually served very well the purpose of MOA studies. However, I also noticed that the IC50/EC50 in Hek293 cells are all expected to be higher than 20 uM based on the data shown. My question is why not to use a more relevant cell-line to do these studies (more related to CKD)? My personal observation is that, many compounds have lower solubility and/or lower cell-permeation in the HEK293 cell media, thus resulting in much lower activity (higher EC50/IC50 values) in HEK293 cells than in those more disease-related cells. For drug discovery, this might give misleading information. For example, the IC50 value in HEK293 cells is 25 uM, but in CDK related cells, it might be ~ 1 uM. In vivo PK of 4G4 in mice might show that 50 mg/kg IP dosing is not enough if based on the cell efficacy data in HEK293 cell (EC50: 25 uM), but it will be enough if based on the cell efficacy data in a CDK related cell line (EC50: ~ 1 uM).

Reviewer #4

(Remarks to the Author)

The authors elucidated the interaction with Rock and ASD2 domain in Shroom3, using mice with ablation of ASD2 domain. They also created the compound that interrupts the interaction with Rock and ASD2, which might be a promising drug for kidney disease. However, there are some points to be addressed.

[Major]

1. The dataset used by the authors in Figure 1A includes the IRI and UUO model at different time points. Therefore, it is necessary to clarify which model and timepoint the authors analyzed. In addition, according to the result from the website

(<https://humphreyslab.com/SingleCell/>), the expression patterns of Shroom3 and Rock are different, suggesting that Rock expression is likely regulated by other molecules.

2. Figure 1A and supplemental Figure 1a show injured PT and fibroblast expressed Shroom3, Rock1 and Rock2 higher than other cells, however, the expression patterns were quite different.

In Figure 1A, Fibroblast expressed these three genes higher, and injured PT had less expression of shroom3. On the other hand, in supplemental Figure 1A, fibroblast expressed Rock1 and Rock2, but did not express Shroom3 while injured PT expressed Shroom3 and Rock2.

3. You adopted UO model in the experiment of Figure 4 and 6 while you used AAN model in Figure 5. In UO model, kidney has less expression of Shroom3 until day 7 according to Humphreys website (see attached file).

Therefore, you could not evaluate how ASD2delta-Sh3 in fibroblast affected kidney in UO model. Please explain the reason that you adopt UO model, not AAN model in Figure 6.

4. Regarding Figure 1G and 1I, the authors concluded that tubular cells from ASD2delta-Sh3 mice had less ability to migrate and proliferate. And ASDdelta-Sh3 was protective against kidney injury. However, in previous reports (J Clin Invest. 2019;129(12):5501–5517, Nat Rev Nephrol 16, 269–288 (2020)), recovery from injury requires proliferative ability in tubular cells, which is not consistent with your result.

5. In Figure 8d, please add the data on Vimentin, Col1a1, Fsp1, Mmp2 as shown in Figure 4.

6. To confirm that D4G is effective for fibrosis, the authors should perform an additional experiment with AAN model.

7. Please clarify in which area of kidney and who evaluated fibrotic score.

[Minor]

1. Another research team has reported that a different SNP (rs142647267) was significantly associated with a decline in eGFR in a study of the Shroom3 locus. Please discuss these other studies in detail.

2. There are many abbreviations in the text without initial spell-out, making it difficult to read, and this needs to be resolved.

Version 1:

Reviewer comments:

Reviewer #1

(Remarks to the Author)

All my comments have been addressed and the manuscript is suitable for publication

Reviewer #2

(Remarks to the Author)

Reviewer #3

(Remarks to the Author)

The authors have answered all of my comments. It is Ok for me to accept it now.

Reviewer #4

(Remarks to the Author)

Thank you for your special efforts to revise this manuscript. I believe it has improved considerably. However, to preserve the robustness of the work in Nat Commun, the following points still require attention; I would appreciate your continued consideration.

1. Contradiction in expression patterns in Figure 1A/S1a

→ From the dot plots, it appears that at most ~10% of cells within the proximal tubule and myofibroblast/fibroblast populations express the genes in question, and likely at quite low levels. If violin plots are shown, it will probably become clear that there is essentially no expression. It is not reasonable to discuss genes that are virtually not expressed in the single-cell data, and this likely contributes to the apparent discrepancy between Figure 1A and S1a. Even without the single-cell data the manuscript is sufficiently interesting, so it would be better to remove these data.

2. Insufficient explanation for the contradiction that ASD2Δ-Sh3 suppresses proliferation yet confers renoprotection
→ Prior reports (e.g., JCI 2019) indicate that tubular cell proliferation is required for renal recovery, but this apparent contradiction has not yet been addressed.

3. Mismatch between Shroom3 and ROCK expression patterns (Humphreys dataset)

→ Although the rebuttal compared multiple datasets to explain this, the main text does not explicitly state it. Please incorporate this discussion into the Discussion section.

4. Other

→ Because numerous new references were added in the revision, many citation numbers have changed. Please carefully check that numbering and cross-references are consistent.

Version 2:

Reviewer comments:

Reviewer #4

(Remarks to the Author)

With regard to the reviewer's concerns, the authors have addressed the issues as follows. As a result, the manuscript has become logically coherent, more readily comprehensible, and in my view has been further polished.

Contradiction in expression patterns in Figure 1A/S1a

→ The authors re-examined the data using violin plots, removed non-essential data in accordance with the reviewer's recommendation, and revised the relevant paragraph in the text.

Contradiction that ASD2Δ-Sh3 suppresses proliferation yet confers renoprotection

→ A new paragraph has been added to the Discussion, elaborating on the apparent discrepancy within the context of AKI versus CKD, with additional explanation involving Ccl2/ROCK signaling.

Mismatch between Shroom3 and ROCK expression patterns (Humphreys dataset)

→ The variability across datasets has now been explicitly acknowledged and incorporated into the Limitations section.

Inconsistencies in reference numbering and cross-references

→ The authors have stated that they have carefully reviewed and corrected these inconsistencies.

Minor points:

- As noted, it would be appropriate to spell out iPTs (injured proximal tubules) at its first appearance in the main text. In fact, the rebuttal document already provides this clarification as "iPTs (injured proximal tubular cells)."
- FBDM-Sh3 (Fyn-binding domain mutant Shroom3): This abbreviation should be defined at its first occurrence.
- P2Is (Shroom3-Rock interaction inhibitors): Although the definition is provided in the text, the introduction of this abbreviation is somewhat abrupt. It would be clearer and more reader-friendly to introduce it more explicitly at first mention, for example: "we developed novel small molecule inhibitors of Shroom3-Rock interaction (P2Is)."
- Several references appear to be incomplete. In particular, references 7, 18, 24, 55, and 79 seem to be missing page numbers.
- Additional inconsistencies may also exist, for example in references 25–27 (murine TIF models) and 65–67 (recently added AKI-related studies). These should be checked carefully to ensure that full bibliographic details, including page ranges, are provided.

To our Editor and Reviewers,

We would like to thank you for the thorough evaluation of our manuscript and providing us with constructive feedback. We greatly appreciate the opportunity to revise and resubmit our work.

We have carefully considered all the comments and concerns raised by the reviewers and the editorial team. In response, we have done substantial revisions and additional experiments, which we believe have significantly strengthened the rigor and impact of the study.

All reviewer comments have been addressed in detail in the accompanying point-by-point response document. Revisions to the manuscript have been clearly highlighted as per journal guidelines.

We have ensured full compliance with all editorial policies and requirements, including:

- Completion of the Editorial Policy Checklist and attached to the Reporting Summary.
- Updated Data Availability section with deposition of datasets in appropriate public repositories.
- Providing a comprehensive Source Data files for all main figures as ppt and tables in Excel format.
- Included more data into the panels from new experiments suggested by all the reviewers.
- Performed adjustments of all figures and bar graphs to appropriate formats for displaying data distribution.
- Consideration and appropriate reporting of sex and gender in line with SAGER guidelines

We enclose in the revised submission (i) a marked copy (yellow highlights for changes in revision), (ii) a clean copy of the manuscript file, (iii) Supplementary information (Supplementary figures/legends and Reporting summary as separate documents) and (iv) Supplementary datasets 1-4.

We confirm that there have been no changes to the author's list. In order to give due credit to Dr Bhaskar Das's contributions in drug discovery/drug development/synthesis part of this manuscript (including the revision) we have included Dr Das as a co-corresponding author for this manuscript.

We sincerely hope that the revised version addresses all the concerns raised and meets the expectations of the reviewers and editorial board.

Point by point response to all reviewers' comments are listed below:

REVIEWER COMMENTS

Reviewer #1 (Remarks to the Author):

Dr. Menon and team investigate the role of the Shroom3-ROCK interaction in fibrosis and explores the potential of Shroom3-ROCK interaction inhibitors (P2Is) as a therapeutic strategy for CKD. The work is comprehensive, employing in vitro, in vivo, and computational approaches. The use of different transgenic mouse models strengthens the study, and the development of D4G as a potential therapeutic is a notable advancement. However, several areas require clarification and improvement. Key mechanistic aspects remain unclear, particularly regarding ROCK1 vs. ROCK2 specificity, the dominant-negative effect of ASD2 Δ -Sh3, and the role of fibroblast-derived Shroom3 in fibrosis. Additional protein-level validation would support RNA-seq findings if possible.

We appreciate the reviewer's positive assessment of our study and their thoughtful suggestions, which have helped us to strengthen and improve this work manuscript.

QR1.1. Figure 1C, D, and Figure S1H: The study primarily focuses on ROCK1, but it remains unclear whether ASD2 Δ -Sh3 also affects ROCK2. Given that ROCK1 and ROCK2 have overlapping yet distinct functions, including Western blot data for ROCK2 in Figure 1E would clarify whether ASD2 Δ -Sh3 selectively impacts ROCK1 or both isoforms. If ROCK2 was not assessed, a brief explanation of the rationale would be helpful

Reply: We thank the reviewer for raising this important question regarding the potential role of ROCK2 in the context of SHROOM3-ASD2 deletion. As ROCK1 and ROCK2 contain conserved Shroom3-binding domains (SBD), the ASD2 deletion is expected to disrupt interactions with both paralogues.

In the revision to address this question by the reviewer, we have included both in vitro and in vivo data that assesses changes in ROCK2. (Supplementary Fig 1)

1. First, we show data confirming Rock2 binding is inhibited by ASD2 deletion (similar to ROCK1 already shown in Figure 1c). Here in figure R1, lysates from HEK cells transfected with GFP, WT-Sh3, and ASD2 Δ -Sh3 plasmids were immunoprecipitated using V5-conjugated agarose beads. The resulting data are included in the revised manuscript as Fig S1c.

1. We repeated the experiment shown in Fig 1D-E comparing IMCD cells expressing either WT-Sh3 or ASD2 Δ -Sh3, this time probing for ROCK2. Consistent with the effect observed for ROCK1 (shown in Fig 1 D-E), we detected a reduction in ROCK2 expression in the ASD2 Δ -Sh3 group. These data are presented here as Fig R2 for the reviewer/editor reference but are not currently included in the revised manuscript to avoid redundancy, as ROCK2 down regulation is referenced in multiple sections of the text.

R2

2. Similarly, in vitro experiments using 3T3 cells expressing ASD2Δ-Sh3 also showed reduced levels of both ROCK2 and ROCK1. These immunoblot data are shown below in response to reviewer comment QR1.3 and are included in the revised manuscript in Fig S1i.

3. Our P2I compounds, which inhibit ASD2-Rock SBD interaction, led to a reduction in both ROCK1 and ROCK2 levels. These data, included in revised manuscript as Figure 8/S8, further support the involvement of both ROCK paralogues in SHROOM3 ASD2-domain.
2. Analogous with our data on the effect of Shroom3 ADS2 domain on Rock2 activation, an independent study used in vitro screening to identify a relatively specific Shroom3-Rock2 interaction inhibitor focused on neurogenesis after stroke. We have cited this paper ¹ in Result #7 and in Methods under the heading 'Development of P2I inhibitors'. To clarify this question in the revised manuscript, we also included the following in the discussion to highlight inhibition of both ROCK paralogues by the Shroom3-ASD2 directed interventions.
"Critically, the protein-protein interaction-based, pro-injury signaling mechanism that we describe allowed for the development of P2Is that inhibited Shroom3-Rock interaction and mitigated TIF¹."

QR1.2. Figure 1E: The pMypt1 blot is normalized to β -Actin rather than total Mypt1, which may not accurately reflect phosphorylation changes. Normalization to total Mypt1 would provide a more precise assessment.

Reply: Reply: We thank the reviewer for this important comment. . In response, we quantified phosphor-MYPT1 (pMYPT1) normalized to total MYPT1 (tMYPT1) rather than β -Actin. As shown in Fig. R4 densitometric analysis

of pMYPT1 normalized to total MYPT1 confirms the results presented in Fig. 1E, supporting the conclusion that phosphorylation changes are independent of total protein abundance

R4

QR1.3. Figure S1H, S7B, and S7F (lower panel): pMypt1 is shown without a corresponding total Mypt1 blot, which is provided in Figure S3E. Including a total Mypt1 blot here would improve data interpretation and completeness.

Reply: We apologize to the reviewer for this oversight. We have now corrected Figs S1h, S7b and S7g (earlier S7F lower panel) to include Mypt1(blots shown below).

Figure S1h has been added with blots of Rock2 and total Mypt1 from 3T3 lysates- WT-Sh3 and ASD2-Sh3 as shown above in Figure R3.

Figure S7b, Drug D4-D7 total-Mypt1

R5

Figure S7g has been added with a blot of total Mypt1 from 3T3 lysates- Drug E, F, G 1-10μM

R6

QR1.4. Figure 1F: The wound healing assay is an appropriate functional test, but the inclusion of ROCK inhibitors in WT-Sh3 and ASD2Δ-Sh3 cells would confirm that ASD2 deletion directly affects migration through ROCK inhibition.

Reply: We thank the reviewer for this valuable suggestion. We repeated the wound healing assay in IMCD cells, adding HF at 10 μM. We used lower dose of HF to avoid any confounding by cell death observed at 30 μM.

The percentage of migrated area was then calculated at 16-hrs (as before) and normalized to the migration by the WT-Sh3-IMCD in the same experiment. We observed that when ROCK activity was reduced by HF-10μM, the difference in migration between ASD2Δ-Sh3 cells vs. WT-Sh3 significantly narrowed but was not abolished.

This lack of complete abolition of differences in migration could be because of two reasons. (1) As shown in Fig 1d-e, MYPT phosphorylation is still detectable in IMCD lysates even at 30 μM HF, suggesting residual Rock activity persists and is not completely abolished by HF. (2) The significantly reduced total endogenous Rock1, Rock2 levels observed with ASD2Δ-Sh3 also represent a difference that would persist regardless of the presence of HF. Hence these data support the role of Shroom3-Rock interaction in mediating differences in migration between ASD2Δ-Sh3 and WT-Sh3.

We have included this data in updated Figure 1f and Supplementary Figure S1h-i and shown here as R7.

R7

QR1.5-6: Questions pertaining to the role of ASD2-domain of Shroom3 in Podocytes vs Tubular cells.

QR1.5. Figure 2B: Prior scRNA-seq data suggest that ASD2Δ-Sh3 expression is minimal in podocytes. Could the authors clarify whether ASD2Δ-Sh3 primarily affects tubular cells and does not influence podocyte function? Explicitly confirming this would strengthen the interpretation of ASD2Δ-Sh3's tissue-specific effects.

QR1.6. Figure 2I-J: Electron microscopy results show that ASD2Δ-Sh3 does not cause podocyte foot process effacement, whereas FBDM-Sh3 induces podocyte structural abnormalities and albuminuria (Figure S2F). Given

the essential role of ROCK in podocyte cytoskeletal regulation, it would be helpful if the authors could elaborate on why ASD2 Δ -Sh3 does not disrupt podocyte structure. Is this due to a cell-type-specific effect, or could differences in ROCK activation between podocytes and tubules explain this phenomenon? A brief discussion on this point would provide additional mechanistic insight.

Reply: We thank the reviewer for highlighting this excellent point and allowing us to discuss in detail why ASD2-deletion mediated lack of ROCK facilitation did not have significant in vivo consequences in podocytes. We believe the reviewer is pointing to 2B which is the immunofluorescence image (and not the scRNA data). The lack of podocyte phenotype in our model of ASD2 Δ -Sh3 overexpression (therefore Rock inhibition) could be due to the following reasons:

1. Regulation of Rho GTPase balance in podocytes: We acknowledge that simultaneous Rock1/Rock2 knockout in podocyte-specific models, have not yet been performed, and is a limitation in the field. However, Rho kinases (which Shroom3 facilitates) activate podocyte RhoA, are extensively studied with conflicting results. This inconsistency may be due to the finely tuned balance of Rho-GTPases that ultimately regulate susceptibility to injury (reviewed by Reiser et al 2012²). For instance, dysfunctional RhoA over-expression in podocytes induced injury³; however, over-expression of active RhoA also induced injury. Podocyte specific Rac1 knockout (i.e. with unopposed RhoA) offered protection in injury models⁴. In the specific context of Shroom3, reports show that Shroom3⁵ regulates Synaptopodin levels via 14-3-3 binding motif⁶, which in turn can facilitate RhoA levels/signaling⁷. Hence, the overall podocyte phenotype involves a combination of factors that regulate Rho-GTPases in each transgenic model including ours.
2. Potential beneficial effects of RhoA/Rock deficiency in podocytes: Next, Podocyte-specific RhoA-deficient mice themselves were normal without proteinuria⁸. Podocyte-specific Rock2-deficient were also normal⁹, and in the setting of injury, Rock2-Knockout mice exhibited *protection* from proteinuria in models of FSGS^{9, 10}. Similarly, Rock1 deficiency also led to less albuminuria^{9, 11}. Analogously, data show that RhoA/Rho-kinase pathway is upregulated in glomerular injury accompanying diabetes¹². In this milieu, Rho-kinase inhibitors reduced glomerular damage from diabetes^{13, 14, 15, 16}. The Rho-kinase inhibitor (Fasudil) also had beneficial effects on podocyte foot process effacement¹². Together these data point to a degree of dispensability of the Rock-RhoA axis downstream of Shroom3 ASD2-domain in podocytes and support the lack of podocyte phenotype in our ASD2 Δ -Sh3 mice.
3. High endogenous Shroom3 expression in Podocytes: We also note that in single cell datasets (including those shown in Figure 1a and S1a-b), constitutive Shroom3 expression in podocytes is indeed high both in homeostasis and during injury. As shown in Figure 2b endogenous Shroom3 protein was higher (by immunofluorescence) in glomeruli of uninjured control mice vs the tubulo-interstitial compartment. After DOX feeding, we also observed overexpression to a similar extent in glomeruli of ASD2 Δ -Sh3 and WT-Sh3 mice by immunofluorescence. Conjecturally, in the presence of high constitutive Shroom3 levels, the overexpressed ASD2-deletion mutant could have less dominant negative effect (especially if the effect is mild) over the endogenous protein's activity. However, the use of D4G (Figure S8d) also did not induce proteinuria.
4. We accept that proteinuric phenotype could also result from defects in reabsorption of albumin as a tubular effect of ASD2-deleted Shroom3 which we did not specifically test.

Hence in summary, the protective role of RhoA-Rock axis in podocytes in homeostasis is controversial, and dependent on Rho-GTPase balance in each model, and inhibition of Shroom3-Rock interaction alone in our model likely does not induce significant podocyte dysfunction. This is consistent with the absent podocyte foot process effacement and/or proteinuria seen in vivo in global ASD2 Δ -Sh3 excess.

On the other hand, FBDM-Sh3 mice exhibit podocyte FPE due to alteration of a different motif in Shroom3. The mutation in the Fyn-binding motif (i.e. -PxxP- to -AxxA-) leads to reduced interaction with Fyn which is crucial for

nephrin cytoplasmic tail phosphorylation in podocytes and maintenance of podocyte cytoskeleton by binding Nck adapter proteins¹⁷. The role of Fyn inactivation or deficiency in podocytes impacting Nphs1 phosphorylation, actin cytoskeleton has been independently reported in animal models^{18, 19} and proteinuric human diseases²⁰.

As suggested, and to clarify the reviewer’s point regarding podocyte function, we have included the following in the revision:

” Indeed, in the setting of diabetic kidney injury, global Rock1 deficiency or podocyte-specific Rock2 deficiency led to less albuminuria and a protective phenotype^{9, 11}. Furthermore, via a distinct 14-3-3 binding motif⁶ Shroom3 could regulate Synaptopodin levels in podocytes¹⁷ and facilitate RhoA signaling⁷, providing a potential explanation for the minimal impact of ASD2-motif antagonism on podocyte phenotypes”

“...high endogenous expression of Shroom3 may attenuate the dominant negative effect of ASD2Δ-Sh3 in cell types such as fibroblasts and podocytes with high constitutive Shroom3 levels.”

QR1.7. Figures 3C-E and 4C-F: While the study demonstrates that ASD2Δ-Sh3 reduces fibrosis in multiple models, it remains unclear whether its protective effects are solely fibrosis-related or if they also extend to direct tubular protection. Since Figure S3A suggests potential tubular preservation in ASD2Δ-Sh3 mice, including tubular injury markers such as Kim-1, NGAL, and Lcn2 would help distinguish whether ASD2Δ-Sh3 protects tubules beyond its effects on fibrosis. Providing this information would strengthen the interpretation of ASD2Δ-Sh3’s role in kidney injury.

Reply: We thank the reviewer for this most helpful suggestion. We have now systematically evaluated profibrotic, tubular injury/inflammation and tubular homeostasis markers in the mRNA expression panel in figure 3k and similarly under each relevant *in vivo* experiment. The injury markers Kim1, Lcn2, Hmgb1, and Il18 tended to be reduced in the ASD2Δ-Sh3 in comparison with the WT-Sh3 kidneys, whereas the tubular homeostasis markers Lrp2 and Cdh16 were upregulated in the ASD2Δ-Sh3 mice.

To evaluate the protein level expression of markers of tubular health in kidney sections, we stained paraffin-embedded sections from AAN kidneys for KSP-cadherin (CDH16) a pan-tubular homeostatic marker using IF. KSP-cadherin was significantly higher in AAN kidneys from ASD2Δ-Sh3 as compared to WT-Sh3 kidneys, suggesting better preservation of homeostatic gene expression in tubules of ASD2Δ-Sh3 mice with reduced TIF. These data have been added in manuscript (Figure S3i-j).

R9

We also evaluated Megalin (LRP2) which showed similar trend of increased expression in ASD2Δ-Sh3, however, this data is only shown here for reviewers' and not included in the manuscript to minimize redundancy and for space concerns.

R10

QR1.8. Figures 3I-K and 4G: RNA-seq analysis identifies fibrosis-related gene changes, but key fibrosis markers (TGF- β , COL1A1, α -SMA) were not validated at the protein level. Western blot or IHC confirmation of these markers would strengthen the mechanistic conclusions.

Reply: We thank the reviewer for suggesting additional validation of RNA-seq data using protein estimation for key fibrosis markers. We have already shown protein levels for COL-I and COL-III for AAN by immunostaining. We now included FIBRONECTIN1 immunostaining on paraffin sections of AAN and UUO kidneys (WT-Sh3 vs. ASD2Δ-Sh3) to show tissue localization. Fibronectin expression was quantified and found consistent with qPCR data, i.e., significantly higher in WT-Sh3 as compared to ASD2Δ-Sh3 in both AAN (Figure-S3c-d) and UUO (Figure-4f-g) models (Figure-R11A, B respectively). We did not include α -SMA in the fibrosis assessment as our staining proved inconsistent for this marker, and α -SMA has been reported to be an unreliable marker for fibrogenic cells in murine experimental models of organ fibrosis ^{21, 22}.

To test intrarenal canonical Tgfb1 signaling, we probed for phosphorylated-Smad3/Smad3 in kidney lysates from our AAN TIF model. We identified reduced P-Smad3 in ASD2Δ-Sh3 vs WT-SH3- AAN lysates. These data are shown in figure 3l-m, below (R11C) and are consistent with similarly reduced P-Smad3 which was seen in ASD2Δ-Sh3 UUO kidneys. We chose P-Smad3 levels, as Total TGFB1 levels are an unreliable assay since TGFB1 is activated extracellularly before receptor binding by cleavage of Latent peptide ^{23, 24, 25, 26}

QR1.9 Figure 5: ASD2Δ-Sh3 overexpression in Pax8-rtTA mice reduces fibrosis, but its direct effects on tubular injury remain unclear. Similar to Figures 3 and 4, including tubular injury markers (Kim-1, NGAL, Lcn2) would clarify whether ASD2Δ-Sh3 protects tubules independently of its fibrosis-reducing effects.

Reply: We thank the reviewer for this important suggestion. We have now analyzed tubular injury and homeostasis markers in AAN-kidneys of Pax8-rtTA mice as described below and confirmed that tubules are more preserved in the ASD2Δ-Sh3 kidneys. The tubular injury/inflammation markers *Kim1*, *Lcn2* and *Il18*, and *Ccl2* were significantly lower in tubular-specific ASD2Δ-Sh3 overexpressing kidneys vs. WT-SH3 (R12).

R12

On the other hand, the tubular homeostasis markers *Lrp2* and *Cdh16* were more preserved in ASD2Δ-Sh3 (R12 and Figure 5i). Similar findings were noted at the protein level by IF, showing better-preservation of tubular expression of CDH16 in ASD2Δ-Sh3 kidneys compared to WT-Sh3 (Figure R13 and Figure 5c & f).

R13

QR1.10. In addition, analyzing the correlation between fibrosis severity and ACR would provide further insights into ASD2Δ-Sh3's functional impact on kidney health.

Reply: We believe the reviewer is referring to 5H and S5D and appreciate the interesting observation. We performed linear regression (person correlation) analyses comparing the Fibrosis scores (Masson's staining) and the albuminuria (Albumin Creatinine Ratio). These plots are shown below for the reviewer Figure R14 (red triangles represent ASD2Δ-Sh3 mice in each plot). We could not identify clear or significant correlations in these analyses in either global or tubular specific Shroom3 excess ($P > 0.05$), although the reduced fibrosis along with tendency to reduced ACR was suggested in the tubular specific ASD2Δ-Sh3 overexpression mice.

R14

QR1.11. Figure 6B, D: Most of the figures in this manuscript, Collagen I was primarily used to assess fibrosis, while Figure 6 focuses on Collagen III staining. Could the authors share the rationale behind this choice? Was the collagen I also analyzed in this fibroblast-specific ASD2Δ-Sh3 overexpressing mice compared to WT mice?

Reply: As the reviewer requested, we now also measured Collagen I in the UUO kidneys of the fibroblast-specific Shroom3 (ASD2Δ-Sh3 vs. WT) by immunofluorescence. The data shown below for the reviewer (R15) are included in the revised manuscript Figure 6d-e. Comparing either Collagen I and III, fibroblast-specific ASD2Δ-Sh3 vs WT-Sh3 UUO kidneys showed no significant differences in fibrosis.

QR1.12. Figure 7D-F: D4G effectively inhibits ROCK activity, but its selectivity for ROCK1 vs. ROCK2 remains unclear. Conducting ROCK1 vs. ROCK2 knockdown experiments would clarify whether D4G preferentially inhibits one isoform.

Reply: We are grateful for the reviewer's important suggestion.

We did find a significant reduction in both Rock1 and Rock2 protein levels in vivo on D4G treatment (*Revised Figure 8/S8*). Our in vitro experiments also showed that in tubular and fibroblast cell lines overexpressing the Shroom3 variants, Rock2 levels were lower in the ASD2Δ-Sh3 vs WT-Sh3. All these are consistent with the concept that both Rock paralogs are expected to be inhibited by D4G.

To investigate whether D4G had preferential inhibitory effects on either ROCK1 or ROCK2 paralogs, we silenced either Rock1 or Rock2 using specific siRNA separately in WT-Sh3 overexpressing- HEK293T cells. We confirmed gene knockdown in each case by immunoblot (R16 or Fig 7n).

We then used the Rock assay (#MBL-CY-1160.) to test Rock activity inhibition by D4G in conditions of either Rock1 or Rock2 knockdown in WT-Sh3 overexpressing HEK293 cells (representing activity of D4G on SHROOM3-ROCK2 or SHROOM3-ROCK1 interactions, respectively). Our lead P2I D4G inhibited ROCK2 activation in a dose-dependent fashion (by ROCK assay) when ROCK1 was selectively knocked down using siRNA (Human-specific), and vice versa (Fig R17 or Fig 7o). The EC50 obtained from ROCK1 and ROCK2 silencing were similar (57 vs 50 μM).

R17

These experiments confirmed similar activity of our lead P2I on both paralog interactions. These data have been included in the revised manuscript as Figure 7n-o.

“To evaluate any paralog-specific effects of D4G, we knocked down either Rock1 or Rock2 in WT-Sh3 HEK-293 cells. After confirming ROCK1 and ROCK2 knockdown [Figure-7n], we performed ROCK activity assay using D4G. In these experiments, D4G inhibited ROCK1 and ROCK2 nearly identically [7o], suggesting the lack of any preferential inhibition of either paralog by our lead P2I “

QR1.13. Figure 8: D4G reduces fibrosis and ROCK activity in UO-induced injury, but further validation is needed. Western blot analysis for fibrosis markers such as α -SMA, COL1A1, and TGF- β 1 would provide further mechanistic confirmation of D4G’s anti-fibrotic effects.

Reply: As the reviewer suggested, we have measured more differentially expressed genes, including tissue expression of Fibronectin and Collagen-III. We also performed protein expression of Fn1 and Cdh16 by western blot. Collagen-III and Fibronectin IF images are now included in the revised Figure S8(j-m) and shown below as R18.

R18

D4G treated Pax8-WT-Sh3 UJO kidneys also showed significantly preserved CDH16 levels by both IF and immunoblotting in D4G vs DMSO treated Pax8-WT-Sh3 UJO kidneys. These data are shown below for the reviewer/editors, and included in revised supplementary figure as S8g-h. We did not include Cdh16 or Fn1 immunoblot lanes in the manuscript as we show significant differences in IF for these same proteins (to avoid redundancy).

These new data show consistent changes at the protein and mRNA levels and validating the anti-TIF and tubular preserving effect of D4G on TIF outcomes, and mimicking signals seen with ASD2 Δ -Sh3 overexpressing mice.

Additional Minor comments:

QR1.14. Figure S2A: The expression of the Shroom3 mutant in Figure S2A is not clearly distinguishable. It would be helpful if the authors could indicate the expression regions with arrows or other markers to improve clarity. This would make it easier for readers to interpret the data.

Reply: We now added arrows to clearly mark the Shroom3 bands.

QR1.15. Figure Organization & Consistency: For better readability and consistency, it would be beneficial if all figures followed a uniform layout and orientation. Aligning the figure panels in a more structured and consistent manner would improve the overall presentation of the data.

Reply: We apologize for this inconvenience and thank the reviewer for bringing this concern here. We have reformatted all figure panels again to fit into A4 size as recommended by the journal. All figures are in a portrait orientation except Fig 3, which needed to be in landscape orientation to accommodate the data panels without impeding legibility.

Reviewer #2 (Remarks to the Author):

We greatly appreciate the overall positive feedback from Reviewer 1 and 2 and believe the revised manuscript has improved with the incorporation of these suggestions.

Reviewer #3 (Remarks to the Author):

This manuscript describes the identification of a Shroom3 and ROCK interaction site and its potential applications for developing drugs to treat CKD. The manuscript is well written and organized, and the MOA studies are clear and well demonstrated. The identification of hit/probe molecules for this PP interaction by AI drug discovery were also well described and the compound optimization (limited) is also rational. The identifications of both the PP interaction and the probe compounds are novel. Therefore, the manuscript could be accepted for publication in Nat Comm. From the viewpoint of drug discovery and medicinal chemistry, for both the authors and the editor to consider, the following comments might be taken into consideration (optional):

We appreciate the thoughtful and very positive review by the reviewer, which has allowed us to provide details on the drug discovery aspect of this manuscript and improve this work.

QR3.1, The potassium trifluoroborate moiety (in cmpd 4G4 or BT1137) has never before been used in a drug-like compound, what is the rationale the researchers advanced this compound into animal studies? any anticipated tox and side effects?

Reply: We appreciate this interesting question from the reviewer. We used a boron-based compound since when Boron atoms are introduced into biologically active molecular frameworks, they interact with target proteins producing potent biological activity. Boron-containing pharmacophore groups interact with a target protein not only through hydrogen bonds but also through reversible covalent bonds to produce potent biological activities (i.e. antifungal, antiparasitic, protease inhibitors, and other drugs). This concept is well supported in the literature^{27, 28}. Proof of this concept is further provided by boron-containing compounds that are FDA approved such as the proteasome inhibitor Velcade (bortezomib) for multiple myeloma, antifungal Keydin (tavaborole, approved in 2014), Crisaborole (eucrisa, approved in 2016) for atopic dermatitis, and Vabomere (vaborbactam, approved 2017), an antibacterial β -lactamase inhibitor for complicated urinary tract infections and pyelonephritis. Notably, as formulated, tavaborole is only FDA approved for topical use for as a 5% solution to treat onychomycosis (nail fungus). A boron-based small molecule was used as a TYRO3 agonist and reported as a therapy for glomerular disease in a JCI-Insight article²⁹ by our co-authors.

For the lead compound we have performed toxicity studies in vitro (WT-Sh3 overexpressing IMCD) and is illustrated in Figure 7. In the in vivo setting, no overt adverse effects, ie no change in bodyweight and albuminuria (Figure S8c-d) were observed in D4G treated mice compared to DMSO treated mice. We also evaluated kidney injury in non-obstructed left kidneys from D4G-mice, which did not demonstrate increases in mRNA expression of injury markers, vs vehicle [shown as Figure-S8i] nor did they exhibit overt injury in PAS-stained sections [shown as Figure S8j]. We also performed targeted kinase profiling (in comparison with HF – a ROCK inhibitor) where D4G was incubated with 8 kinases (that have been reported variably to be inhibited by HF) at an ATP concentration of 10mM (Figure S7). In this kinase assay, D4G showed no off target inhibition of these kinases.

In this work, we identified our Target-to-Hit D4G compound, and structure activity relationship (SAR) studies are ongoing to develop further Hit-to-Lead compounds. As noted by the reviewer, while reflecting proof-of-concept inhibition and in vivo anti-TIF effect, the EC50 for D4G does not show high potency (in micromolar range; see response to QR3.2 below). Therefore, to design potent Hit-to-lead compounds in future work, we will be guided by the Lipinski rule of 5. New P2Is will be designed with considerations of the propensity for (1) biological activity, i.e., more sp³ carbon atoms; (2) ease of syntheses; and (3) moderate compound complexity, to minimize toxicity & off-target effects^{30, 31}. We will use ACDLabs ADMET suites (<https://www.acdlabs.com/products/percepta-platform/>) and Toxicity suites to predict drug-like properties of our designed compounds, then use Go/No-Go criteria to develop new compounds. These points will guide our molecular design choices and minimize the risk of potential toxic effects as we develop more potent P2Is in future studies.

QR3.2, In animal studies, Cmpd D4G was dosed IP at 50 mg/kg. What is the rationale for this dosage and the rationale for IP dosing (not po dosing). Any in vitro and in vivo PK data obtained for cmpd 4G4? What is or expected to be the plasma exposure level of cmpd 4G4 at this IP dosage? Is this exposure level much higher than the cell-based EC50/IC50 (also need to take into calculation for the plasma-protein binding of cmpd 4G4 in mice plasma). If the 50 mg/kg dosage was chosen to present just because it showed good enough efficacy? If so, maybe the dose-response graph should be shown.

Reply: We greatly appreciate these questions regarding the dosage and route we chose for our in vivo data.

In our pilot in vivo studies, we used the IP route to avoid any incidental reduction in bioavailability (and therefore efficacy) of the compound D4G due to the digestive enzymes, gastric acidity, or first pass liver metabolism. We acknowledge that formal *in vivo* PK/PD studies in small animals have not been done yet, which we have acknowledged as a limitation of the manuscript in its original submission. However, we did a priori insilico ADMET studies 2.0 and identified good-to-excellent predicted T1/2 and a predicted clearance of 1.39 ml/min/kg. The predicted plasma protein binding was 98.5% for D4G.

We performed in vitro PK studies using ROCK assay (#MBL-CY-1160) on WT-Sh3-IMCD cells. The EC50 of D4G is calculated to be 57.92µM from the equation derived from the percentage Inhibition plot (R20A). EC50 obtained in HEK-293T cells overexpressing WT-SH3 were also similar after either ROCK1 or ROCK2 knockdown (Fig R20B; 58.9 and 50.2 µM respectively). These data are included now as S8a-b.

On the other hand, the in vitro IC50 calculated from the Cell-glo toxicity assay on WT-SH3-IMCD cells was 613.2 µM (shown in Figure7 and the non-linear regression shown in R21).

Hence, we surmised that the initial efficacious dose, without risking significant toxicity was between EC50 and IC50 (57 to 613.2 μ M).

As we observed EC50 of \sim 57 μ M, we first calculated an in vivo dose of 20mg/kg to approximate the EC50 (assuming 100% bioavailability of the IP route). In these studies using WT-Sh3 mice we did not observe significant reduction in TIF in a UUO model. These data are now shown below for the reviewer (R22).

Since we did not observe protection at 20 mg/kg, we empirically escalated the dose by \sim 2.5 times the EC50 (i.e. \sim 142.5 μ M or 50 mg/kg), which is still well below the IC50 and repeated the efficacy study. This dose i.e. 50 mg/kg – based on which in vivo data included in Fig 8 is based in the manuscript – showed efficacy without significant toxicity (weight loss, proteinuria or histologic injury in control kidneys).

Our goal in ongoing work is to develop further potent Hit-to-lead compounds and perform formal PK/PD studies, which will include Plasma stability, protein binding, T1/2 and maximum tolerated dose studies on newer compounds.

QR3.3, what is the formulation for the IP dosing? Does this formulation help the complete dissolution of 4G4 after IP dosing? A common problem is that, for some compounds which are not so soluble, IP dosing might overestimate the efficacy due to the solubility issues. If 4G4 precipitated after IP dosing, 4G4 will be slowly released, and thus IP dosing will functionally enhance the pseudo in vivo stability of 4G4 and hence increase the in vivo half-life.

Reply: The compound was first dissolved in DMSO and then diluted to a final working concentration with corn oil to keep it in a non-aqueous solvent mixture. When we solubilized in DMSO, our compound was fully soluble without visible precipitation.

As we continue our studies with D4G, and develop refined formulations using SAR studies modifying the D4G pharmacophore, we anticipate formulating the drugs using cyclodextrin and Captisol, which will further increase the solubility.

We acknowledge the limitation that precipitation of the drug is a possibility; however, since the drug-treated mice showed a response, we conclude that there are significant amount of active molecules in circulation using our IP dosing schedule.

QR3.4, I noticed that cell assays in HEK293 cells were used to screen compounds and to evaluate the cell-based efficacy of the compounds. I have no objections for this and it actually served very well the purpose of MOA studies. However, I also noticed that the IC₅₀/EC₅₀ in Hek293 cells are all expected to be higher than 20 uM based on the data shown. My question is why not to use a more relevant cell-line to do these studies (more related to CKD)? My personal observation is that, many compounds have lower solubility and/or lower cell-permeation in the HEK293 cell media, thus resulting in much lower activity (higher EC₅₀/IC₅₀ values) in HEK293 cells than in those more disease-related cells. For drug discovery, this might give misleading information. For example, the IC₅₀ value in HEK293 cells is 25 uM, but in CKD related cells, it might be ~ 1 uM. In vivo PK of 4G4 in mice might show that 50 mg/kg IP dosing is not enough if based on the cell efficacy data in HEK293 cell (EC₅₀: 25 uM), but it will be enough if based on the cell efficacy data in a CKD related cell line (EC₅₀: ~ 1 uM).

Reply: We appreciate the reviewers point about the varying efficacy (EC₅₀) that could be observed due to different cell lines, specifically HEK-293 cells. We would like to clarify that we used 3 types of Shroom3-overexpressing cell lines in our efficacy and toxicity studies in experiments in Fig 7/S7/S8. For the majority of experiments, we used a WT-SH3-IMCD cell line to simulate the Shroom3 excess rendered by the CKD-associated SNPs, to assay the EC₅₀ and toxicity in vitro. We also used 3T3 cell lines (human fibroblast) to perform additional studies in fibroblasts. Finally, we also used HEK-293 cells in ROCK assay screening for compounds D1-D7, and in *ROCK1*, *ROCK2* knockdown experiments with D4G (shown in response to QR1.12 where EC₅₀ was calculated). The EC 50 reported in Fig S8 in the revision are from WT-SH3 IMCD (57 mcM). This was the basis of our initial low dose in vivo testing using 20 mg/Kg (57mcM) (shown in response to reviewer QR2.2). This initial dose lacked in vivo efficacy against TIF in UUO as shown, and we thus used a higher dose 50 mg/Kg (as described above and shown in Fig 8/S8). We accept that small animal models pharmacokinetic/pharmacodynamic studies have not been done yet, but we plan to do these steps after this initial proof-of-principle work. SAR studies are also ongoing with the goal to improve potency, improve solubility and minimize any toxicity, before subsequent PK/PD studies.

Reviewer #4 (Remarks to the Author):

The authors elucidated the interaction with Rock and ASD2 domain in Shroom3, using mice with ablation of ASD2 domain. They also created the compound that interrupts the interaction with Rock and ASD2, which might be a promising drug for kidney disease. However, there are some points to be addressed.

[Major]

Q.R4.1. The dataset used by the authors in Figure 1A includes the IRI and UO model at different time points. Therefore, it is necessary to clarify which model and timepoint the authors analyzed.

Reply: We appreciate the reviewer's suggestion and apologize for the lack of clarity in the figure 1A. The dot plot is a depiction of gene expression from cells of healthy (baseline), unilateral-IRI and UO kidneys including all timepoints used in the cited study. We have now added this information to the revised manuscript and Figure legend.

Q.R4.2 In addition, according to the result from the website (<https://humphreyslab.com/SingleCell/>), the expression patterns of Shroom3 and Rock are different, suggesting that Rock expression is likely regulated by other molecules.

Reply: We thank the reviewer for this insightful observation. In Figure 1A, our primary goal was to illustrate that Shroom3-Rock protein-protein interaction is possible in the highlighted cell types since both Shroom3 and Rock mRNAs are expressed in those specific cell types (including in injured proximal tubular cells and Fibroblasts).

However, we agree with the reviewer that Shroom3 and ROCK are likely regulated by distinct upstream signals, and their expression may diverge across other cell populations or states. For instance, both Rock isoforms have multiple promoter binding sites for KLF family transcription factors, and experimental data show that SP6 and SP1 (KLF family members) can increase Rock1 levels^{32, 33}. During cell division/mitosis, proteins required for mitosis (including Rock1, Rock2 needed for cytokinesis) are upregulated by multiple transcription factors including E2F/SP1, FOXM1, and NF-Y^{34, 35}. Similarly TFAP2C (AP-2 family) increases expression of both Rock paralogs³⁶. To date, the roles of these TFs in Shroom3 regulation have not yet been reported.

On the other hand, multiple groups including ours, have reported Shroom3 to be upregulated by pro-fibrogenic signaling pathways including Wnt/Ctnnb1 and Tgf β ^{37, 38, 39, 40}. Notably, both Rock1 and Rock2 promoters also have SMAD4 binding sequences, and experimental data from multiple cell lines have shown increased Rock expression with TGFB1 stimulation^{41, 42}. Rock1 is also regulated by Tgf β in a Smad-independent manner^{43, 44}. Hence, TGFB1 is a potential signal, which in the setting of TIF could simultaneously upregulate both Shroom3 and Rock1/Rock2. To reflect the reviewer's important point, we have included the following in the revised manuscript.

"Rock1 and Rock2 are regulated by distinct set of transcription factors during cell proliferation^{50, 51}, but during TIF states may be regulated by TGFB1-signaling^{52, 53}, which also upregulates Shroom3^{54, 55, 56}."

Q.R4.3. Figure 1A and supplemental Figure 1a show injured PT and fibroblast expressed Shroom3, Rock1 and Rock2 higher than other cells, however, the expression patterns were quite different. In Figure 1A, Fibroblast expressed these three genes higher, and injured PT had less expression of shroom3. On the other hand, in supplemental Figure 1A, fibroblast expressed Rock1 and Rock2, but did not express Shroom3 while injured PT expressed Shroom3 and Rock2.

Reply: As the reviewer correctly observed, Figure 1A is a representation of cell-type specific gene expression of *Shroom3*, *Rock1*, *Rock2* in UO/IRI kidneys. Hence, it represents the average expression of genes in each cell type regardless of condition, or time points of the injuries.

Figure S1a data was based on cell-specific gene expressions at different time points (Days 7, 30) of Unilateral IRI kidney compared to the sham control kidney performed by our collaborator/co-authors⁴⁵. Notably, this is a single cell RNA-seq (scRNA-seq) dataset. We agree with the reviewer that in this dataset shown in Fig S1A, Pdgfrb+ myofibroblast cells exhibit low expression level of Shroom3 at the IRI timepoints. In this dataset generated by our collaborator using a fibrotic mouse model, fibroblasts and myofibroblasts could not be separated, likely accounting for the lower overall expression level of Shroom3 due to the inclusion of myofibroblasts along with fibroblasts⁴⁵. This distinction was apparent when examining data from Li et al 2022 (a single nuclear RNA-seq i.e. snRNA-seq), fibroblast and myofibroblast were separately annotated (shown below for the reviewer Fig R23). As shown, Fibroblasts in UUO kidneys express Shroom3 from Day2-Day14 (now included in *Revised Figure S1a*), while myofibroblasts display markedly lower expression level of Shroom3.

Expression of Shroom3 and Rock1/2 in fibroblasts of UUO kidneys from Li et al., 2022

We agree with the reviewer's observation that *Rock1* and *Rock2* are differentially expressed spatially and temporally in each model of TIF. Either/both paralogs could be involved in the profibrotic signaling in the context of excess Shroom3. To specifically investigate the individual roles of the Rock paralogs in the pro-fibrotic signaling in vivo, we obtained *Rock1*^{ff}⁴⁶ and *Rock2*^{ff}^{47, 48} mice from Dr Anne Eichmann (collaborator, coauthor) to be crossed with inducible Shroom3 overexpression mice. These studies will dissect paralog-specific contributions to Shroom3-mediated fibrotic signaling and are planned future directions.

We also want to point out that the lack of expression of *Rock1* in injured tubular cells in later phases of IRI (shown in Fig S1b), was not observed in Li et al's dataset. As shown below [Fig R24], from Li et al's dataset *Rock1* expression was also present in iPTs at day 28 (in addition to *Rock2*). This contrasts with Xu et al.'s scRNA-seq dataset and may reflect methodological differences (snRNA-seq vs. scRNA-seq). We therefore have used two datasets and which include two injury models, and two modalities of single cell transcriptome assessment to identify putative cells of origin for Shroom3-Rock interaction for this manuscript.

Q. R4.4. You adopted UUO model in the experiment of Figure 4 and 6 while you used AAN model in Figure 5.
 Reply: We thank the reviewer for this important point. We have adopted both AAN (Figure 3) and UUO (Figure 4) models in global WT-Sh3 and ASD2Δ-Sh3 mice.

In response to the question raised by the reviewer, we now performed UJO model with tubular specific, Pax8-WTSh3/ASD2ΔSh3 mice. As shown in Figure 5 and S5 of the revision, Pax8-rtTA/ ASD2ΔSh3 mice showed benefit in TIF outcomes vs Pax8-rtTA/ WT-Sh3 (n=4 vs 6). These data are summarized below as R25.

Hence, we now conclude that both in the global- or tubular-specific overexpression mice, impairing Shroom3-Rock interaction by overexpressing the ASD2 deletion provided benefit in both AAN- & UJO- models of TIF. These data support the model-independent beneficial effect of impairing Shroom3-Rock interaction, and support the strategy of using either model to test the efficacy of our P2Is.

In UJO model, kidney has less expression of Shroom3 until day 7 according to Humphrey's website (see attached file). Therefore, you could not evaluate how ASD2delta-Sh3 in fibroblast affected kidney in UJO model.

Reply: We appreciate this question raised by the reviewer. We want to first note that in our model, *Shroom3* expression itself is driven by DOX induction, and not dependent on regulation by endogenous stimuli. However,

we admit *Rock1* and *Rock2* expression would be temporally and spatially regulated by endogenous transcription factors and signals.

In response to this important question raised by the reviewer, we show here data from the Li et al, 2022 [Fig R26], SnRNA dataset that *Shroom3*, *Rock1* and *Rock2* are all expressed as early as day 2 post-UUO and remain detectable through day 6 after UUO. Specifically, injured and repairing tubular cells express these genes as observed in Fig R26 (lower panel) and in S1a. Furthermore, as shown in response to QR4.3, fibroblasts also express *Shroom3* and *Rock1*, *Rock2* in early stages of UUO. Hence, for these reasons we used the UUO model as a TIF model in this work.

QR4.5. Please explain the reason that you adopt UUO model, not AAN model in Figure 6.

Reply: We appreciate this concern raised by the reviewer. We now performed UUO model (in addition to the previous AAN model) with Pax8-WTSh3/ASD2 Δ Sh3 mice and found mitigation of fibrosis in the ASD2 Δ Sh3 mice similar to what was observed in the AAN model (data shown above in response to QR4.4). This indicates that readouts from either of the TIF models could test the antifibrotic effect of ASD2 domain deletion, if present.

For Figure 6, in fibroblast-specific overexpression mice, UUO model did not reveal any difference in fibrosis progression WTSh3 vs. ASD2 Δ Sh3. Hence, we did not proceed with a second model of fibrosis-AAN with the fibroblast-specific overexpression mice.

QR4.6. Regarding Figure 1G and 1I, the authors concluded that tubular cells from ASD2 Δ -Sh3 mice had less ability to migrate and proliferate. And ASD Δ -Sh3 was protective against kidney injury. However, in previous reports (J Clin Invest. 2019;129(12):5501–5517, Nat Rev Nephrol 16, 269–288 (2020)), recovery from injury requires proliferative ability in tubular cells, which is not consistent with your result.

Reply: We thank the reviewer for this important and interesting point and agree that we observed that ASD Δ -Sh3 overexpression inhibited tubular cell proliferation. We believe this is likely due to the role of Rho-kinases in spindle formation and cytokinesis during cell division, and inhibiting activation of ROCKs via ASD2 may underlie this observation.

We agree that this could potentially worsen recovery from acute kidney injury, and admit we did not evaluate acute injury or test recovery from AKI here. Our focus in this work was on the later TIF outcome after remodeling/progressive injury. To test the role of ASD Δ -Sh3 overexpression in acute injury, would require sequential time course experiments which we have not performed here.

However, we do want to note that we continuously induced ASD2 mutant expression in UJO and AAN models, throughout the acute and post-injury phases and still observed a benefit in TIF phenotype, suggesting that the acute benefit from Rock activation during recovery may ultimately not be of greater importance in TIF outcome. We have discussed this in the last paragraph of Discussion listing the limitations of the study.

“First, in contra-distinction to multiple reported adverse associations of Shroom3-SNPs with CKD, a potential protective role for a linked intronic Shroom3-SNP in AKI in humans was reported recently⁴⁹, and increased AKI also occurred in heterozygous global Shroom3-knockout mice⁵⁰. Hence, delineating the role of increased Shroom3-Rock interaction during acute injury, recovery from AKI vs TIF/CKD will avoid any potential harm during therapeutic use. Although we did not conduct such time course experiments here, we note that we continuously induced ASD2 Δ -Sh3 throughout the injury phase and still observed benefit in TIF.”

QR4.7. In Figure 8d, please add the data on Vimentin, Col1a1, Fsp1, Mmp2 as shown in Figure 4. Reply: We appreciate the reviewer’s suggestion and have included more genes (including Vim, Col1a1, Fsp1 and Mmp2) to the mRNA expression profile (*Revised Figure 8n*) and improved the data with more animals (n=6 each condition). This data is shown in Fig R27. This data have improved the rigor of our conclusions regarding the benefit of D4G in TIF in vivo.

R27

QR4.8. To confirm that D4G is effective for fibrosis, the authors should perform an additional experiment with AAN model.

Reply: We thank the reviewer for the suggestion. While our invitro data shows clear efficacy with minimal toxicity of P2Is, we do acknowledge that a second in vivo model (AAN) would provide greater reassurance of the effect of P2Is on TIF in vivo.

In data included here, we now show that interfering with Shroom3-Rock interaction in states of global or tubular specific Shroom3 excess using ASD2 Δ -Sh3 mice was similarly protective for TIF from either AAN or UJO models. Our data shows clear impact on Rock signaling mechanisms, profibrotic TGFB1 or Wnt signals, proinflammatory genes, as well as preservation of genes of tubular homeostasis in ASD2 Δ -Sh3 mice. Further, in the in vivo experiments with D4G, we have now expanded the sample size in our UJO experiment in the revision, and included robust protein level data from this model which shows similar alterations in Rock signaling pathway, profibrotic and proinflammatory genes, as well as preservation of tubular homeostasis mechanisms with D4G (analogous to observations with ASD2 Δ -Sh3) [Revised Figures 8/S8; and response to QR1.13]. These

data in the revision provide proof-of-concept for feasibility and efficacy of our novel Shroom3-Rock P2Is as antifibrotic therapy in conditions of Shroom3 excess.

Despite the requirement of additional time and resources for a potential AAN experiment (beyond the 3-month timeline provided for the revision), we can provide this if felt to be critical by the reviewer/editors for publication. We have also acknowledged the lack of a second model as a limitation in the revision as follows.

“Finally, although the beneficial role of inhibiting Shroom3-Rock interaction on TIF was not model-specific, and we demonstrated that D4G has *in vivo* efficacy as a P2I in a UUO model of TIF, evaluation of P2Is in other *in vivo* models including pharmacokinetics/pharmacodynamic profiles, human organoid testing and, longer term toxicity studies, are all essential. All these studies are critical for translational potential and the ultimate development of refined Shroom3-Rock P2Is to treat CKD related to Shroom3-excess & Shroom3 SNPs.”

QR4.9. Please clarify in which area of kidney and who evaluated fibrotic score.

Reply: We thank the reviewer for the query. We quantified fibrosis scores from the cortical area of the kidney. For evaluating cortical fibrosis, slides were reviewed by Pathologist from Yale renal Pathology (S.P.) and digital images were evaluated by G. B. using ImageJ (published Macro). The fibrosis scores estimated using ImageJ were similar to the Pathologist's scoring (both blinded).

[Minor]

QR4.10 Another research team has reported that a different SNP (rs142647267) was significantly associated with a decline in eGFR in a study of the Shroom3 locus. Please discuss these other studies in detail.

Reply: We appreciate this observation by the reviewer. Indeed, multiple studies have identified intronic SNPs in Shroom3 as associated with eGFR traits (eGFR-creatinine; eGFR-Cystatin C; prevalent CKD and recently AKI). The SNP mentioned by the reviewer has been identified in a Japanese population study. We have now included sentences in the discussion to allude to these many studies and call out these SNP variants in the text.

“For instance, intronic variants in the Shroom3 gene have been repeatedly identified as associated with CKD^{51, 52, 53}.”

“Independent work has also supported a regulatory role for intronic Shroom3 variants (rs17219731, rs142647267, rs4859682) in renal epithelial cells^{5, 54}, and more specifically in proximal tubular cells⁵².”

“..a potential protective role for a linked intronic Shroom3-SNP in AKI in humans was reported recently⁴⁹,...”

Reply: We believe these multiple studies repeatedly support the relevance of Shroom3 protein in CKD and the rationale of our experimental work in this manuscript.

QR4.11. There are many abbreviations in the text without initial spell-out, making it difficult to read, and this needs to be resolved.

Reply: As per the reviewer's suggestion, we revised the manuscript for uniformity in abbreviations in the running text.

We greatly appreciate the expert reviews and feedback by the reviewers and editors which have improved our manuscript considerably.

Reviewer References:

1. Dickson HM, Wilbur A, Reinke AA, Young MA, Vojtek AB. Targeted inhibition of the Shroom3-Rho kinase protein-protein interaction circumvents Nogo66 to promote axon outgrowth. *BMC Neurosci* **16**, 34 (2015).
2. Kistler AD, Altintas MM, Reiser J. Podocyte GTPases regulate kidney filter dynamics. *Kidney Int* **81**, 1053-1055 (2012).
3. Wang L, *et al.* Mechanisms of the proteinuria induced by Rho GTPases. *Kidney Int* **81**, 1075-1085 (2012).
4. Blattner SM, *et al.* Divergent functions of the Rho GTPases Rac1 and Cdc42 in podocyte injury. *Kidney Int* **84**, 920-930 (2013).
5. Prokop JW, *et al.* Characterization of Coding/Noncoding Variants for SHROOM3 in Patients with CKD. *J Am Soc Nephrol*, (2018).
6. Faul C, *et al.* The actin cytoskeleton of kidney podocytes is a direct target of the antiproteinuric effect of cyclosporine A. *Nature medicine* **14**, 931-938 (2008).
7. Asanuma K, Yanagida-Asanuma E, Faul C, Tomino Y, Kim K, Mundel P. Synaptopodin orchestrates actin organization and cell motility via regulation of RhoA signalling. *Nature cell biology* **8**, 485-491 (2006).
8. Scott RP, *et al.* Podocyte-specific loss of Cdc42 leads to congenital nephropathy. *J Am Soc Nephrol* **23**, 1149-1154 (2012).
9. Matoba K, *et al.* ROCK2-induced metabolic rewiring in diabetic podocytopathy. *Commun Biol* **5**, 341 (2022).
10. Matoba K, *et al.* Deletion of podocyte Rho-associated, coiled-coil-containing protein kinase 2 protects mice from focal segmental glomerulosclerosis. *Commun Biol* **7**, 402 (2024).
11. Zhou L, *et al.* Amelioration of albuminuria in ROCK1 knockout mice with streptozotocin-induced diabetic kidney disease. *American journal of nephrology* **34**, 468-475 (2011).
12. Peng F, *et al.* RhoA/Rho-kinase contribute to the pathogenesis of diabetic renal disease. *Diabetes* **57**, 1683-1692 (2008).
13. Komers R. Rho kinase inhibition in diabetic kidney disease. *Br J Clin Pharmacol* **76**, 551-559 (2013).

14. Gojo A, *et al.* The Rho-kinase inhibitor, fasudil, attenuates diabetic nephropathy in streptozotocin-induced diabetic rats. *Eur J Pharmacol* **568**, 242-247 (2007).
15. Komers R, *et al.* Rho kinase inhibition protects kidneys from diabetic nephropathy without reducing blood pressure. *Kidney Int* **79**, 432-442 (2011).
16. Kolavennu V, Zeng L, Peng H, Wang Y, Danesh FR. Targeting of RhoA/ROCK signaling ameliorates progression of diabetic nephropathy independent of glucose control. *Diabetes* **57**, 714-723 (2008).
17. Wei C, *et al.* SHROOM3-FYN Interaction Regulates Nephrin Phosphorylation and Affects Albuminuria in Allografts. *J Am Soc Nephrol* **29**, 2641-2657 (2018).
18. Verma R, *et al.* Fyn binds to and phosphorylates the kidney slit diaphragm component Nephrin. *The Journal of biological chemistry* **278**, 20716-20723 (2003).
19. Verma R, Kovari I, Soofi A, Nihalani D, Patrie K, Holzman LB. Nephrin ectodomain engagement results in Src kinase activation, nephrin phosphorylation, Nck recruitment, and actin polymerization. *The Journal of clinical investigation* **116**, 1346-1359 (2006).
20. Audard V, *et al.* Occurrence of minimal change nephrotic syndrome in classical Hodgkin lymphoma is closely related to the induction of c-mip in Hodgkin-Reed Sternberg cells and podocytes. *Blood* **115**, 3756-3762 (2010).
21. Sun KH, Chang Y, Reed NI, Sheppard D. alpha-Smooth muscle actin is an inconsistent marker of fibroblasts responsible for force-dependent TGFbeta activation or collagen production across multiple models of organ fibrosis. *Am J Physiol Lung Cell Mol Physiol* **310**, L824-836 (2016).
22. Zhao W, Wang X, Sun KH, Zhou L. alpha-smooth muscle actin is not a marker of fibrogenic cell activity in skeletal muscle fibrosis. *PLoS One* **13**, e0191031 (2018).
23. Miyazono K, Heldin CH. Latent forms of TGF-beta: molecular structure and mechanisms of activation. *Ciba Found Symp* **157**, 81-89; discussion 89-92 (1991).
24. Miyazono K, Olofsson A, Colosetti P, Heldin CH. A role of the latent TGF-beta 1-binding protein in the assembly and secretion of TGF-beta 1. *EMBO J* **10**, 1091-1101 (1991).
25. Sun Z, *et al.* Multiscale genetic architecture of donor-recipient differences reveals intronic LIMS1 mismatches associated with kidney transplant survival. *The Journal of clinical investigation* **133**, (2023).
26. Wong MG, Panchapakesan U, Qi W, Silva DG, Chen XM, Pollock CA. Cation-independent mannose 6-phosphate receptor inhibitor (PXS25) inhibits fibrosis in human proximal tubular

cells by inhibiting conversion of latent to active TGF-beta1. *Am J Physiol Renal Physiol* **301**, F84-93 (2011).

27. Das BC, *et al.* Boron chemicals in diagnosis and therapeutics. *Future Med Chem* **5**, 653-676 (2013).
28. Fernandes GFS, Denny WA, Dos Santos JL. Boron in drug design: Recent advances in the development of new therapeutic agents. *Eur J Med Chem* **179**, 791-804 (2019).
29. Zhong F, *et al.* Tyro3 is a podocyte protective factor in glomerular disease. *JCI Insight* **3**, (2018).
30. Lovering F, Bikker J, Humblet C. Escape from flatland: increasing saturation as an approach to improving clinical success. *J Med Chem* **52**, 6752-6756 (2009).
31. Maurer TS, Edwards M, Hepworth D, Verhoest P, Allerton CMN. Designing small molecules for therapeutic success: A contemporary perspective. *Drug Discov Today* **27**, 538-546 (2022).
32. Yanuarieska RD, *et al.* Sp6 regulation of Rock1 promoter activity in dental epithelial cells. *J Med Invest* **61**, 306-317 (2014).
33. Elliott EG, Ettinger AS, Leaderer BP, Bracken MB, Deziel NC. A systematic evaluation of chemicals in hydraulic-fracturing fluids and wastewater for reproductive and developmental toxicity. *J Expo Sci Environ Epidemiol* **27**, 90-99 (2017).
34. Zanutto E, Shah ZH, Jacobs HT. The bidirectional promoter of two genes for the mitochondrial translational apparatus in mouse is regulated by an array of CCAAT boxes interacting with the transcription factor NF-Y. *Nucleic Acids Res* **35**, 664-677 (2007).
35. Wang IC, *et al.* Forkhead box M1 regulates the transcriptional network of genes essential for mitotic progression and genes encoding the SCF (Skp2-Cks1) ubiquitin ligase. *Mol Cell Biol* **25**, 10875-10894 (2005).
36. Farooqi AA, *et al.* Regulation of ROCK1/2 by long non-coding RNAs and circular RNAs in different cancer types. *Oncol Lett* **23**, 159 (2022).
37. Menon MC, *et al.* Intronic locus determines SHROOM3 expression and potentiates renal allograft fibrosis. *The Journal of clinical investigation*, (2014).
38. Durbin MD, O'Kane J, Lorentz S, Firulli AB, Ware SM. SHROOM3 is downstream of the planar cell polarity pathway and loss-of-function results in congenital heart defects. *Dev Biol* **464**, 124-136 (2020).

39. Yoon J, Sun J, Lee M, Hwang YS, Daar IO. Wnt4 and ephrinB2 instruct apical constriction via Dishevelled and non-canonical signaling. *Nat Commun* **14**, 337 (2023).
40. Li A, *et al.* Shroom3, a Gene Associated with CKD, Modulates Epithelial Recovery after AKI. *Kidney360* **3**, 51-62 (2022).
41. Ji H, *et al.* Rho/Rock cross-talks with transforming growth factor-beta/Smad pathway participates in lung fibroblast-myofibroblast differentiation. *Biomed Rep* **2**, 787-792 (2014).
42. Igarashi N, Honjo M, Aihara M. mTOR inhibitors potentially reduce TGF-beta2-induced fibrogenic changes in trabecular meshwork cells. *Sci Rep* **11**, 14111 (2021).
43. Papadimitriou E, Kardassis D, Moustakas A, Stournaras C. TGFbeta-induced early activation of the small GTPase RhoA is Smad2/3-independent and involves Src and the guanine nucleotide exchange factor Vav2. *Cell Physiol Biochem* **28**, 229-238 (2011).
44. Wei YH, Liao SL, Wang SH, Wang CC, Yang CH. Simvastatin and ROCK Inhibitor Y-27632 Inhibit Myofibroblast Differentiation of Graves' Ophthalmopathy-Derived Orbital Fibroblasts via RhoA-Mediated ERK and p38 Signaling Pathways. *Front Endocrinol (Lausanne)* **11**, 607968 (2020).
45. Xu L, Guo J, Moledina DG, Cantley LG. Immune-mediated tubule atrophy promotes acute kidney injury to chronic kidney disease transition. *Nat Commun* **13**, 4892 (2022).
46. Huang H, *et al.* Rho-kinase regulates energy balance by targeting hypothalamic leptin receptor signaling. *Nat Neurosci* **15**, 1391-1398 (2012).
47. Okamoto R, *et al.* FHL2 prevents cardiac hypertrophy in mice with cardiac-specific deletion of ROCK2. *FASEB J* **27**, 1439-1449 (2013).
48. Zarkada G, *et al.* Chylomicrons Regulate Lacteal Permeability and Intestinal Lipid Absorption. *Circ Res* **133**, 333-349 (2023).
49. Siew ED, *et al.* Genome-wide association study of hospitalized patients and acute kidney injury. *Kidney Int* **106**, 291-301 (2024).
50. Lawlor A, *et al.* Minimal Kidney Disease Phenotype in Shroom3 Heterozygous Null Mice. *Can J Kidney Health Dis* **10**, 20543581231165716 (2023).
51. Kottgen A, *et al.* New loci associated with kidney function and chronic kidney disease. *Nature genetics* **42**, 376-384 (2010).
52. Loeb GB, *et al.* Variants in tubule epithelial regulatory elements mediate most heritable differences in human kidney function. *Nature genetics*, (2024).

53. Ghasemi-Semeskandeh D, *et al.* Clinical and Metabolic Signatures of FAM47E-SHROOM3 Haplotypes in a General Population Sample. *Kidney Int Rep* **10**, 1495-1508 (2025).
54. Matsuura R, *et al.* SHROOM3, the gene associated with chronic kidney disease, affects the podocyte structure. *Sci Rep* **10**, 21103 (2020).

To our Editor and Reviewers,

We would like to thank you for the thorough evaluation of our manuscript and providing us with constructive feedback. We greatly appreciate the opportunity to revise and resubmit our work.

All comments from the reviewer have been addressed in the accompanying point-by-point response document. Revisions to the manuscript have been clearly highlighted as per journal guidelines.

We have ensured full compliance with all editorial policies and requirements, including:

- Submission of the Editorial Policy Checklist and attached to the Reporting Summary (Similar to first revision).
- Updated Data Availability section with deposition of datasets in appropriate public repositories (Similar to first revision).
- Providing a comprehensive Source Data files for all main figures as ppt and tables in Excel format (Similar to first revision).
- Edited data in the Figure panels and Supplementary information as suggested by the reviewer #4.
- Consideration and appropriate reporting of sex and gender in line with SAGER guidelines (Similar to first revision).

We enclose in the revised submission (i) a marked copy (yellow highlights for changes in revision), (ii) a clean copy of the manuscript file, (iii) Supplementary information: Supplementary figures/legends (Marked and Unmarked copies); Reporting summary and Editorial Policy Checklist and (iv) Supplementary datasets 1-4.

We confirm that there have been no changes to the author's list.

We sincerely hope that the revised version addresses all the concerns raised and meets the expectations of the reviewer and editorial board.

Point by point response to all reviewers' comments are listed below:

Reviewer #1 (Remarks to the Author):

All my comments have been addressed and the manuscript is suitable for publication

We appreciate the reviewer's time and effort and favorable review of our work.

Reviewer #2 (Remarks to the Author):

We appreciate the reviewer's time and effort and favorable review of our work.

Reviewer #3 (Remarks to the Author):

The authors have answered all of my comments. It is Ok for me to accept it now.

We appreciate the reviewer's time and effort and favorable review of our work.

Reviewer #4 (Remarks to the Author):

Thank you for your special efforts to revise this manuscript. I believe it has improved considerably. However, to preserve the robustness of the work in Nat Commun, the following points still require attention; I would appreciate your continued consideration.

We appreciate the reviewer's comments and detailed suggestions which have helped us improve our manuscript, and appreciate the favorable review of our work.

1. Contradiction in expression patterns in Figure 1A/S1a

→ From the dot plots, it appears that at most ~10% of cells within the proximal tubule and myofibroblast/fibroblast populations express the genes in question, and likely at quite low levels. If violin plots are shown, it will probably become clear that there is essentially no expression. It is not reasonable to discuss genes that are virtually not expressed in the single-cell data, and this likely contributes to the apparent discrepancy between Figure 1A and S1a. Even without the single-cell data the manuscript is sufficiently interesting, so it would be better to remove these data.

-We believe the reviewer is referring to Figures 1a, S1a and S1b (all single cell or single nuclear data related figures), and appreciate the reviewer pointing this issue of *Shroom3*, *Rock1* and *Rock2* expression levels. Overall, we appreciate this valuable suggestion from the reviewer.

-To respond to this question raised by the reviewer we generated violin plots from our co-author's publication ¹. As shown below, iPTs during the fibrosis stage show increased expression of *Shroom3*, *Rock1* and *Rock2*, supporting mechanistic experiments in tubular cell lines. Myofibroblast cluster (which includes myofibroblasts and fibroblasts together) robustly express *Rock1* and *Rock2*, but inconsistently express *Shroom3* in this IRI dataset. As observed in the Li et al dataset, myofibroblasts do have lower expression of *Shroom3*.

-We also evaluated recent independent data from a multispecies-integrated Single-Cell Kidney Atlas, by Klotzner, Susztak et al ², which evaluated ScRNA data from healthy and diseased murine kidneys from multiple publications (total cells after QC =447499 (Healthy =64948 ,diseased =382551) [located at <https://susztaklab.com/SISKA/mouse>]. Here the proportion of iPT cells expressing *Shroom3* ranged from 33 to 45%, and in stromal cells (which includes fibroblasts/ myofibroblasts) *Shroom3* expression ranged from 52 to 53%. These corresponding data for *Rock1* were 33-36% in iPTs and 31-32% in stromal cells, and for *Rock2* were 41-50% in iPTs and 47-56% in stromal cells, respectively. These again reflect variability in cell-type specific expression of these genes in single cell transcriptomic data across different publications, but support their expression in iPTs and stromal cells. We have now included this recent reference in the revision.

Hence, these data and others suggest that there is sufficient expression of these genes in PTs and fibroblast cells providing rationale for our sequential mechanistic experiments, which then allowed us to infer the central role of *Shroom3*-*Rock* interaction in iPTs (vs fibroblasts) during TIF. As suggested by the reviewer, since these derivative data from prior publications are not essential for the manuscript, and to avoid inconsistency between figures, we have now removed these panels from the revised manuscript. We have redrafted the paragraph referring to these data as follows.

“We previously reported that tubular-specific *Shroom3* knockdown alone could mitigate TIF in a UO model ⁸. Recent functional genomics studies evaluating CKD-associated-SNPs using single cell-ATACseq also concluded that single-cell, cis-eQTL effects of *Shroom3* intronic variants are identifiable in proximal tubular cells (PTs) in human CKD²⁴. These pointed to PTs as key players for TIF from excess *Shroom3*. We also considered that fibroblasts which are key cells for matrix production during TIF, express *Shroom3* and show high *Rock1*, *Rock2* expression in published data from murine TIF models^{25, 26, 27, 28}. Podocytes show high expression of *Shroom3* at rest and injury but are not expected to play key roles in these TIF models after PT cell injury^{25, 26, 27}. Hence, we focused on tubular and fibroblast cell lines which were potential sites of *Shroom3* interaction with *Rock1* and/or *Rock2* during TIF for subsequent mechanistic experiments.”

2. Insufficient explanation for the contradiction that *ASD2Δ*-*Sh3* suppresses proliferation yet confers renoprotection

→ Prior reports (e.g., JCI 2019) indicate that tubular cell proliferation is required for renal recovery, but this apparent contradiction has not yet been addressed.

We appreciate this point raised by the reviewer. We have now added a paragraph in the discussion to detail potential explanations behind the dichotomous association seen in epidemiologic studies of enhancer Shroom3 SNPs (ie with more Shroom3-Rock interaction) with *improved AKI but consistently increased CKD* risk. We believe that Rock-activation mediated Ccl2-excess from iPTs sets up cross-talk to with immune cells (via Ccr2) which in turn activate resident stromal cells setting up “fibro-inflammation” and CKD progression. Ccl2 was consistently observed to be inhibited significantly during genetic or pharmacologic inhibition of ASD2-Rock interaction in vitro and in vivo in our data. This is also suggested from the multiple data showing consistent benefit of Rock-inhibitors in AKI^{3,4}. We have also accepted as limitation that we have not currently done time-course studies where we induced Shroom3-Rock interaction inhibition (genetically or pharmacologically) during AKI, during AKI recovery or during TIF stage alone.

To summarize this in the manuscript, we have added the following to the discussion.

“It is also important to discuss implications of our work for acute kidney injury (AKI). We note that in contrast to multiple reported adverse associations of Shroom3-SNPs with CKD, a potential protective role for a linked intronic Shroom3-SNP in AKI in humans was reported recently⁶⁵, and increased AKI also occurred in heterozygous global Shroom3-knockout mice³¹. Facilitation of Rho-kinases by SHROOM3 could aid in spindle formation and cytokinesis during cell division, and inhibiting activation of ROCKs via ASD2 could reduce PT proliferation after injury^{66, 67}, consistent with our observations in vitro in IMCD lines. Since cell proliferation is required for tubular repair⁶⁸, delineating the role of increased Shroom3-Rock interaction during AKI, recovery from AKI vs TIF/CKD is needed to avoid any potential harm during therapeutic use of P2Is. We acknowledge that we did not conduct such time course experiments. However, we note that ASD2Δ-Sh3 was continuously expressed in our models throughout the AKI phase and we still observed benefit in TIF. Next, in spite of this potential adverse effect of inhibiting ROCKs during AKI, Rock-inhibitors have consistently shown benefit in models of AKI^{66, 69}. A possible explanation is the observation that inhibition of Shroom3-Rock interaction in tubular cells alone markedly inhibited *Ccl2*, a chemokine signal known to be facilitated by rho-kinase activation^{32, 70, 71}. *Ccl2* mediates immune cell recruitment via its receptor *Ccr2* and is a potent facilitator of tubular-inflammatory cell crosstalk after injury²⁸. Hence, in spite of potential deleterious effects on tubular cell proliferation during AKI, excess Shroom3-Rock interaction in iPTs overexpressing *SHROOM3* would enhance crosstalk between iPTs *and* interstitial - inflammatory and/or stromal -cells, culminating in increased “fibro-inflammation” and progressive CKD.”

3. Mismatch between Shroom3 and ROCK expression patterns (Humphreys dataset)
→ Although the rebuttal compared multiple datasets to explain this, the main text does not explicitly state it. Please incorporate this discussion into the Discussion section.

We apologize for not including these lines from the rebuttal statement in the manuscript. We have now included the following in the limitations section of the discussion as follows to address the Reviewers initial concern.

“From the multiple single cell expression datasets^{24, 25, 27 26}, it is apparent to us that cell-type specific expression patterns of each Rock paralog and Shroom3 are variable and sometimes mismatched between datasets. This variability may have resulted from technical variability between datasets (single cell vs single nuclear, etc). To minimize the impact of this in our work, we mined multiple datasets which included multiple injury models and modalities of single cell transcriptome assessment to provide cell-type rationale for our mechanistic studies.”

4. Other

→ Because numerous new references were added in the revision, many citation numbers have changed. Please carefully check that numbering and cross-references are consistent.

We apologize for this oversight. We cross-checked the references and ran the reference library for the Response document separate from the main manuscript document.

1. Xu L, Guo J, Moledina DG, Cantley LG. Immune-mediated tubule atrophy promotes acute kidney injury to chronic kidney disease transition. *Nat Commun* **13**, 4892 (2022).
2. Klotzer KA, *et al.* Analysis of individual patient pathway coordination in a cross-species single-cell kidney atlas. *Nature genetics* **57**, 1922-1934 (2025).
3. Prakash J, *et al.* Inhibition of renal rho kinase attenuates ischemia/reperfusion-induced injury. *J Am Soc Nephrol* **19**, 2086-2097 (2008).
4. Liang H, *et al.* CXCL16/ROCK1 signaling pathway exacerbates acute kidney injury induced by ischemia-reperfusion. *Biomed Pharmacother* **98**, 347-356 (2018).

REVIEWERS' COMMENTS

We appreciate the reviewers detailed and favorable responses.

Reviewer #4 (Remarks to the Author):

With regard to the reviewer's concerns, the authors have addressed the issues as follows. As a result, the manuscript has become logically coherent, more readily comprehensible, and in my view has been further polished.

Contradiction in expression patterns in Figure 1A/S1a

→ The authors re-examined the data using violin plots, removed non-essential data in accordance with the reviewer's recommendation, and revised the relevant paragraph in the text.

Contradiction that ASD2Δ-Sh3 suppresses proliferation yet confers renoprotection

→ A new paragraph has been added to the Discussion, elaborating on the apparent discrepancy within the context of AKI versus CKD, with additional explanation involving Ccl2/ROCK signaling.

Mismatch between Shroom3 and ROCK expression patterns (Humphreys dataset)

→ The variability across datasets has now been explicitly acknowledged and incorporated into the Limitations section.

Inconsistencies in reference numbering and cross-references

→ The authors have stated that they have carefully reviewed and corrected these inconsistencies.

-We appreciate the positive evaluation of our revisions by the reviewer.

Minor points:

- As noted, it would be appropriate to spell out iPTs (injured proximal tubules) at its first appearance in the main text. In fact, the rebuttal document already provides this clarification as "iPTs (injured proximal tubular cells)."

Reply: We thank the reviewer for the correction. We added the expansion for iPTs at the first occurrence.

- FBDM-Sh3 (Fyn-binding domain mutant Shroom3): This abbreviation should be defined at its first occurrence.

Reply: We thank the reviewer for pointing out this. We have now expanded the abbreviation at the first occurrence.

- P2Is (Shroom3-Rock interaction inhibitors): Although the definition is provided in the text, the introduction of this abbreviation is somewhat abrupt. It would be clearer and more reader-friendly to introduce it more explicitly at first mention, for example: “we developed novel small molecule inhibitors of Shroom3-Rock interaction (P2Is).”

Reply: We appreciate the reviewer’s suggestion and have incorporated it.

- Several references appear to be incomplete. In particular, references 7, 18, 24, 55, and 79 seem to be missing page numbers.

We appreciate these issues that were pointed out. We have corrected and highlighted in marked copy.

Reference 7 & 11 – we have included page numbers for these references

Reference 18 and 11 were duplicated. Hence in the revision Ref 18 has been deleted, and all references below 18 have been pushed up by 1.

References 24, 55, 79 have now become 23, 54 and 78, respectively. We have added in page numbers obtained from pubmed and the journal websites.

- Additional inconsistencies may also exist, for example in references 25–27 (murine TIF models) and 65–67 (recently added AKI-related studies). These should be checked carefully to ensure that full bibliographic details, including page ranges, are provided.

We checked these references carefully. The endnote program picked these PMIDs from pubmed and cited these in Nature communications output format. The only potentially missing detail to be added in to the manuscript are the issue numbers – which we have now added (marked copy). These may not be automatically included in Nature communications but regardless we have now added these references.